# ADAPTIVE PERSONALIZED FEDERATED LEARNING

## ABSTRACT

Investigation of the degree of personalization in federated learning algorithms has shown that only maximizing the performance of the global model will confine the capacity of the local models to personalize. In this paper, we advocate an adaptive personalized federated learning (APFL) algorithm, where each client will train their local models while contributing to the global model. We derive the generalization bound of mixture of local and global models, and find the optimal mixing parameter. We also propose a communication-efficient optimization method to collaboratively learn the personalized models and analyze its convergence in both smooth strongly convex and nonconvex settings. The extensive experiments demonstrate the effectiveness of our personalization schema, as well as the correctness of established generalization theories.

## 1 INTRODUCTION

With the massive amount of data generated by the proliferation of mobile devices and the internet of things (IoT), coupled with concerns over sharing private information, collaborative machine learning and the use of federated optimization (FO) is often crucial for the deployment of large-scale machine learning (McMahan et al., 2017; Kairouz et al., 2019; Li et al., 2020b). In FO, the ultimate goal is to learn a global model that achieves uniformly good performance over almost all participating clients without sharing raw data. To achieve this goal, most of the existing methods pursue the following procedure to learn a global model: (i) a subset of clients participating in the training is chosen at each round and receive the current copy of the global model; (ii) each chosen client updates the local version of the global model using its own local data, (iii) the server aggregates over the obtained local models to update the global model, and this process continues until convergence (McMahan et al., 2017; Mohri et al., 2019; Karimireddy et al., 2019; Pillutla et al., 2019). Most notably, FedAvg by McMahan et al. (2017) uses averaging as its aggregation method over local models.

Due to inherent diversity among local data shards and highly non-IID distribution of the data across clients, FedAvg is hugely sensitive to its hyperparameters, and as a result, does not benefit from a favorable convergence guarantee (Li et al., 2020c). In Karimireddy et al. (2019), authors argue that if these hyperparameters are not carefully tuned, it will result in the divergence of FedAvg, as local models may drift significantly from each other. Therefore, in the presence of statistical data heterogeneity, the global model might not generalize well on the local data of each client individually (Jiang et al., 2019). This is even more crucial in fairness-critical systems such as medical diagnosis (Li & Wang, 2019), where poor performance on local clients could result in damaging consequences. This problem is exacerbated even further as the diversity among local data of different clients is growing. To better illustrate this fact, we ran a simple experiment on MNIST dataset where each client's local training data is sampled from a subset of classes to simulate heterogeneity. Obviously, when each client has samples from less number of classes of training data, the heterogeneity among them will be high and if each of them has samples from all

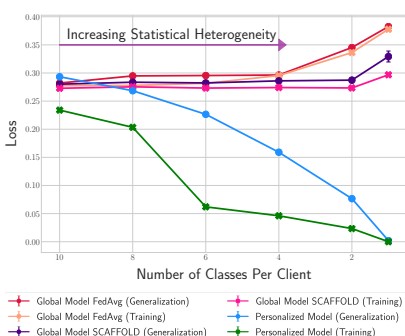

Figure 1: Comparing the generalization and training losses of our proposed personalized model with the global models of FedAvg and SCAFFOLD by increasing the diversity among the data of clients on MNIST dataset with a logistic regression model.

classes, the distribution of their local training data becomes almost identical, and thus heterogeneity will be low. The results of this experiment are depicted in Figure 1, where the generalization and training losses of the global models of the FedAvg (McMahan et al., 2017) and SCAFFOLD (Karimireddy et al., 2019) on local data diverge when the diversity among different clients' data increases. This observation illustrates that solely optimizing for the global model's accuracy leads to a poor generalization of local clients. To embrace statistical heterogeneity and mitigate the effect of negative transfer, it is necessary to integrate the personalization into learning instead of finding a single consensus predictor. This pluralistic solution for FO has recently resulted in significant research in personalized learning schemes (Eichner et al., 2019; Smith et al., 2017; Dinh et al., 2020; Mansour et al., 2020; Fallah et al., 2020; Li et al., 2020a).

To balance the trade-off between the benefit from collaboration with other users and the disadvantage from the statistical heterogeneity among different users' domains, in this paper, we propose an **adaptive personalized federated learning** (APFL) algorithm which aims to learn a personalized model for each device that is a mixture of optimal local and global models. We theoretically analyze the generalization ability of the personalized model on local distributions, with dependency on mixing parameter, the divergence between local and global distributions, as well as the number of local and global training data. To learn the personalized model, we propose a communication efficient optimization algorithm that adaptively learns the model by leveraging the relatedness between local and global models as learning proceeds. As it is shown in Figure 1, by progressively increasing the diversity, the personalized model found by the proposed algorithm demonstrates a better generalization compared to the global models learned by FedAvg and SCAFFOLD. We supplement our theoretical findings with extensive corroborating experimental results that demonstrate the superiority of the proposed personalization schema over the global and localized models of commonly used federated learning algorithms.

## 2 PERSONALIZED FEDERATED LEARNING

In this section, we propose a personalization approach for federated learning and analyze its statistical properties. Following the statistical learning theory, in a federated learning setting each client has access to its own data distribution $\mathcal{D}_i$ on domain $\Xi := \mathcal{X} \times \mathcal{Y}$, where $\mathcal{X} \in \mathbb{R}^d$ is the input domain and $\mathcal{Y}$ is the label domain. For any hypothesis $h \in \mathcal{H}$ the loss function is defined as $\ell : \mathcal{H} \times \Xi \to \mathbb{R}^+$. The true risk at local distribution is denoted by $\mathcal{L}_{\mathcal{D}_i}(h) = \mathbb{E}_{(\boldsymbol{x},y) \sim \mathcal{D}_i}[\ell(h(\boldsymbol{x}), y)]$. We use $\hat{\mathcal{L}}_{\mathcal{D}_i}(h)$ to denote the empirical risk of $h$ on distribution $\mathcal{D}_i$. We use $\bar{\mathcal{D}} = (1/n) \sum_{i=1}^{n} \mathcal{D}_i$ to denote the average distribution over all clients.

### 2.1 PERSONALIZED MODEL

In a standard federated learning scenario, where the goal is to learn a global model for all devices cooperatively, the learned global model obtained by minimizing the joint empirical distribution $\bar{\mathcal{D}}$, i.e., $\min_{h \in \mathcal{H}} \hat{\mathcal{L}}_{\bar{\mathcal{D}}}(h)$ by proper weighting. However, as alluded to before, a single consensus predictor may not perfectly generalize on local distributions when the heterogeneity among local data shards is high (i.e., the global and local optimal models drift significantly). Meanwhile, from the local user perspective, the key incentive to participate in "federated" learning is the desire to seek a reduction in the local generalization error with the help of other users' data. In this case, the ideal situation would be that the user can utilize the information from the global model to compensate for the small number of local training data while minimizing the negative transfer induced by heterogeneity among distributions. This motivates us to mix the global model and local model with a controllable weight as a joint prediction model, namely, the personalized model.

Here we formally introduce our proposed adaptive personalized learning schema, where the goal is to find the optimal combination of the global and the local models, in order to achieve a better client-specific model. In this setting, global server still tries to train the global model by minimizing the empirical risk on the aggregated domain $\bar{\mathcal{D}}$, i.e., $\bar{h}^* = \arg\min_{h \in \mathcal{H}} \hat{\mathcal{L}}_{\bar{\mathcal{D}}}(h)$, while each user trains a local model while partially incorporating the global model, with some mixing weight $\alpha_i$, i.e., $\hat{h}_{loc,i}^* = \arg\min_{h \in \mathcal{H}} \hat{\mathcal{L}}_{\mathcal{D}_i}(\alpha_i h + (1 - \alpha_i)\bar{h}^*)$. Finally, the personalized model for $i$th client is

a convex combination of $\bar{h}^*$ and $\hat{h}^*_{loc,i}$:

$$h_{\alpha_i} = \alpha_i \hat{h}^*_{loc,i} + (1 - \alpha_i)\bar{h}^*, \tag{1}$$

It is worth mentioning that, $h_{\alpha_i}$ is not necessarily the minimizer of empirical risk $\hat{\mathcal{L}}_{\mathcal{D}_i}(\cdot)$, because we optimize $\hat{h}^*_{loc,i}$ with partially incorporating the global model.

**Example 1.** *Let us illustrate a simple situation where mixed model does not necessarily coincide with local ERM model. To this end, consider a setting where the hypothesis class $\mathcal{H}$ is the set of all vectors in $\mathbb{R}^2$, lying in $\ell_2$ unit ball: $\mathcal{H} = \{\boldsymbol{h} \in \mathbb{R}^2 : \|\boldsymbol{h}\|_2 \le 1\}$. Assume the local empirical minimizer is known to be $[1,0]^\top$, and $\bar{\boldsymbol{h}}^* = [-1,0]^\top$, and $\alpha$ is set to be 0.5. Now, if we wish to find a $\hat{\boldsymbol{h}}^*_{loc,i}$, such that $\boldsymbol{h}_{\alpha_i} = \alpha * \hat{\boldsymbol{h}}^*_{loc,i} + (1-\alpha) * \bar{\boldsymbol{h}}^*$ coincides with local empirical minimizer, we have to solve: $0.5 * \boldsymbol{h} + 0.5 * [-1,0]^\top = [1,0]^\top$, subject to $\|\boldsymbol{h}\|_2 \le 1$. This equation has no feasible solution, implying that it is not necessarily true that $\boldsymbol{h}_{\alpha_i}$ coincides with local empirical minimizer.*

In fact, in most cases, as we will show in the convergence of the proposed algorithm, $h_{\alpha_i}$ will incur a residual risk if evaluated on the training set drawn from $\mathcal{D}_i$.

## 2.2 GENERALIZATION GUARANTEES

We now characterize the generalization of the mixed model. We present the learning bounds for classification and regression tasks. For classification, we consider a binary classification task, with squared hinge loss $\ell(h(\boldsymbol{x}), y) = (\max\{0, 1 - yh(\boldsymbol{x})\})^2$. In the regression task, we consider the MSE loss $\ell(h(\boldsymbol{x}), y) = (h(\boldsymbol{x}) - y)^2$. Even though we present learning bounds under these two loss functions, our analysis can be generalized to any convex smooth loss. Before formally presenting the generalization bound, we introduce the following quantity to measure the empirical complexity of a hypothesis class $\mathcal{H}$ over a training set $\mathcal{S}$.

**Definition 1.** *Let $S$ be a fixed set of samples and consider a hypothesis class $\mathcal{H}$. The worst case disagreement between a pair of models measured by absolute loss is quantified by: $\lambda_{\mathcal{H}}(S) = \sup_{h,h' \in \mathcal{H}} \frac{1}{|S|} \sum_{(\boldsymbol{x},y) \in S} |h(\boldsymbol{x}) - h'(\boldsymbol{x})|$.*

The empirical discrepancy characterizes the complexity of hypothesis class over some finite set. The similar concepts are also employed in the related multiple source PAC learning or domain adaption (Kifer et al., 2004; Mansour et al., 2009; Ben-David et al., 2010; Konstantinov et al., 2020; Zhang et al., 2020).

We now state the main result on the generalization of the proposed personalization schema. The proof of the theorem is provided in Appendix D.

**Theorem 1.** *Let hypothesis class $\mathcal{H}$ be compact closed set with finite VC dimension $d$. Assume loss function $\ell$ is Lipschitz continuous with constant $G$, and bounded in $[0, B]$. Then with probability at least $1-\delta$, there exists a constant $C$, such that the risk of the mixed model $h_{\alpha_i} = \alpha_i \hat{h}^*_{loc,i} + (1-\alpha_i)\bar{h}^*$ on the $i$th local distribution $\mathcal{D}_i$ is bounded by:*

$$
\begin{aligned}
\mathcal{L}_{\mathcal{D}_i}(h_{\alpha_i}) \le\ & 2\alpha_i^2 \left( \mathcal{L}_{\mathcal{D}_i}(h_i^*) + 2C\sqrt{\frac{d + \log(1/\delta)}{m_i}} + G\lambda_{\mathcal{H}}(\mathcal{S}_i) \right) \\
& + 2(1-\alpha_i)^2 \left( \hat{\mathcal{L}}_{\bar{\mathcal{D}}}(\bar{h}^*) + B\|\bar{\mathcal{D}} - \mathcal{D}_i\|_1 + C\sqrt{\frac{d + \log(1/\delta)}{m}} \right),
\end{aligned}
\tag{2}
$$

*where $m_i, i = 1, 2, \ldots, n$ is the number of training data at $i$th user, $m = m_1 + \ldots + m_n$ is the total number of all data, $\mathcal{S}_i$ to be the local training set drawn from $\mathcal{D}_i$, $\|\bar{\mathcal{D}} - \mathcal{D}_i\|_1 = \int_\Xi |\mathbb{P}_{(\boldsymbol{x},y)\sim\bar{\mathcal{D}}} - \mathbb{P}_{(\boldsymbol{x},y)\sim\mathcal{D}_i}| d\boldsymbol{x}dy$, is the difference between distributions $\bar{\mathcal{D}}$ and $\mathcal{D}_i$, and $h_i^* = \arg\min_{h\in\mathcal{H}} \mathcal{L}_{\mathcal{D}_i}(h)$.*

**Remark 1.** *We note that a very analogous work to ours is Mansour et al. (2020), where a generalization bound is provided for mixing global and local models. However, their bound does not depend on $\alpha_i$, and hence we cannot see how it impacts the generalization ability.*

In Theorem 1, by omitting constant terms, we observe that the generalization risk of $h_{\alpha_i}$ on $\mathcal{D}_i$ mainly depends on three key quantities: i) $m$: the number of global data drawn from $\bar{\mathcal{D}}$, ii) divergence between distributions $\bar{\mathcal{D}}$ and $\mathcal{D}_i$, and iii) $m_i$: the amount of local data drawn from $\mathcal{D}_i$. Usually,

the first quantity $m$, the amount of global data is fairly large compared to individual users, so global model usually has a better generalization. The second quantity characterizes the data heterogeneity between the average distribution and $i$th local distribution. If this divergence is too high, then the global model may hurt the local generalization. For the third quantity, as amount of local data $m_i$ is often small, the generalization performance of local model can be poor.

**Optimal mixing parameter.** We can also find the optimal mixing parameter $\alpha_i^*$ that minimizes generalization bound in Theorem 1. Notice that the RHS of (2) is quadratic in $\alpha_i$, so it admits a minimum value at

$$
\alpha_i^* = \frac{\left( \hat{\mathcal{L}}_{\bar{\mathcal{D}}}(\bar{h}^*) + B\|\bar{\mathcal{D}} - \mathcal{D}_i\|_1 + C\sqrt{\frac{d + \log(1/\delta)}{m}} \right)}{\left( \hat{\mathcal{L}}_{\bar{\mathcal{D}}}(\bar{h}^*) + B\|\bar{\mathcal{D}} - \mathcal{D}_i\|_1 + C\sqrt{\frac{d + \log(1/\delta)}{m}} \right) + \left( \mathcal{L}_{\mathcal{D}_i}(h_i^*) + 2C\sqrt{\frac{d + \log(1/\delta)}{m_i}} + G\lambda_{\mathcal{H}}(\mathcal{S}_i) \right)}.
$$

The optimal mixture parameter is strictly bounded in $[0, 1]$, which matches our intuition. If the divergence term is large, then the value becomes close to 1, which implies if local distribution drifts too much from average distribution, it is preferable to take more local models. If $m_i$ is small, this value will be negligible, indicating that we need to mix more of the global model into the personalized model. Conversely, if $m_i$ is large, then this term will be again roughly 1, which means taking the majority of local model will give the desired generalization performance.

## 3 OPTIMIZATION METHOD

To optimize the learning problem we cast in the previous section, here we propose a communication efficient adaptive algorithm to learn the personalized local models and the global model. To do so, we let every hypothesis $h$ in the hypothesis space $\mathcal{H}$ to be parameterized by a vector $\boldsymbol{w} \in \mathcal{W} \subset \mathbb{R}^d$ where $\mathcal{W}$ is some convex closed set and denote the empirical risk at $i$th device by local objective function $f_i(\boldsymbol{w})$. Adaptive personalized federated learning can be formulated as a two-phase optimization problem: globally update the shared model, and locally update users' local models. Similar to FedAvg algorithm, the server will solve the following optimization problem:

$$
\min_{\boldsymbol{w} \in \mathcal{W}} \left[ F(\boldsymbol{w}) := \frac{1}{n} \sum_{i=1}^{n} \{ f_i(\boldsymbol{w}) := \mathbb{E}_{\xi_i} [f_i(\boldsymbol{w}, \xi_i)] \} \right], \tag{3}
$$

where $f_i(.)$ is the local objective at $i$th client, $\xi_i$ is a minibatch of data in data shard at $i$th client, and $n$ is the total number of clients. Motivated by the trade-off between the global model and local model generalization errors in Theorem 1, we need to learn a personalized model as in (1) to optimize the local empirical risk. To this end, each client needs to solve this optimization over its local data:

$$
\min_{\boldsymbol{v} \in \mathcal{W}} f_i \left( \alpha_i \boldsymbol{v} + (1 - \alpha_i) \boldsymbol{w}^* \right), \tag{4}
$$

where $\boldsymbol{w}^* = \arg\min_{\boldsymbol{w}} F(\boldsymbol{w})$ is the optimal global model. The balance between these two models is governed by a parameter $\alpha_i$, which is associated with the diversity of the local model and the global model. We first state the algorithm for a pre-defined proper $\alpha_i$, and then propose an adaptive schema to learn this parameter as learning proceeds.

**Remark 2.** *As mentioned in Section 2.1, when the hypothesis class is bounded, the mixed model will not coincide with local ERM model. However, if the class is unbounded, the mixed model will eventually converge to local ERM model, which means the personalization fails. Hence, to make sure the correctness of our algorithm, we need to require the parameter comes from some bounded domain $\mathcal{W}$*

**Local Descent APFL.** To efficiently optimize the problem we cast in (3) and (4), in this subsection we propose our bilevel optimization algorithm, Local Descent APFL. At each communication round, server uniformly random selects $K$ clients as a set $U_t$. Each selected client will maintain three models at iteration $t$: local version of the global model $\boldsymbol{w}_i^{(t)}$, its own local model $\boldsymbol{v}_i^{(t)}$, and the mixed personalized model $\bar{\boldsymbol{v}}_i^{(t)} = \alpha_i \boldsymbol{v}_i^{(t)} + (1 - \alpha_i) \boldsymbol{w}_i^{(t)}$. Then, selected clients will perform the following updates locally on their own data for $\tau$ iterations:

$$
\boldsymbol{w}_i^{(t)} = \prod_{\mathcal{W}} \left( \boldsymbol{w}_i^{(t-1)} - \eta_t \nabla f_i \left( \boldsymbol{w}_i^{(t-1)}; \xi_i^t \right) \right), \quad \boldsymbol{v}_i^{(t)} = \prod_{\mathcal{W}} \left( \boldsymbol{v}_i^{(t-1)} - \eta_t \nabla_{\boldsymbol{v}} f_i \left( \bar{\boldsymbol{v}}_i^{(t-1)}; \xi_i^t \right) \right), \tag{5}
$$

---

**Algorithm 1:** Local Descent `APFL`

---

**input:** Mixture weights $\alpha_1, \cdots, \alpha_n$, Synchronization gap $\tau$.

**for** $t = 0, \cdots, T$ **do**

    **parallel for** $i \in U_t$ **do**

        **if** *t not divides $\tau$* **then**

$$\boldsymbol{w}_i^{(t)} = \prod_{\mathcal{W}} \left( \boldsymbol{w}_i^{(t-1)} - \eta_t \nabla f_i \left( \boldsymbol{w}_i^{(t-1)}; \xi_i^t \right) \right),$$

$$\boldsymbol{v}_i^{(t)} = \prod_{\mathcal{W}} \left( \boldsymbol{v}_i^{(t-1)} - \eta_t \nabla_{\boldsymbol{v}} f_i \left( \bar{\boldsymbol{v}}_i^{(t-1)}; \xi_i^t \right) \right)$$

$$\bar{\boldsymbol{v}}_i^{(t)} = \alpha_i \boldsymbol{v}_i^{(t)} + (1 - \alpha_i) \boldsymbol{w}_i^{(t)}, U_t \leftarrow U_{t-1}$$

        **else**

            each selected client sends $\boldsymbol{w}_i^{(t)}$ to the server

            $\boldsymbol{w}^{(t)} = \frac{1}{|U_t|} \sum_{j \in U_t} \boldsymbol{w}_j^{(t)}$

            server uniformly samples a subset $U_t$ of $K$ clients.

            server broadcast $\boldsymbol{w}^{(t)}$ to all chosen clients

        **end**

    **end**

**end**

---

where $\nabla f_i (.; \xi)$ denotes the stochastic gradient of $f(.)$ evaluated at mini-batch $\xi$. Then, using the updated version of the global model and the local model, we update the personalized model $\bar{\boldsymbol{v}}_i^{(t)}$ as well. The clients that are not selected in this round will keep their previous step local model $\boldsymbol{v}_i^{(t)} = \boldsymbol{v}_i^{(t-1)}$. After these $\tau$ local updates, selected clients will send their local version of the global model $\boldsymbol{w}_i^{(t)}$ to the server for aggregation by averaging: $\boldsymbol{w}^{(t)} = \frac{1}{|U_t|} \sum_{j \in U_t} \boldsymbol{w}_j^{(t)}$. Then the server will choose another set of $K$ clients for the next round of training and broadcast this new model to them.

**Adaptively updating $\alpha$.** Even though in Section 2.2, we give the information theoretically optimal mixing parameter, in practice we usually do not know the distance between user's distribution and the average distribution. Thus, finding the optimal $\alpha$ is infeasible. However, we can infer it empirically during optimization. Based on the local objective defined in (4), the empirical optimum value of $\alpha$ for each client can be found by solving $\alpha_i^* = \arg\min_{\alpha_i \in [0,1]} f_i (\alpha_i \boldsymbol{v} + (1 - \alpha_i)\boldsymbol{w})$, where we can use the gradient descent to optimize it at every communication round, using the following step:

$$\alpha_i^{(t)} = \alpha_i^{(t-1)} - \eta_t \nabla_\alpha f_i \left( \bar{\boldsymbol{v}}_i^{(t-1)}; \xi_i^t \right) = \alpha_i^{(t-1)} - \eta_t \left\langle \boldsymbol{v}_i^{(t-1)} - \boldsymbol{w}_i^{(t-1)}, \nabla f_i \left( \bar{\boldsymbol{v}}_i^{(t-1)}; \xi_i^t \right) \right\rangle, \quad (6)$$

which shows that the mixing coefficient $\alpha$ is updated based on the correlation between the difference of the personalized and the local version of global models, and the gradient at the in-device personalized model. Meaning, when the global model is drifting from the personalized model, the value of $\alpha$ changes to adjust the balance between local data and shared knowledge among all devices captured by the global model.

## 4 CONVERGENCE ANALYSIS

In this section we provide the convergence analysis of Local Descent APFL with fixed $\alpha_i$ for strongly convex and nonconvex functions. To have a tight analysis, as well as putting the optimization results in the context of generalization bounds discussed above, we define the following parameterization-invariant quantities that only depend on the distributions of local data across clients and the geometry of loss functions.

**Definition 2.** *We define the following quantity to measure the diversity among local gradients with respect to the gradient of the ith client:* $\zeta_i = \sup_{\boldsymbol{w} \in \mathbb{R}^d} \|\nabla F(\boldsymbol{w}) - \nabla f_i(\boldsymbol{w})\|_2^2$ *(Woodworth et al., 2020a). We also define the sum of gradient diversities of $n$ clients as:* $\zeta = \sum_{i=1}^n \zeta_i$.

**Definition 3.** *We define* $\Delta_i = \|\boldsymbol{v}_i^* - \boldsymbol{w}^*\|_2^2$, *where* $\boldsymbol{v}_i^* = \arg\min_{\boldsymbol{v}} f_i(\boldsymbol{v})$, *and* $\boldsymbol{w}^* = \arg\min_{\boldsymbol{w}} F(\boldsymbol{w})$ *to measure the gap between optimal local model and optimal global model.*

We also need the following standard assumption on the stochastic gradients at local objectives.

**Assumption 1** (Bounded Variance). *The variance of stochastic gradients computed at each local data shard is bounded, i.e., $\forall i \in [n]$:$\mathbb{E}[\|\nabla f_i(\boldsymbol{x}; \xi) - \nabla f_i(\boldsymbol{x})\|^2] \leq \sigma^2$.*

**Strongly Convex Loss.** We now turn to establishing the convergence of local descent `APFL` on smooth strongly convex functions. Specifically, the following theorem characterizes the convergence of the personalized local model to the optimal local model. The proof is provided in Appendix E.2.3.

**Theorem 2.** *Assume each client's objective function is $\mu$-strongly convex and $L$-smooth, and satisfies Assumption 1. Also let $\kappa = L/\mu$, $b = \min\left\{\frac{K}{n}, \frac{1}{2}\right\}$. Using Algorithm 1, by choosing the mixing weight $\alpha_i \geq \max\{1 - \frac{1}{4\sqrt{6}\kappa}, 1 - \frac{1}{4\sqrt{6}\kappa\sqrt{\mu}}\}$, learning rate: $\eta_t = \frac{16}{\mu(t+a)}$, where $a = \max\{128\kappa, \tau\}$, and using average scheme $\hat{\boldsymbol{v}}_i = \frac{1}{S_T} \sum_{t=1}^{T} p_t(\alpha_i \boldsymbol{v}_i^{(t)} + (1-\alpha_i)\frac{1}{K} \sum_{j \in U_t} \boldsymbol{w}_j^{(t)})$, where $p_t = (t+a)^2$, $S_T = \sum_{t=1}^{T} p_t$, and letting $f_i^*$ to denote the local minimum of the $i$th client, then the following convergence rate holds for all clients $i \in [n]$:*

$$\mathbb{E}[f_i(\hat{\boldsymbol{v}}_i)] - f_i^* \leq \alpha_i^2 O\left(\frac{\sigma^2}{\mu b T}\right) + (1-\alpha_i)^2 O\left(\frac{\kappa^2 \sigma^2}{\mu b K T} + \frac{\kappa^2 \tau\left(\tau \zeta_i + \kappa^2 \tau \frac{\zeta}{K}\right)}{\mu b T^2} + \frac{\zeta_i + \frac{\zeta}{K}}{\mu b} + \frac{\kappa L \Delta_i}{b}\right).$$

*If we choose $\tau = \sqrt{T/K}$, then:*

$$\mathbb{E}[f_i(\hat{\boldsymbol{v}}_i)] - f_i^* \leq \alpha_i^2 O\left(\frac{\sigma^2}{\mu T}\right) + (1-\alpha_i)^2 O\left(\frac{\kappa^2 \sigma^2 + \kappa^2 \zeta_i + \kappa^4 \frac{\zeta}{K}}{\mu K T}\right) + (1-\alpha_i)^2 O\left(\frac{\zeta_i + \frac{\zeta}{K}}{\mu} + \kappa L \Delta_i\right).$$

A few remarks about the convergence of personalized local model are in place: (1) If we set $\alpha_i = 1$, then we recover $O\left(\frac{1}{T}\right)$ convergence rate of single machine SGD. If we only focus on the terms with $(1-\alpha_i)^2$, which is contributed by the global model's convergence, and omit the residual error, we achieve the convergence rate of $O(1/KT)$ using only $\sqrt{KT}$ communication, which matches with the convergence rate of vanilla local SGD (Stich, 2018; Woodworth et al., 2020a), and (2) The residual error is related to the gradient diversity $\zeta_i$ and local-global optimality gap $\Delta_i$. It shows that taking any proportion of the global model will result in a sub-optimal ERM model. As we discussed in Section 2.1, $h_{\alpha_i}$ will not be the empirical risk minimizer in most cases. Also, we assume that $\alpha_i$ needs to be larger than some value in order to get a tight rate. This condition can be alleviated, but the residual error will be looser. The analysis of this relaxation is presented in Appendix F.

**Nonconvex Loss.** The following theorem establish the convergence rate of personalized model learned by `APFL` for nonconvex smooth loss functions. The proof is provided in Appendix G.3.

**Theorem 3.** *Let $\hat{\boldsymbol{v}}_i^{(t)} = \alpha_i \boldsymbol{v}_i^{(t)} + (1-\alpha_i)\frac{1}{K} \sum_{j \in U_t} \boldsymbol{w}_j^{(t)}$. If each client's objective function is $L$-smooth and domain $\mathcal{W}$ be bounded by $D_{\mathcal{W}}$, that is, $\forall \boldsymbol{w}, \boldsymbol{w}' \in \mathcal{W}, \|\boldsymbol{w} - \boldsymbol{w}'\|^2 \leq D_{\mathcal{W}}$. Using Algorithm 1 with full gradient, by choosing $K = n$ and learning rate $\eta = \frac{1}{2\sqrt{5}L\sqrt{T}}$, we have*

$$\frac{1}{T} \sum_{t=1}^{T} \left\|\nabla f_i(\hat{\boldsymbol{v}}_i^{(t)})\right\|^2 \leq O\left(\frac{L}{\sqrt{T}}\right) + (1-\alpha_i)^2 O\left(\frac{L}{\sqrt{T}}\right) + (1-\alpha_i^2)^2 \left(\zeta_i + L^2 D_{\mathcal{W}}\right)$$

$$+ \alpha_i^4 (1-\alpha_i)^2 O\left(\frac{\tau^4 \zeta}{n T^2} + \frac{\tau^2 \zeta_i}{T}\right) + (1-\alpha_i)^2 O\left(\frac{\tau^2 \zeta}{n T}\right).$$

*By choosing $\tau = n^{-1/4} T^{1/4}$, it holds that:*

$$\frac{1}{T} \sum_{t=1}^{T} \left\|\nabla f_i(\hat{\boldsymbol{v}}_i^{(t)})\right\|^2 \leq O\left(\frac{1}{\sqrt{T}}\right) + (1-\alpha_i)^2 O\left(\frac{1}{\sqrt{T}} + \frac{1}{\sqrt{nT}}\right) + (1-\alpha_i^2)^2 \left(\zeta_i + L^2 D_{\mathcal{W}}\right).$$

Here we show that APFL will converge to stationary point on nonconvex function with sublinear rate plus some residual error, with $n^{3/4} T^{3/4}$ communication rounds. The rate with factor $(1-\alpha_i)^2$ is contributed from the global model convergence, and here we have some additive residual error reflected by $\zeta_i$ and $D_{\mathcal{W}}$. Compared to most related work by Haddadpour & Mahdavi (2019) regarding the convergence of local SGD on nonconvex functions, they obtain $O(1/\sqrt{nT})$, while we only have speedup in $n$ on partial terms. This could be solved by using different learning rate for local and global update. Additionally, we assume $K = n$ to derive the convergence in nonconvex setting, and leave the analysis for partial participation as a future work.

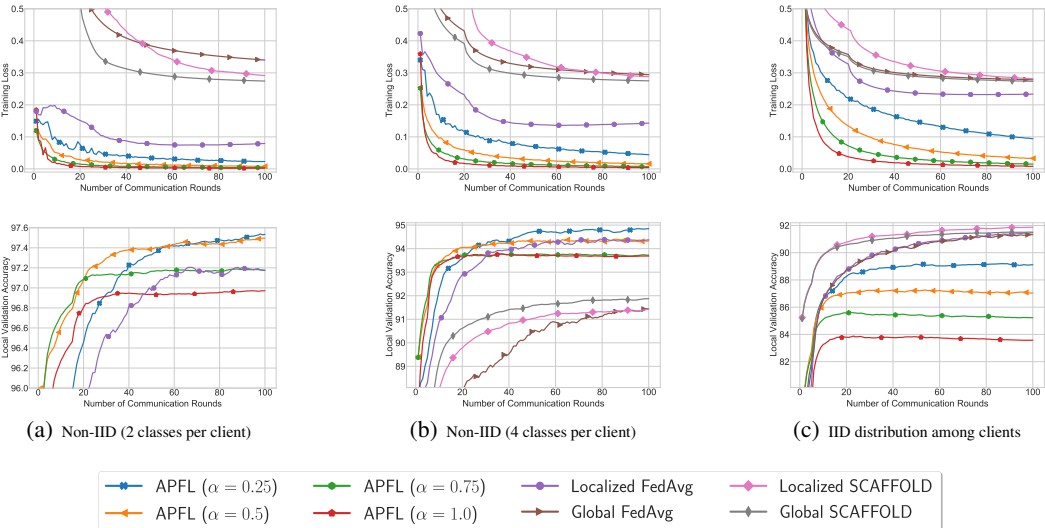

Figure 2: Comparing the the performance of APFL with FedAvg (APFL with $\alpha = 0$) and SCAFFOLD on the MNIST dataset. Top row is the training loss and the bottom row is the generalization accuracy on training and validation data, respectively. In (a), the accuracy lines of SCAFFOLD and FedAvg global models are removed since their low values degrade the readability of the plot.

## 5 EXPERIMENTS

In this section, we empirically show the effectiveness of the proposed algorithm in personalized federated learning. Due to lack of space, some experimental results are deferred to Appendix B.

**Experimental setup.** We run our experiments on Microsoft Azure systems, using Azure ML API. The code is developed on PyTorch (Paszke et al., 2019) using its "distributed" API with MPI. We deploy this code on Standard F64s family of VMs in Azure. We use four datasets for our experiments, MNIST, CIFAR10 (Krizhevsky et al., 2009), EMNIST (Cohen et al., 2017), and a synthetic dataset. For more information on datasets used in the following experiments refer to Appendix B.1. For all the experiments, we have 100 users (except for EMNIST dataset), each of which has access to its own data only. The local dataset is randomly divided into 80% for training and 20% for validation, which is the standard way to examine the local models for personalized use cases. For the learning rate, we use the linear decay structure with respect to local steps, suggested by Bottou (2012). At each iteration the learning rate is decreased by 1%, unless otherwise stated. We report the performance over training data for optimization error and local validation data (from the same distribution as training data for each client) for the generalization accuracy. Throughout these experiments we report the results for the following three models:

- **Global Model**: Referring to the global model of FedAvg or SCAFFOLD.

- **Localized Global Model**: Referring to the fine-tuned version of the global model at each round of communication after $\tau$ steps of local SGD. Here, we have either the localized FedAvg or the localized SCAFFOLD. The reported results are for the average of the performance over all the local models on each online client. In all the experiments $\tau = 10$, unless otherwise stated.

- **Personalized Model**: This model is the personalized model produced by our proposed algorithm APFL. The reported results are the average of the respective performance of personalized models over all online clients at each round of communication.

**Strongly convex loss.** First, we run a set of experiments on the MNIST dataset, with different levels of non-IIDness by assigning certain number of classes to each client. We use logistic regression with parameter regularization as our strongly convex loss function. In this part, all clients are online for each round, however, the results when client sampling is involved is discussed in Appendix B.2. We compare the personalized model of APFL with different rates of personalization as $\alpha$ with global and localized models of FedAvg and SCAFFOLD, as well as their global models. The initial learning

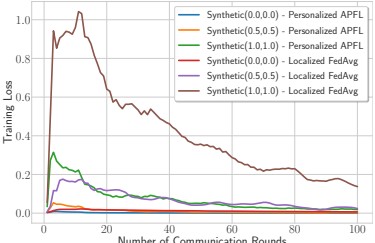 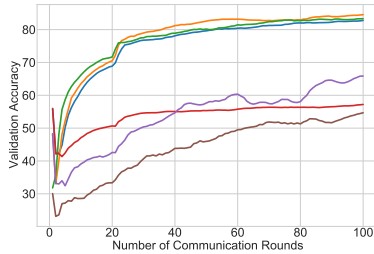

Figure 3: Comparing the APFL with adaptive $\alpha$ and the localized FedAvg. The left figure is the training performance, and the right one is the accuracy of these models on local validation data.

rate is set to $0.1$ and it is decaying as mentioned before. The results of running this experiment on 100 clients and after 100 rounds of communication are depicted in Figure 2, where we move from highly non-IID data distribution (left) to IID data distribution (right). As it can be seen, global models learned by FedAvg and SCAFFOLD have high local training losses. On the other hand, taking more proportion of the local model in the personalized model (namely, increasing $\alpha$) will result in the lower training losses. For generalization ability, the best performance is given by personalized model with $\alpha = 0.25$ in both (a) and (b) cases, which outperforms the global (FedAvg and SCAFFOLD) and their localized versions. However, as we move toward IID distribution, the advantage of personalization vanishes as expected. Hence, as expected by the theoretical findings, we can benefit from personalization the most when there is a statistical heterogeneity between the data of different clients. When the data are distributed IID, local models of FedAvg or SCAFFOLD are preferable.

An interesting observation from the results in Figure 2, which is inline with our theoretical findings is the relationship of $\alpha$ with both optimization and generalization losses. As it can be seen from the first row, $\alpha$ has a linear relationship with the optimization loss, that is, with smaller $\alpha$, training loss is getting closer to the global model of FedAvg in terms of optimization loss, which matches with our convergence theory. However, from the second row, it can be inferred that there is no linear relationship between $\alpha$ and generalization. In fact, according to (2), we know that generalization bound is quadratic in $\alpha$, and hence, the generalization performance does not simply increase or decrease monotonically with $\alpha$.

**Adaptive $\alpha$ update.** In this part, we want to show how adaptively learning the value of $\alpha$ across different clients, based on (6), will affect the training and generalization performance of APFL's personalized models. We use the three synthetic datasets as described in Appendix B.1, with logistic regression as the loss function. We set the initial value of $\alpha_i^{(0)} = 0.01$ for every $i \in [n]$. The results of this training are depicted in Figure 3, where both optimization and generalization of the learned models are compared. As it can be inferred, in training, APFL outperforms FedAvg in the same datasets. More interestingly, in generalization of learned APFL personalized models, all datasets achieve almost the same performance as a result of adaptively updating $\alpha$ values, while the FedAvg algorithm has a huge gap with them. This shows that, when we do not know the degree of diversity among data of different clients, we should adaptively update $\alpha$ values to guarantee the best generalization performance. We also have results on EMNIST dataset with adaptive tuning of $\alpha$ in Appendix B.2, wih a 2-layer MLP.

**Nonconvex loss.** To showcase the results for a nonconvex loss, we use CIFAR10 dataset that is distributed in a non-IID way with 2 classes per client. We apply it to a CNN model with 2 convolution layers, followed by 2 fully connected layers, using cross entropy as the loss function. The initial learning rates of APFL and FedAvg algorithms are set to $0.1$ with the mentioned decay structure, while for SCAFFOLD this value is $0.05$ with $5\%$ decay per iteration to avoid divergence. As it can be inferred from the results in Table 1, the personalized model learned by APFL outperforms the localized models of FedAvg and SCAFFOLD, as well as their global models, in both optimization and generalization. In this case adaptively tuning the $\alpha$ achieves the best training loss, while $\alpha = 0.25$ case reaching the best generalization performance.

**Comparison with other personalization methods.** We now compare our proposed APFL with two recent approaches for personalization in federated learning. In addition to FedAvg, we compare with perFedAvg introduced in Fallah et al. (2020) using a meta-learning approach, and pFedMe

| | APFL | | | | FedAvg | | SCAFFOLD | |
|---|---|---|---|---|---|---|---|---|
| | $\alpha = 0.25$ | $\alpha = 0.5$ | $\alpha = 0.75$ | Adaptive $\alpha$ | Global Model | Localized Model | Global Model | Localized Model |
| Training Loss | 0.154± 0.003 | 0.113± 0.008 | 0.103± 0.007 | **0.101± 0.013** | 1.789± 0.004 | 0.369± 0.005 | 1.70± 0.001 | 0.593± 0.012 |
| Validation Accuracy | **89.33%± 0.26%** | 88.74%± 0.14% | 89.04%± 0.22% | 88.87%± 0.51% | 32.51%± 0.47% | 83.16%± 0.37% | 37.16%± 0.3% | 85.25%± 0.2% |

Table 1: The results of training a CNN model on CIFAR10 dataset using different algorithms.

introduced in Dinh et al. (2020) using a regularization with Moreau envelope function. We run these algorithms to train an MLP with 2 hidden layers, each with 200 neurons, on a non-IID MNIST dataset with 2 classes per client. For perFedAvg, similar to their setting, we use learning rates of $\alpha = 0.01$ (different from the $\alpha$ in our APFL) and $\beta = 0.001$. To have a fair comparison, we use the same validation for perFedAvg and we use $10\%$ of training data as the test dataset that updates the meta-model. For pFedMe, following their setting, we use $\lambda = 15$, $\eta = 0.01$. We use $\tau = 20$ with total number of communications to 100 and the batch size is 20. The results of these experiments are presented in Table 2, where APFL clearly outperforms all other models in both training and generalization. The APFL model with $\alpha = 0.75$ has the lowest training loss, and the one with adaptive $\alpha$ has the best validation accuracy. perFedAvg is slightly better than the localized FedAvg, however, it is worse than APFL models. pFedMe performs better than the global model of FedAvg, but it cannot surpass neither the localized model of FedAvg nor APFL models.

| | APFL | | | | FedAvg | | perFedAvg | pFedMe |
|---|---|---|---|---|---|---|---|---|
| | $\alpha = 0.25$ | $\alpha = 0.5$ | $\alpha = 0.75$ | Adaptive $\alpha$ | Global Model | Localized Model | Personalized Model | Personalized Model |
| Training Loss | 0.011± 0.0007 | 0.004± 0.0004 | **0.002± 0.0001** | 0.004± 0.0008 | 0.240± 0.006 | 0.041± 0.002 | 0.039± 0.002 | 0.182± 0.004 |
| Validation Accuracy | 98.07%± 0.10% | 98.04%± 0.08% | 97.86%± 0.09% | **98.10%± 0.10%** | 93.81%± 0.29% | 97.75%± 0.15% | 97.83%± 0.12% | 95.92%± 0.1% |

Table 2: The results of training an MLP on MNIST dataset with different personalization methods.

# 6 CONCLUSIONS

In this paper, we proposed an adaptive federated learning algorithm that learns a mixture of local and global models as the personalized model. Motivated by learning theory in domain adaptation, we provided generalization guarantees for our algorithm that demonstrated the dependence on the diversity between each clients' data distribution and the representative sample of the overall distribution of data, and the number of per-device samples as key factors in personalization. Moreover, we proposed a communication-reduced optimization algorithm to learn the personalized models and analyzed its convergence rate for both smooth strongly convex and nonconvex functions. Finally, we empirically backed up our theoretical results by conducting experiments in a federated setting.

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

# Supplementary Material:
# Adaptive Personalized Federated Learning

## Table of Contents

## A  Additional Related Work

The number of research in federated learning is proliferating during the past few years. In federated learning, the main objective is to learn a global model that is good enough for yet to be seen data and has fast convergence to a local optimum. This indicates that there are several uncanny resemblances between federated learning and meta-learning approaches (Finn et al., 2017; Nichol et al., 2018). However, despite this similarity, meta-learning approaches are mainly trying to learn multiple models, personalized for each new task, whereas in most federated learning approaches, the main focus is on the single global model. As discussed by Kairouz et al. (2019), the gap between the performance of global and personalized models shows the crucial importance of personalization in federated learning. Several different approaches are trying to personalize the global model, primarily focusing on optimization error, while the main challenge with personalization is during the inference time. Some of these works on the personalization of models in a decentralized setting can be found in Vanhaesebrouck et al. (2017); Almeida & Xavier (2018), where in addition to the optimization error, they have network constraints or peer-to-peer communication limitation (Bellet et al., 2017; Zantedeschi et al., 2019). In general, as discussed by Kairouz et al. (2019), there are three significant categories of personalization methods in federated learning, namely, local fine-tuning, multi-task learning, and contextualization. Yu et al. (2020) argue that the global model learned by federated learning, especially with having differential privacy and robust learning objectives, can hurt the performance of many clients. They indicate that those clients can obtain a better model by using only their own data. Hence, they empirically show that using these three approaches can boost the performance of those clients. In addition to these three, there is also another category that fits the most to our proposed approach, which is mixing the global and local models.

**Local fine-tuning:** The dominant approach for personalization is local fine-tuning, where each client receives a global model and tune it using its own local data and several gradient descent steps. This approach is predominantly used in meta-learning methods such as MAML by Finn et al. (2017) or domain adaptation and transfer learning (Ben-David et al., 2010; Mansour et al., 2009; Pan & Yang, 2009). Jiang et al. (2019) discuss the similarity between federated learning and meta-learning approaches, notably the Reptile algorithm by Nichol et al. (2018) and FedAvg, and combine them to personalize local models. They observed that federated learning with a single objective of performance of the global model could limit the capacity of the learned model for personalization. In Khodak et al. (2019), authors using online convex optimization to introduce a meta-learning approach that can be used in federated learning for better personalization. Fallah et al. (2020) borrow ideas from MAML to learn personalized models for each client with convergence guarantees. Similar to fine-tuning, they update the local models with several gradient steps, but they use second-order information to update the global model, like MAML. Another approach adopted for deep neural networks is introduced by Arivazhagan et al. (2019), where they freeze the base layers and only change the last "personalized" layer for each client locally. The main drawback of local fine-tuning is that it minimizes the optimization error, whereas the more important part is the generalization performance of the personalized model. In this setting, the personalized model is pruned to overfit.

**Multi-task learning:** Another view of the personalization problem is to see it as a multi-task learning problem similar to Smith et al. (2017). In this setting, optimization on each client can be considered as a new task; hence, the approaches of multi-task learning can be applied. One other approach, discussed as an open problem in Kairouz et al. (2019), is to cluster groups of clients based on some features such as region, as similar tasks, similar to one approach proposed by Mansour et al. (2020).

**Contextualization:** An important application of personalization in federated learning is using the model under different contexts. For instance, in the next character recognition task in Hard et al. (2018), based on the context of the use case, the results should be different. Hence, we need a personalized model on one client under different contexts. This requires access to more features about the context during the training. Evaluation of the personalized model in such a setting has been investigated by Wang et al. (2019), which is in line with our approach in experimental results in Section 5. Liang et al. (2020) propose to directly learn the feature representation locally, and train the discriminator globally, which reduces the effect of data heterogeneity and ensures the fair learning.

**Personalization via model regularization:** Another significant trial for personalization is model regularization. There are several studies to introduce different personalization approaches for federated learning by regularize the difference between the global and local models. Hanzely & Richtárik (2020) try to introduce a new formulation for federated learning where they add the regularization term on the distance of local and global models. In their effort, they use a mixing parameter, which controls the degree of optimization for both local models and the global model. The FedAvg (McMahan et al., 2017) can be considered a special case of this approach. They show that the learned model is in the convex haul of both local and global models, and at each iteration, depend on the local models' optimization parameters, the global model is getting closer to the global model learned by FedAvg. Similarly, Huang et al. (2020) and Dinh et al. (2020) also propose to use the regularization between local and global model, to realize the personalized learning. Shen et al. (2020) propose a knowledge distillation way to achieve personalization, where they apply the regularization on the predictions between local model and global model.

**Personalization via model interpolation:** Parallel to our work, there are other studies to introduce different personalization approaches for federated learning by mixing the global and local models. The closest approach for personalization to our proposal is introduced by Mansour et al. (2020). In fact, they propose three different approaches for personalization with generalization guarantees, namely, client clustering, data interpolation, and model interpolation. Out of these three, the first two approaches need some meta-features from all clients that makes them not a feasible approach for federated learning, due to privacy concerns. The third schema, which is the most promising one in practice as well, has a close formulation to ours in the interpolation of the local and global models. However, in their theory, the generalization bound does not demonstrate the advantage of mixing models, but in our analysis, we show how the model mixing can impact the generalization

bound, by presenting its dependency on the mixture parameter, data diversity and optimal models on local and global distributions.

Beyond different techniques for personalization in federated learning, Kairouz et al. (2019) ask an essential question of "*when is a global FL-trained model better?*", or as we can ask, when is personalization better? The answer to these questions mostly depends on the distribution of data across clients. As we theoretically prove and empirically verify in this paper, when the data is distributed IID, we cannot benefit from personalization, and it is similar to the local SGD scenario (Stich, 2018; Haddadpour et al., 2019a;b; Woodworth et al., 2020b). However, when the data is non-IID across clients, which is mostly the case in federated learning, personalization can help to balance between shared and local knowledge. Then, the question becomes, what degree of personalization is best for each client? While this was an open problem in Mohri et al. (2019) on how to appropriately mix the global and local model, we answer this question by adaptively tuning the degree of personalization for each client, as discussed in Section 3, so it can perfectly become agnostic to the local data distributions.

## B    ADDITIONAL EXPERIMENTAL RESULTS

In this section, we present additional experimental results to demonstrate the efficacy of the proposed `APFL` algorithm. First, we describe different datasets we have used in this paper, and then, present additional results.

### B.1    DATASETS

For the experiments we use 4 different data sources as follows:

**MNIST and CIFAR10**    For the MNIST and CIFAR10 datasets to be similar to the setting in federated learning, we need to manually distribute them in a non-IID way, hence the data distribution is pathologically heterogeneous. To this end, we follow the steps used by McMahan et al. (2017), where they partitioned the dataset based on labels and for each client draw samples from some limited number of classes. We use the same way to create 3 datasets for the MNIST, that are, MNIST non-IID with 2 classes per client, MNIST non-IID with 4 classes per client, and MNIST IID, where the data is distributed uniformly random across different clients. Also, we create a non-IID CIFAR10 dataset, where each client has access to only 2 classes of data.

**EMNIST**    In addition to pathological heterogeneous data distributions, we applied our algorithm on a real-world heterogeneous dataset, which is an extension to MNIST dataset. The EMNIST dataset includes images of characters divided by authors, where each author has a different style, make their distributions different Caldas et al. (2018). We use only digit characters and 1000 authors' data to train our models on.

**Synthetic**    For generating the synthetic dataset, we follow the procedure used by Li et al. (2018), where they use two parameters, say $\mathtt{synthetic}(\gamma, \beta)$, that control how much the local model and the local dataset of each client differ from that of other clients, respectively. Using these parameters, we want to control the diversity between data and model of different clients. The procedure is that for each client we generate a weight matrix $\boldsymbol{W}_i \in \mathbb{R}^{m \times c}$ and a bias $\boldsymbol{b} \in \mathbb{R}^c$, where the output for the $i$th client is $y_i = \arg\max\left(\sigma\left(\boldsymbol{W}_i^\top \boldsymbol{x}_i + b\right)\right)$, where $\sigma(.)$ is the softmax. In this setting, the input data $\boldsymbol{x}_i \in \mathbb{R}^m$ has $m$ features and the output $y$ can have $c$ different values indicating number of classes. The model is generated based on a Gaussian distribution $\boldsymbol{W}_i \sim \mathcal{N}(\boldsymbol{\mu}_i, 1)$ and $\boldsymbol{b}_i \sim \mathcal{N}(\boldsymbol{\mu}_i, 1)$, where $\boldsymbol{\mu}_i \sim \mathcal{N}(0, \gamma)$. The input is drown from a Gaussian distribution $\boldsymbol{x}_i \sim \mathcal{N}(\boldsymbol{\nu}_i, \boldsymbol{\Sigma})$, where $\boldsymbol{\nu}_i \sim \mathcal{N}(V_i, 1)$ and $V_i \sim \mathcal{N}(0, \beta)$. Also the variance $\boldsymbol{\Sigma}$ is a diagonal matrix with value of $\boldsymbol{\Sigma}_{k,k} = k^{-1.2}$. Using this procedure, we generate three different datasets, namely $\mathtt{synthetic}(0.0, 0.0)$, $\mathtt{synthetic}(0.5, 0.5)$, and $\mathtt{synthetic}(1.0, 1.0)$, where we move from an IID dataset to a highly non-IID data.

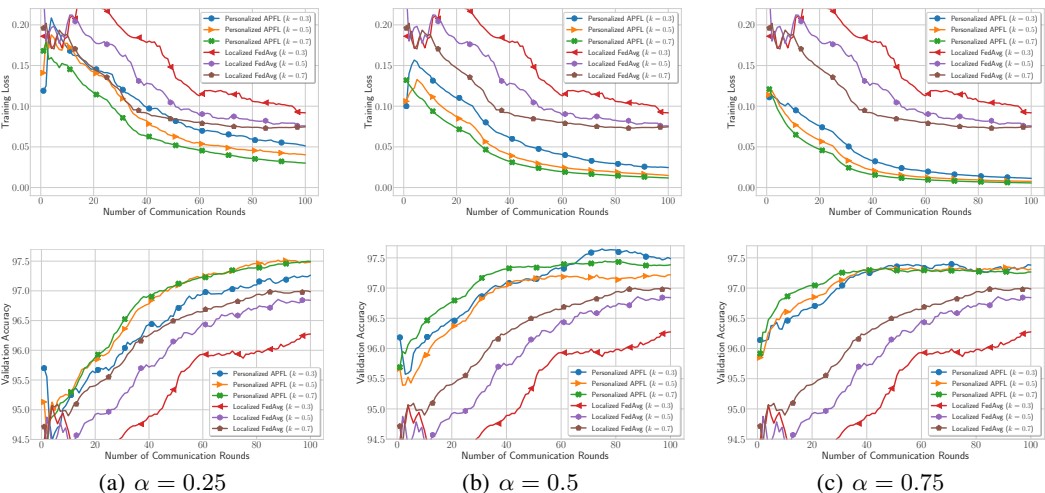

Figure 4: Evaluating the effect of sampling on `APFL` and FedAvg algorithm using the MNIST dataset that is non-IID with only 2 classes per client with logistic regression as the loss. The first row is training performance on the local model of FedAvg and personalized model of `APFL` with different sampling rates from $\{0.3, 0.5, 0.7\}$. The second row is the generalization performance of models on local validation data, aggregated over all clients. It can be inferred that despite the sampling ratio, `APFL` can superbly outperform FedAvg.

## B.2  ADDITIONAL RESULTS

In this part, we present more experimental results that can further illustrate the effectiveness of `APFL` on other datasets and models.

**Effect of sampling.**  To understand how the sampling of different clients will affect the performance of the `APFL` algorithm, we run the same experiment with different sampling rates for the MNIST dataset. The results of this experiment are depicted in Figure 4, where we run the experiment for different sampling rates of $K \in \{0.3, 0.5, 0.7\}$. Also, we run it with different values of $\alpha \in \{0.25, 0.5, 0.75\}$. The results are reported for the personalized model of `APFL` and localized FedAvg. As it can be inferred, decreasing the sampling ratio has a negative impact on both the training and generalization performance of FedAvg. However, we can see that despite the sampling ratio, `APFL` is outperforming local model of the FedAvg in both training and generalization. Also, from the results of Figure 2, we know that for this dataset that is highly non-IID, larger $\alpha$ values are preferred. Increasing $\alpha$ can diminish the negative impacts of sampling on personalized models both in training and generalization.

**Natural heterogeneous data**  In addition to the CIFAR10 and MNIST datasets with pathological heterogeneous data distributions, we apply our algorithm on a natural heterogeneous dataset, EM-NIST (Caldas et al., 2018). We use the data from 1000 clients, and for each round of communication we randomly select 10% of clients to participate in the training. We use an MLP model with 2 hidden layers, each with 200 neurons and ReLU as the activation function, using cross entropy as the loss function. For `APFL`, we use the adaptive $\alpha$ scheme with initial value of 0.5 for each client. We run both algorithms for 250 rounds of communication. In each round, each online client performs the local updates for 1 epoch on its data. Figure 5 shows the results of this experiment for personalized model of `APFL` and the localized model of the FedAvg. `APFL` with adaptive $\alpha$ can reach to the same training loss of the local FedAvg, while greatly outperforms the local FedAvg model in generalization on local validation data.

**Data distribution using Dirichlet distribution**  Another approach to distribute data in a non-IID way is to use the Dirichlet distribution as discussed in Hsu et al. (2019); Yurochkin et al. (2019). We use this approach and set the parameter of the Dirichlet distribution to 1.0 and repeat the experiments

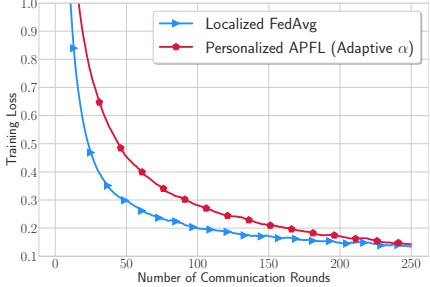 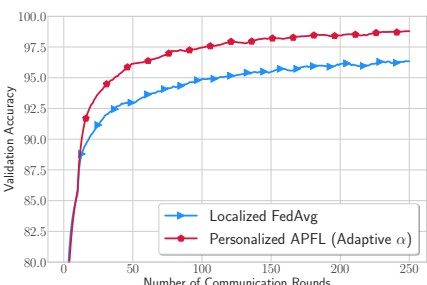

Figure 5: The results of applying FedAvg and APFL (with adaptive $\alpha$) on an MLP model using EM-NIST dataset, which is naturally heterogeneous. APFL achieves the same training loss of localized FedAVG, while outperforms it in validation accuracy.

for MNIST dataset with MLP model in the main body. Here, again we have 100 clients and run the experiments for 100 rounds of communication each with 1 epoch of training. The results are summarized in Table 3. Again, it can be inferred that APFL can generalize well on the local test dataset of different clients.

| | APFL | | | FedAVG | |
|---|---|---|---|---|---|
| | $\alpha = 0.25$ | $\alpha = 0.5$ | $\alpha = 0.75$ | Global Model | Localized Model |
| Training Loss | 0.004± 0.0009 | 0.001± 0.0003 | 0.001 | 0.1946± 0.003 | 0.022± 0.0023 |
| Validation Accuracy | 98.43%± 0.07% | 98.52%± 0.09% | 98.34%± 0.06% | 93.71%± 0.07% | 98.04%± 0.19% |

Table 3: The results of training an MLP on MNIST dataset with APFL and FedAvg using Dirichlet distribution for splitting data across clients. The parameter of the Dirichlet distribution is set to 1.

## C DISCUSSIONS AND EXTENSIONS

**Connection between learning guarantee and convergence.** As Theorem 1 suggests, the generalization bound depends on the divergence of the local and global distributions. In the language of optimization, the counter-part of divergence of distribution is the gradient diversity; hence, the gradient diversity appears in our empirical loss convergence rate (Theorem 2). The other interesting discovery is in the generalization bound, we have the term $\lambda_{\mathcal{H}}$ and $\mathcal{L}_{\mathcal{D}_i}(h_i^*)$, which are intrinsic to the distributions and hypothesis class. Meanwhile, in the convergence result, we have the term $\|v_i^* - w^*\|^2$, which also only depends on the data distribution and hypothesis class we choose. In addition, $\|v_i^* - w^*\|^2$ also reveals the divergence between local and global optimal solutions.

**Why APFL is "Adaptive".** Both information-theoretically (Theorem 1) and computationally (Theorem 2), we prove that when the local distribution drifts far away from the average distribution, the global model does not contribute too much to improve the local generalization and we have to tune the mixing parameter $\alpha$ to a larger value. Thus it is necessary to make $\alpha$ updated adaptively during empirical risk minimization. In Section 3, (6) shows that the update of $\alpha$ depends on the correlation of local gradient and deviation between local and global models. Experimental results show that our method can adaptively tune $\alpha$, and can outperform the training scheme using fixed $\alpha$.

**Comparison with local ERM model** A crucial question about personalization is *when it is preferable to employ a mixed model?*, and *how bad a local ERM model will be?* In the following corollary, we answer this by showing that the risk of local ERM model can be strictly worse than that of our personalized model.

**Corollary 1.** *Continuing with Theorem 1, there exist a distribution $\mathcal{D}_i$, constant $C_1$ and $C_2$, such that with probability at least $1 - \delta$, the following upper bound for the difference between risks of*

*personalized model $h_{\alpha_i}$ and local ERM model $\hat{h}_i^*$ on $\mathcal{D}_i$, holds :*

$$\mathcal{L}_{\mathcal{D}_i}(h_{\alpha_i}) - \mathcal{L}_{\mathcal{D}_i}(\hat{h}_i^*) \leq (2\alpha_i^2 - 1)\mathcal{L}_{\mathcal{D}_i}(h_i^*) + (2\alpha_i^2 C_1 - C_2)\sqrt{\frac{d + \log(1/\delta)}{m_i}} + 2\alpha_i^2 G\lambda_{\mathcal{H}}(\mathcal{S}_i)$$

$$+ 2(1 - \alpha_i)^2 \left( \hat{\mathcal{L}}_{\bar{\mathcal{D}}}(\bar{h}^*) + B\|\bar{\mathcal{D}} - \mathcal{D}_i\|_1 + C_1\sqrt{\frac{d + \log(1/\delta)}{m}} \right).$$

By examining the above bound, the personalized model is preferable to local model if this value is less than 0. In this case, we require $(2\alpha^2 - 1)$ and $(2\alpha_i^2 C_1 - C_2)$ to be negative, which is satisfied by choosing $\alpha_i \leq \min\{\frac{\sqrt{2}}{2}, \sqrt{\frac{C_2}{2C_1}}\}$. Then, the term $\sqrt{\frac{d + \log(1/\delta)}{m_i}}$, should be sufficiently large, and the divergence term, as well as the global model generalization error has to be small. In this case, from the local model perspective, it can benefit from incorporate some global model. Using the similar technique, we can prove the supremacy of mixed model over global model as well.

*Proof of Corollary 1.* Since in Theorem 1, we already obtained upper bound for $\mathcal{L}_{\mathcal{D}_i}(h_{\alpha_i})$ as following,

$$\mathcal{L}_{\mathcal{D}_i}(h_{\alpha_i}) \leq 2\alpha_i^2 \left( \mathcal{L}_{\mathcal{D}_i}(h_i^*) + 2C_1\sqrt{\frac{d + \log(1/\delta)}{m_i}} + G\lambda_{\mathcal{H}}(\mathcal{S}_i) \right)$$

$$+ 2(1 - \alpha_i)^2 \left( \hat{\mathcal{L}}_{\bar{\mathcal{D}}}(\bar{h}^*) + B\|\bar{\mathcal{D}} - \mathcal{D}_i\|_1 + C_1\sqrt{\frac{d + \log(1/\delta)}{m}} \right),$$

to find the upper bound of $\mathcal{L}_{\mathcal{D}_i}(h_{\alpha_i}) - \mathcal{L}_{\mathcal{D}_i}(\hat{h}_i^*)$, we just need the lower bound of $\mathcal{L}_{\mathcal{D}_i}(\hat{h}_i^*)$. The fundamental theorem of statistical learning (Shalev-Shwartz & Ben-David, 2014; Mohri et al., 2018) states a lower risk bound for agnostic PAC learning: for a hypothesis class with finite VC dimension $d$, then there exists a distribution $\mathcal{D}$, such that for any learning algorithm, which learns a hypothesis $h \in \mathcal{H}$ on $m$ i.i.d. samples from $\mathcal{D}$, there exists a constant $C$, with the probability at least $1 - \delta$, we have:

$$\mathcal{L}_{\mathcal{D}}(h) - \min_{h' \in \mathcal{H}} \mathcal{L}_{\mathcal{D}}(h') \geq C\sqrt{\frac{d + \log(1/\delta)}{m}}.$$

Since $\hat{h}_i^*$ is learnt by ERM algorithm, the agnostic PAC learning lower risk bound also holds for it, so in worst case it might hold that under distribution $\mathcal{D}_i$, if $\hat{h}_i^*$ is learnt by ERM algorithm using $m_i$ samples, then there is a $C_2$, such that with probability at least $1 - \delta$, we have:

$$\mathcal{L}_{\mathcal{D}_i}(\hat{h}_i^*) \geq \mathcal{L}_{\mathcal{D}_i}(h_i^*) + C_2\sqrt{\frac{d + \log(1/\delta)}{m_i}}.$$

Thus we can bound $\mathcal{L}_{\mathcal{D}_i}(h_{\alpha_i}) - \mathcal{L}_{\mathcal{D}_i}(\hat{h}_i^*)$ as Corollary 1 claims. $\qquad\square$

**Personalization for new participant nodes.** Suppose we already have a trained global model $\hat{w}$, and now a new device $k$ joins in the network, which is desired to personalize the global model to adapt its own domain. This can be done by performing a few local stochastic gradient descent updates from the given global model as an initial local model:

$$\boldsymbol{v}_k^{(t+1)} = \boldsymbol{v}_k^{(t)} - \eta_t \nabla_{\boldsymbol{v}} f_k(\alpha_k \boldsymbol{v}_k^{(t)} + (1 - \alpha_k)\hat{\boldsymbol{w}}; \xi_k^{(t)}) \tag{7}$$

to quickly learn a personalized model for the newly joined device. One thing worthy of investigation is the difference between APFL and meta-learning approaches, such as model-agnostic meta-learning (Finn et al., 2017). Our goal is to share the knowledge among the different users, in order to reduce the generalization error; while meta-learning cares more about how to build a meta-learner, to help training models faster and with fewer samples. In this scenario, similar to FedAvg, when a new node joins the network, it gets the global model and takes a few stochastic steps based on its own data to update the global model. In Figure 6, we show the results of applying

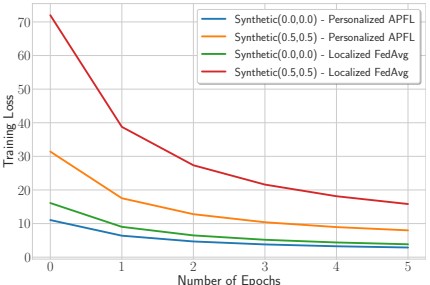 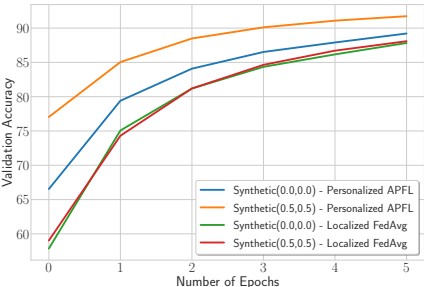

Figure 6: Comparing the effect of fine-tuning with the local model of FedAvg and with the personalized model of `APFL` on the synthetic datasets. The model is trained for $100$ rounds of communication with 97 clients, and then 3 clients will join in fine-tuning the global model based on their own data. It can be seen that the model from `APFL` can better personalize the global model with respect to the FedAvg method both in training loss and validation accuracy. Increasing diversity makes it harder to personalize, however, `APFL` surpasses FedAvg again.

FedAvg and `APFL` on synthetic data with two different rates of diversity, `synthetic`$(0.0, 0.0)$ and `synthetic`$(0.5, 0.5)$. In this experiment, we keep 3 nodes with their data off in the entire training for 100 rounds of communication between 97 nodes. In each round, each client updates its local and personalized models for one epoch. After the training is done, those 3 clients will join the network and get the latest global model and start training local and personalized models of their own. Figure 6 shows the training loss and validation accuracy of these 3 nodes during the 5 epochs of updates. The local model represents the model that will be trained in FedAvg, while the personalized model is the one resulting from `APFL`. Although the goal of `APFL` is to adaptively learn the personalized model during the training, it can be inferred that `APFL` can learn a better personalized model in a meta-learning scenario as well.

**Agnostic global model.** As pointed out by Mohri et al. (2019), the global model can be distributionally robust if we optimize the agnostic loss:

$$\min_{\boldsymbol{w}\in\mathbb{R}^d} \max_{\boldsymbol{q}\in\Delta_n} F(\boldsymbol{w}) := \sum_{i}^{n} q_i f_i(\boldsymbol{w}), \tag{8}$$

where $\Delta_n = \{\boldsymbol{q} \in \mathbb{R}^n_+ \mid \sum q_i = 1\}$ is the $n$-dimensional simplex. We call this scenario "Adaptive Personalized Agnostic Federated Learning". In this case, the analysis will be more challenging since the global empirical risk minimization is performed at a totally different domain, so the risk upper bound for $h_{\alpha_i}$ we derived does not hold anymore. Also, from a computational standpoint, since the resulted problem is a minimax optimization problem, the convergence analysis of agnostic `APFL` will be more involved, which we will leave as an interesting future work.

## D    PROOF OF GENERALIZATION BOUND

In this section we present the proof of generalization bound for APFL algorithm. Recall that we define the following hypotheses on $i$th local true and empirical distributions:

$$\hat{h}_i^* = \arg\min_{h \in \mathcal{H}} \hat{\mathcal{L}}_{\mathcal{D}_i}(h) \qquad \text{(LOCAL EMPIRICAL RISK MINIMIZER)}$$

$$h_i^* = \arg\min_{h \in \mathcal{H}} \mathcal{L}_{\mathcal{D}_i}(h) \qquad \text{(LOCAL TRUE RISK MINIMIZER)}$$

$$\bar{h}^* = \arg\min_{h \in \mathcal{H}} \mathcal{L}_{\bar{\mathcal{D}}}(h) \qquad \text{(GLOBAL EMPIRICAL RISK MINIMIZER)}$$

$$\hat{h}_{loc,i}^* = \arg\min_{h \in \mathcal{H}} \hat{\mathcal{L}}_{\mathcal{D}_i}(\alpha_i h + (1-\alpha_i)\bar{h}^*) \quad \text{(MIXED EMPIRICAL RISK MINIMIZER)}$$

$$h_{loc,i}^* = \arg\min_{h \in \mathcal{H}} \mathcal{L}_{\mathcal{D}_i}(\alpha_i h + (1-\alpha_i)\bar{h}^*) \quad \text{(MIXED TRUE RISK MINIMIZER)}$$

where $\hat{\mathcal{L}}_{\mathcal{D}_i}(h)$ and $\mathcal{L}_{\mathcal{D}_i}(h)$ denote the empirical and true risks on $\mathcal{D}_i$, respectively.

From a high-level technical view, since we wish to bound the risk of the mixed model on local distribution $\mathcal{D}_i$, first we need to utilize the convex property of the risk function, and decompose it into two parts: $\mathcal{L}_{\mathcal{D}_i}\left(\hat{h}_{loc,i}^*\right)$ and $\mathcal{L}_{\mathcal{D}_i}\left(\bar{h}^*\right)$. To bound $\mathcal{L}_{\mathcal{D}_i}\left(\hat{h}_{loc,i}^*\right)$, a natural idea is to characterize it by the risk of optimal model $\mathcal{L}_{\mathcal{D}_i}(h_i^*)$, plus some excess risk. However, due to fact that $\hat{h}_{loc,i}^*$ is not the sole local empirical risk minimizer, rather it partially incorporates the global model, we need to characterize to what extent it drifts from the local empirical risk minimizer $\hat{h}_i^*$. This drift can be depicted by the hypothesis capacity, so that is our motivation to define $\lambda_{\mathcal{H}}(\mathcal{S})$ to quantify the empirical loss discrepancy over $\mathcal{S}$ among pair of hypotheses in $\mathcal{H}$. We have to admit that there should be a tighter theory to bound this drift, depending how global model is incorporated, which we leave it as a future work.

The following simple result will be useful in the proof of generalization.

**Lemma 1.** *Let $\mathcal{H}$ be a hypothesis class and $\mathcal{D}$ and $\mathcal{D}'$ denote two probability measures over space $\Xi$. Let $\mathcal{L}_{\mathcal{D}}(h) = \mathbb{E}_{(\boldsymbol{x},y) \sim \mathcal{D}}[\ell(h(\boldsymbol{x}),y)]$ denote the risk of $h$ over $\mathcal{D}$ . If the loss function $\ell(\cdot)$ is bounded by $B$, then for every $h \in \mathcal{H}$:*

$$\mathcal{L}_{\mathcal{D}}(h) \leq \mathcal{L}_{\mathcal{D}'}(h) + B\|\mathcal{D} - \mathcal{D}'\|_1, \tag{9}$$

*where $\|\mathcal{D} - \mathcal{D}'\|_1 = \int_{\Xi} |\mathbb{P}_{(\boldsymbol{x},y) \sim \mathcal{D}} - \mathbb{P}_{(\boldsymbol{x},y) \sim \mathcal{D}'}| d\boldsymbol{x} dy.$*

*Proof.*

$$\mathcal{L}_{\mathcal{D}}(h) \leq \mathcal{L}_{\mathcal{D}'}(h) + |\mathcal{L}_{\mathcal{D}}(h) - \mathcal{L}_{\mathcal{D}'}(h)|$$

$$\leq \mathcal{L}_{\mathcal{D}}(h) + \int_{\Xi} |\ell(y, h(\boldsymbol{x}))| |\mathbb{P}_{(\boldsymbol{x},y) \sim \mathcal{D}} - \mathbb{P}_{(\boldsymbol{x},y) \sim \mathcal{D}'}| d\boldsymbol{x} dy$$

$$= \mathcal{L}_{\mathcal{D}}(h) + B\|\mathcal{D} - \mathcal{D}'\|_1.$$

$\square$

**Proof of Theorem 1**    We now turn to proving the generalization bound for the proposed APFL algorithm. Recall that for the classification task we consider squared hinge loss, and for the regression case we consider MSE loss. We will first prove that in both cases we can decompose the risk as follows:

$$\mathcal{L}_{\mathcal{D}_i}(h_{\alpha_i}^*) \leq 2\alpha_i^2 \mathcal{L}_{\mathcal{D}_i}\left(\hat{h}_{loc,i}^*\right) + 2(1-\alpha_i)^2 \mathcal{L}_{\mathcal{D}_i}\left(\bar{h}^*(\boldsymbol{x})\right). \tag{10}$$

We start with the classification case first. Note that, hinge loss: $\max\{0, 1 - z\}$ is convex in $z$, so $\max\{0, 1 - y(\alpha_i h + (1-\alpha_i)h')\} \leq \alpha_i \max\{0, 1-yh\} + (1-\alpha_i)\max\{0, 1-yh'\}$, according to

Jensen's inequality. Hence, we have:

$$
\begin{aligned}
\mathcal{L}_{\mathcal{D}_i}(h^*_{\alpha_i}) &= \mathcal{L}_{\mathcal{D}_i}(\alpha_i \hat{h}^*_{loc,i} + (1 - \alpha_i)\bar{h}^*) \\
&= \mathbb{E}_{(\boldsymbol{x},y)\sim\mathcal{D}_i}\left(\max\{0, 1 - y(\alpha_i \hat{h}^*_{loc,i}(\boldsymbol{x}) + (1-\alpha_i)\bar{h}^*(\boldsymbol{x}))\}\right)^2 \\
&= \mathbb{E}_{(\boldsymbol{x},y)\sim\mathcal{D}_i}\left(\alpha_i \max\{0, 1 - y\hat{h}^*_{loc,i}(\boldsymbol{x})\} + (1-\alpha_i)\max\{0, 1 - y\bar{h}^*(\boldsymbol{x})\}\right)^2 \\
&\leq 2\alpha_i^2 \mathbb{E}_{(\boldsymbol{x},y)\sim\mathcal{D}_i}\left(\max\{0, 1 - y\hat{h}^*_{loc,i}(\boldsymbol{x})\}\right)^2 \\
&\quad + 2(1-\alpha_i)^2 \mathbb{E}_{(\boldsymbol{x},y)\sim\mathcal{D}_i}\left(\max\{0, 1 - y\bar{h}^*(\boldsymbol{x})\}\right)^2 \\
&\leq 2\alpha_i^2 \mathcal{L}_{\mathcal{D}_i}\left(\hat{h}^*_{loc,i}\right) + 2(1-\alpha_i)^2 \mathcal{L}_{\mathcal{D}_i}\left(\bar{h}^*\right).
\end{aligned}
$$

For regression case:

$$
\begin{aligned}
\mathcal{L}_{\mathcal{D}_i}(h^*_{\alpha_i}) &= \mathcal{L}_{\mathcal{D}_i}(\alpha_i \hat{h}^*_{loc,i} + (1 - \alpha_i)\bar{h}^*) \\
&= \mathbb{E}_{(\boldsymbol{x},y)\sim\mathcal{D}_i}\left\|y - (\alpha_i \hat{h}^*_{loc,i}(\boldsymbol{x}) + (1-\alpha_i)\bar{h}^*(\boldsymbol{x}))\right\|^2 \\
&= \mathbb{E}_{(\boldsymbol{x},y)\sim\mathcal{D}_i}\left\|\alpha_i y - \alpha_i \hat{h}^*_{loc,i}(\boldsymbol{x}) + (1-\alpha_i)y - (1-\alpha_i)\bar{h}^*(\boldsymbol{x})\right\|^2 \\
&\leq 2\alpha_i^2 \mathbb{E}_{(\boldsymbol{x},y)\sim\mathcal{D}_i}\left\|y - \hat{h}^*_{loc,i}(\boldsymbol{x})\right\|^2 + 2(1-\alpha_i)^2 \mathbb{E}_{(\boldsymbol{x},y)\sim\mathcal{D}_i}\left\|y - \bar{h}^*(\boldsymbol{x})\right\|^2 \\
&\leq 2\alpha_i^2 \mathcal{L}_{\mathcal{D}_i}\left(\hat{h}^*_{loc,i}\right) + 2(1-\alpha_i)^2 \mathcal{L}_{\mathcal{D}_i}\left(\bar{h}^*\right)
\end{aligned}
$$

Thus we can conclude:

$$
\mathcal{L}_{\mathcal{D}_i}(h^*_{\alpha_i}) \leq 2\alpha_i^2 \underbrace{\mathcal{L}_{\mathcal{D}_i}\left(\hat{h}^*_{loc,i}\right)}_{T_1} + 2(1-\alpha_i)^2 \underbrace{\mathcal{L}_{\mathcal{D}_i}\left(\bar{h}^*\right)}_{T_2}. \tag{11}
$$

We proceed to bound the terms $T_1$ and $T_2$ in RHS of above inequality. We first bound $T_1$ as follows. The first step is to utilize uniform VC dimension error bound over $\mathcal{H}$ Mohri et al. (2018); Shalev-Shwartz & Ben-David (2014):

$$
\forall h \in \mathcal{H}, |\mathcal{L}_{\mathcal{D}_i}(h) - \hat{\mathcal{L}}_{\mathcal{D}_i}(h)| \leq C\sqrt{\frac{d + \log(1/\delta)}{m_i}},
$$

where $C$ is constant factor. So we can bound $T_1$ as:

$$
\begin{aligned}
T_1 = \mathcal{L}_{\mathcal{D}_i}(\hat{h}^*_{loc,i}) &= \mathcal{L}_{\mathcal{D}_i}(h^*_i) + \mathcal{L}_{\mathcal{D}_i}(\hat{h}^*_{loc,i}) - \mathcal{L}_{\mathcal{D}_i}(h^*_i) \\
&= \mathcal{L}_{\mathcal{D}_i}(h^*_i) \\
&\quad + \underbrace{\mathcal{L}_{\mathcal{D}_i}(\hat{h}^*_{loc,i}) - \hat{\mathcal{L}}_{\mathcal{D}_i}(\hat{h}^*_{loc,i})}_{\leq C\sqrt{\frac{d+\log(1/\delta)}{m_i}}} + \hat{\mathcal{L}}_{\mathcal{D}_i}(\hat{h}^*_{loc,i}) - \hat{\mathcal{L}}_{\mathcal{D}_i}(h^*_i) + \underbrace{\hat{\mathcal{L}}_{\mathcal{D}_i}(h^*_i) - \mathcal{L}_{\mathcal{D}_i}(h^*_i)}_{\leq C\sqrt{\frac{d+\log(1/\delta)}{m_i}}} \\
&\leq \mathcal{L}_{\mathcal{D}_i}(h^*_i) + 2C\sqrt{\frac{d + \log(1/\delta)}{m_i}} + \hat{\mathcal{L}}_{\mathcal{D}_i}(\hat{h}^*_{loc,i}) - \hat{\mathcal{L}}_{\mathcal{D}_i}(h^*_i).
\end{aligned}
$$

Note that

$$
\hat{\mathcal{L}}_{\mathcal{D}_i}(\hat{h}^*_{loc,i}) - \hat{\mathcal{L}}_{\mathcal{D}_i}(\hat{h}^*_i) \leq G\frac{1}{|\mathcal{S}_i|}\sum_{(\boldsymbol{x},y)\in\mathcal{S}_i}|\hat{h}^*_{loc,i}(\boldsymbol{x}) - \hat{h}^*_i(\boldsymbol{x})| \leq G\lambda_{\mathcal{H}}(\mathcal{S}_i),
$$

As a result we can bound $T_1$ by:

$$
T_1 \leq \mathcal{L}_{\mathcal{D}_i}(h^*_i) + 2C\sqrt{\frac{d + \log(1/\delta)}{m_i}} + G\lambda_{\mathcal{H}}(\mathcal{S}_i).
$$

We now turn to bounding $T_2$. Plugging Lemma 1 in (11) and using uniform generalization risk bound will immediately give:

$$T_2 \leq \hat{\mathcal{L}}_{\bar{\mathcal{D}}}(\bar{h}^*) + B^2\|\mathcal{D} - \bar{\mathcal{D}}\|_1 + C\sqrt{\frac{d + \log(1/\delta)}{m}}.$$

Plugging $T_1$ and $T_2$ back into (11) concludes the proof. □

**Remark 3.** *One thing worth mentioning is that, we assume the customary boundedness of loss functions. Actually it can be satisfied if the data and the parameters of hypothesis are bounded. For example, considering the scenario where we are learning a linear model $w$ with the constraint $\|w\| \leq 1$, and also the data tuples $(x, y)$ are drawn from some bounded domain, then the loss is obviously bounded by some finite real value.*

**Remark 4.** *As $\mathcal{L}_{\mathcal{D}_i}\left(\hat{h}^*_{loc,i}\right)$ is the risk of the empirical risk minimizer on $\mathcal{D}_i$ after incorporating a model learned on a different domain (i.e., global distribution), one might argue that generalization techniques established in multi-domain learning theory (Ben-David et al., 2010; Mansour et al., 2009; Zhang et al., 2020) can be utilized to serve our purpose. However, we note that the techniques developed in Ben-David et al. (2010); Mansour et al. (2009); Zhang et al. (2020) are only applicable to a settings where we aim at directly learning a model in some combination of source and target domain, while in our setting, we partially incorporate the model learned from source domain and then perform ERM on joint model over target domain. Moreover, their results only apply to very simple loss functions, e.g., absolute loss or MSE loss, while we consider squared hinge loss in the classification case. Analogous to multiple domain theory, we derive the multi domain learning bound based on the divergence of source and target domains but measured in absolute distance, $\|\cdot\|_1$. As Mansour et al. (2009) points out, divergence measured by absolute loss can be large, and as a result we leave the development of a more general multiple domain learning theory that can deal with most popular loss functions like hinge loss, cross entropy loss and optimal transport, with tighter divergence measure on distributions as an open question.*

## E    PROOF OF CONVERGENCE RATE IN CONVEX SETTING

In this section, we present the proof of convergence raters. For ease of mathematical derivations, we first consider the case *without sampling clients* at each communication step and then generalize the proof to the setting where $K$ *devices are sampled uniformly at random* by the server as employed in the proposed algorithm.

**Technical challenges.**    The analysis of convergence rates in our setting is more involved compared to analysis of local SGD with periodic averaging by Stich (2018); Woodworth et al. (2020a). The key difficulty arises from the fact that unlike local SGD where local solutions are evolved by employing mini-batch SGD, in our setting we also partially incorporate the global model to compute stochastic gradients over local data. In addition, our goal is to find the convergence rate of the mixed model, rather than merely the local model or global model. To better illustrate this, let us first clarify the notations of models that will be used in analysis. Let us consider the simple case for now where we set $K = n$ (all device participate averaging). We define three virtual sequences: $\{w^{(t)}\}_{t=1}^T$, $\{\bar{v}^{(t)}\}_{t=1}^T$ and $\{\hat{v}^{(t)}\}_{t=1}^T$ where $w^{(t)} = \frac{1}{n}\sum_{j=1}^n w_i^{(t)}, \bar{v}_i^{(t)} = \alpha_i v_i^{(t)} + (1-\alpha_i)w_i^{(t)}$ $\hat{v}_i^{(t)} = \alpha_i v_i^{(t)} + (1-\alpha_i)w^{(t)}$. Since the personalized model incorporates $1-\alpha_i$ percentage of global model, then the key challenge in the convergence analysis is to find out how much the global model benefits/hurts the local convergence. To this end, we analyze how much the dynamics of personalized model $\hat{v}_i^{(t)}$ and global model $w^{(t)}$ differ from each other at each iteration. To be more specific, we study the distance between gradients $\|\nabla f_i(\hat{v}_i^{(t)}) - \nabla F(w^{(t)})\|^2$. Surprisingly, we relate this distance to gradient diversity, personalized model convergence, global model convergence and local-global optimality gap:

$$\mathbb{E}\left[\|\nabla f_i(\hat{v}_i^{(t)}) - \nabla F(w^{(t)})\|^2\right] \leq 6\zeta_i + 2L^2\mathbb{E}\left[\|\hat{v}_i^{(t)} - v^*\|^2\right] + 6L^2\mathbb{E}\left[\|w^{(t)} - w^*\|^2\right] + 6L^2\Delta_i.$$

$\mathbb{E}\left[\|\hat{v}_i^{(t)} - v^*\|^2\right]$ and $\mathbb{E}\left[\|w^{(t)} - w^*\|^2\right]$ will converge very fast under smooth strongly convex objective, and $\zeta_i$ and $\Delta_i$ will serve as residual error that indicates the heterogeneity among local functions.

---

**Algorithm 2:** `Local Descent APFL (without sampling)`

---

**input:** Mixture weights $\alpha_1, \cdots, \alpha_n$, Synchronization gap $\tau$, Local models $\boldsymbol{v}_i^{(0)}$ for $i \in [n]$ and local version of global model $\boldsymbol{w}_i^{(0)}$ for $i \in [n]$.

**for** $t = 0, \cdots, T$ **do**

    **if** $t$ *not divides* $\tau$ **then**

$$\boldsymbol{w}_i^{(t)} = \prod_{\mathcal{W}} \left( \boldsymbol{w}_i^{(t-1)} - \eta_t \nabla f_i \left( \boldsymbol{w}_i^{(t-1)}; \xi_i^t \right) \right)$$

$$\boldsymbol{v}_i^{(t)} = \prod_{\mathcal{W}} \left( \boldsymbol{v}_i^{(t-1)} - \eta_t \nabla_{\boldsymbol{v}} f_i \left( \bar{\boldsymbol{v}}_i^{(t-1)}; \xi_i^t \right) \right)$$

$$\bar{\boldsymbol{v}}_i^{(t)} = \alpha_i \boldsymbol{v}_i^{(t)} + (1 - \alpha_i) \boldsymbol{w}_i^{(t)}$$

    **else**

        each client sends $\boldsymbol{w}_j^{(t)}$ to the server

        $\boldsymbol{w}^{(t)} = \frac{1}{n} \sum_{j=1}^n \boldsymbol{w}_j^{(t)}$

        server broadcast $\boldsymbol{w}^{(t)}$ to all clients

    **end**

**end**

**for** $i = 1, \cdots, n$ **do**

    **output:** Personalized model: $\hat{\boldsymbol{v}}_i = \frac{1}{S_T} \sum_{t=1}^T p_t (\alpha_i \boldsymbol{v}_i^{(t)} + (1 - \alpha_i) \frac{1}{n} \sum_{j=1}^n \boldsymbol{w}_j^{(t)})$;

             Global model: $\hat{\boldsymbol{w}} = \frac{1}{n S_T} \sum_{t=1}^T p_t \sum_{j=1}^n \boldsymbol{w}_j^{(t)}$.

**end**

---

### E.1    PROOF WITHOUT SAMPLING

Before giving the proof of convergence analysis of the Algorithm 1 in the main paper, we first discuss a warm-up case: local descent APFL without client sampling. As Algorithm 2 shows, all clients will participate in the averaging stage every $\tau$ iterations. The convergence of global and local models in Algorithm 2 are given in the following theorems. We start by stating the convergence of global model.

**Theorem 4** (Global model convergence of `Local Descent APFL without Sampling`)**.** *If each client's objective function is $\mu$-strongly convex and $L$-smooth, and satisfies Assumption 1, using Algorithm 2, choosing the mixing weight $\alpha_i \geq \max\{1 - \frac{1}{4\sqrt{6}\kappa}, 1 - \frac{1}{4\sqrt{6}\kappa\sqrt{\mu}}\}$, learning rate $\eta_t = \frac{16}{\mu(t+a)}$, where $a = \max\{128\kappa, \tau\}$, and using average scheme $\hat{\boldsymbol{w}} = \frac{1}{n S_T} \sum_{t=1}^T p_t \sum_{j=1}^n \boldsymbol{w}_j^{(t)}$, where $p_t = (t+a)^2$, $S_T = \sum_{t=1}^T p_t$, then the following convergence holds:*

$$\mathbb{E}\left[F(\hat{\boldsymbol{w}})\right] - F(\boldsymbol{w}^*) \leq O\left(\frac{\mu}{T^3}\right) + O\left(\frac{\kappa^2 \tau \left(\sigma^2 + \tau \frac{\varsigma}{n}\right)}{\mu T^2}\right) + O\left(\frac{\kappa^2 \tau \left(\sigma^2 + \tau \frac{\varsigma}{n}\right) \ln T}{\mu T^3}\right) + O\left(\frac{\sigma^2}{nT}\right),$$

*where $\boldsymbol{w}_* = \arg\min_{\boldsymbol{w}} F(\boldsymbol{w})$ is the optimal global solution.*

*Proof.* Proof is deferred to Appendix E.1.2. $\qquad\qquad\square$

The following theorem obtains the convergence of personalized model in Algorithm 2.

**Theorem 5** (Personalized model convergence of `Local Descent APFL without Sampling`)**.** *If each client's objective function is $\mu$-strongly convex and $L$-smooth, and satisfies Assumption 1, using Algorithm 2, choosing the mixing weight $\alpha_i \geq \max\{1 - \frac{1}{4\sqrt{6}\kappa}, 1 - \frac{1}{4\sqrt{6}\kappa\sqrt{\mu}}\}$, learning rate $\eta_t = \frac{16}{\mu(t+a)}$, where $a = \max\{128\kappa, \tau\}$, and using average scheme $\hat{\boldsymbol{v}}_i = \frac{1}{S_T} \sum_{t=1}^T p_t (\alpha_i \boldsymbol{v}_i^{(t)} + (1 - \alpha_i) \frac{1}{n} \sum_{j=1}^n \boldsymbol{w}_j^{(t)})$, where $p_t = (t+a)^2$, $S_T = \sum_{t=1}^T p_t$, and $f_i^*$ is the local minimum of the ith client, then the following convergence holds for all $i \in [n]$:*

$$\mathbb{E}[f_i(\hat{\boldsymbol{v}}_i)] - f_i^* \leq O\left(\frac{\mu}{T^3}\right) + \alpha_i^2 O\left(\frac{\sigma^2}{\mu T}\right) + (1 - \alpha_i)^2 O\left(\frac{\zeta_i}{\mu} + \kappa L \Delta_i\right)$$

$$+ (1 - \alpha_i)^2 \left( O\left(\frac{\kappa L \ln T}{T^3}\right) + O\left(\frac{\kappa^2 \sigma^2}{\mu n T}\right) + O\left(\frac{\kappa^2 \tau \left(\sigma^2 + \tau(\zeta_i + \frac{\varsigma}{n})\right)}{\mu T^2}\right) + O\left(\frac{\kappa^4 \tau \left(\sigma^2 + 2\tau \frac{\varsigma}{n}\right)}{\mu T^2}\right) \right).$$

*Proof.* Proof is deferred to Appendix E.1.3. □

### E.1.1 PROOF OF USEFUL LEMMAS

Before giving the proof of Theorem 4 and 5, we first prove few useful lemmas. Recall that we define virtual sequences $\{\boldsymbol{w}^{(t)}\}_{t=1}^T, \{\bar{\boldsymbol{v}}_i^{(t)}\}_{t=1}^T, \{\hat{\boldsymbol{v}}_i^{(t)}\}_{t=1}^T$ where $\boldsymbol{w}^{(t)} = \frac{1}{n}\sum_{i=1}^n \boldsymbol{w}_i^{(t)}, \bar{\boldsymbol{v}}_i^{(t)} = \alpha_i \boldsymbol{v}_i^{(t)} + (1-\alpha_i)\boldsymbol{w}_i^{(t)}, \hat{\boldsymbol{v}}_i^{(t)} = \alpha_i \boldsymbol{v}_i^{(t)} + (1-\alpha_i)\boldsymbol{w}^{(t)}$.

We start with the following lemma that bounds the difference between the gradients of local objective and global objective at local and global models.

**Lemma 2.** *For Algorithm 2, at each iteration, the gap between local gradient and global gradient is bounded by*

$$\mathbb{E}\left[\|\nabla f_i(\hat{\boldsymbol{v}}_i^{(t)}) - \nabla F(\boldsymbol{w}^{(t)})\|^2\right] \leq 2L^2\mathbb{E}\left[\|\hat{\boldsymbol{v}}_i^{(t)} - \boldsymbol{v}^*\|^2\right] + 6\zeta_i + 6L^2\mathbb{E}\left[\|\boldsymbol{w}^{(t)} - \boldsymbol{w}^*\|^2\right] + 6L^2\Delta_i.$$

*Proof.* From the smoothness assumption and by applying the Jensen's inequality we have:

$$\mathbb{E}\left[\|\nabla f_i(\hat{\boldsymbol{v}}_i^{(t)}) - \nabla F(\boldsymbol{w}^{(t)})\|^2\right]$$

$$\leq 2\mathbb{E}\left[\|\nabla f_i(\hat{\boldsymbol{v}}_i^{(t)}) - \nabla f_i(\boldsymbol{v}_i^*)\|^2\right] + 2\mathbb{E}\left[\|\nabla f_i(\boldsymbol{v}_i^*) - \nabla F(\boldsymbol{w}^{(t)})\|^2\right]$$

$$\leq 2L^2\mathbb{E}\left[\|\hat{\boldsymbol{v}}_i^{(t)} - \boldsymbol{v}^*\|^2\right] + 6\mathbb{E}\left[\|\nabla f_i(\boldsymbol{v}_i^*) - \nabla f_i(\boldsymbol{w}^*)\|^2\right]$$

$$\quad + 6\mathbb{E}\left[\|\nabla f_i(\boldsymbol{w}^*) - \nabla F(\boldsymbol{w}^*)\|^2\right] + 6\mathbb{E}\left[\|\nabla F(\boldsymbol{w}^*) - \nabla F(\boldsymbol{w}^{(t)})\|^2\right]$$

$$\leq 2L^2\mathbb{E}\left[\|\hat{\boldsymbol{v}}_i^{(t)} - \boldsymbol{v}^*\|^2\right] + 6L^2\mathbb{E}\left[\|\boldsymbol{v}_i^* - \boldsymbol{w}^*\|^2\right] + 6\zeta_i + 6L^2\mathbb{E}\left[\|\boldsymbol{w}^{(t)} - \boldsymbol{w}^*\|^2\right]$$

$$\leq 2L^2\mathbb{E}\left[\|\hat{\boldsymbol{v}}_i^{(t)} - \boldsymbol{v}^*\|^2\right] + 6L^2\Delta_i + 6\zeta_i + 6L^2\mathbb{E}\left[\|\boldsymbol{w}^{(t)} - \boldsymbol{w}^*\|^2\right].$$

□

**Lemma 3** (Local model deviation without sampling). *For Algorithm 2, at each iteration, the deviation between each local version of the global model $\boldsymbol{w}_i^{(t)}$ and the global model $\boldsymbol{w}^{(t)}$ is bounded by:*

$$\mathbb{E}\left[\|\boldsymbol{w}^{(t)} - \boldsymbol{w}_i^{(t)}\|^2\right] \leq 3\tau\sigma^2\eta_{t-1}^2 + 3(\zeta_i + \frac{\zeta}{n})\tau^2\eta_{t-1}^2,$$

$$\frac{1}{n}\sum_{i=1}^n \mathbb{E}\left[\|\boldsymbol{w}^{(t)} - \boldsymbol{w}_i^{(t)}\|^2\right] \leq 3\tau\sigma^2\eta_{t-1}^2 + 6\tau^2\frac{\zeta}{n}\eta_{t-1}^2,$$

*where $\frac{\zeta}{n} = \frac{1}{n}\sum_{i=1}^n \zeta_i$.*

*Proof.* According to Lemma 8 in Woodworth et al. (2020a):

$$\mathbb{E}\left[\|\boldsymbol{w}^{(t)} - \boldsymbol{w}_i^{(t)}\|^2\right] \leq \frac{1}{n}\sum_{j=1}^n \mathbb{E}\left[\|\boldsymbol{w}_j^{(t)} - \boldsymbol{w}_i^{(t)}\|^2\right]$$

$$\leq 3\left(\sigma^2 + \zeta_i\tau + \frac{\zeta}{n}\tau\right)\sum_{p=t_c}^{t-1}\eta_p^2 \prod_{q=p+1}^{t-1}(1-\mu\eta_q)$$

$$\frac{1}{n}\sum_{i=1}^n \mathbb{E}\left[\|\boldsymbol{w}^{(t)} - \boldsymbol{w}_i^{(t)}\|^2\right] \leq \frac{1}{n^2}\sum_{i=1}^n\sum_{j=1}^n \mathbb{E}\left[\|\boldsymbol{w}_j^{(t)} - \boldsymbol{w}_i^{(t)}\|^2\right]$$

$$\leq 3\left(\sigma^2 + 2\tau\frac{\zeta}{n}\right)\sum_{p=t_c}^{t-1}\eta_p^2 \prod_{q=p+1}^{t-1}(1-\mu\eta_q).$$

Plugging in $\eta_q = \frac{16}{\mu(a+q)}$ yields:

$$\mathbb{E}\left[\|\boldsymbol{w}^{(t)} - \boldsymbol{w}_i^{(t)}\|^2\right] \leq 3\left(\sigma^2 + \zeta_i\tau + \frac{\zeta}{n}\tau\right) \sum_{p=t_c}^{t-1} \eta_p^2 \prod_{q=p+1}^{t-1} \frac{a+q-16}{a+q}$$

$$\leq 3\left(\sigma^2 + \zeta_i\tau + \frac{\zeta}{n}\tau\right) \sum_{p=t_c}^{t-1} \eta_p^2 \prod_{q=p+1}^{t-1} \frac{a+q-16}{a+q}$$

$$\leq 3\left(\sigma^2 + \zeta_i\tau + \frac{\zeta}{n}\tau\right) \sum_{p=t_c}^{t-1} \eta_p^2 \prod_{q=p+1}^{t-1} \frac{a+q-2}{a+q}$$

$$\leq 3\left(\sigma^2 + \zeta_i\tau + \frac{\zeta}{n}\tau\right) \sum_{p=t_c}^{t-1} \eta_p^2 \frac{(a+p-1)(a+p)}{(a+t-2)(a+t-1)}$$

$$\leq 3\left(\sigma^2 + \zeta_i\tau + \frac{\zeta}{n}\tau\right) \sum_{p=t_c}^{t-1} \eta_p^2 \frac{\eta_{t-1}^2}{\eta_p^2}$$

$$\leq 3\tau\left(\sigma^2 + \zeta_i\tau + \frac{\zeta}{n}\tau\right) \eta_{t-1}^2.$$

Similarly,

$$\frac{1}{n}\sum_{i=1}^{n} \mathbb{E}\left[\|\boldsymbol{w}^{(t)} - \boldsymbol{w}_i^{(t)}\|^2\right] \leq 3\tau\sigma^2\eta_{t-1}^2 + 6\tau^2\frac{\zeta}{n}\eta_{t-1}^2.$$

$\square$

**Lemma 4.** *(Convergence of global model) Let $\boldsymbol{w}^{(t)} = \frac{1}{n}\sum_{i=1}^{n} \boldsymbol{w}_i^{(t)}$. Under the setting of Theorem 5, we have:*

$$\mathbb{E}\left[\|\boldsymbol{w}^{(T+1)} - \boldsymbol{w}^*\|^2\right] \leq \frac{a^3}{(T+a)^3}\mathbb{E}\left[\|\boldsymbol{w}^{(1)} - \boldsymbol{w}^*\|^2\right]$$

$$+ \left(T + 16\left(\frac{1}{a+1} + \ln(T+a)\right)\right) \frac{1536a^2\tau\left(\sigma^2 + 2\tau\frac{\zeta}{n}\right)L^2}{(a-1)^2\mu^4(T+a)^3} + \frac{128\sigma^2 T(T+2a)}{n\mu^2(T+a)^3}.$$

*Proof.* Using the updating rule and non-expensive property of projection, as well as applying strong convexity and smoothness assumptions yields:

$$\mathbb{E}\left[\|\boldsymbol{w}^{(t+1)} - \boldsymbol{w}^*\|^2\right]$$

$$\leq \mathbb{E}\left[\left\|\boldsymbol{w}^{(t)} - \boldsymbol{w}^{(t)} - \eta_t\frac{1}{n}\sum_{j=1}^{n}\nabla f_j(\boldsymbol{w}_j^{(t)};\xi_j^t) - \boldsymbol{w}^*\right\|^2\right]$$

$$\leq \mathbb{E}\left[\|\boldsymbol{w}^{(t)} - \boldsymbol{w}^*\|^2\right] - 2\eta_t\mathbb{E}\left[\left\langle\frac{1}{n}\sum_{j=1}^{n}\nabla f_j(\boldsymbol{w}_j^{(t)}), \boldsymbol{w}^{(t)} - \boldsymbol{w}^*\right\rangle\right]$$

$$+ \eta_t^2\frac{\sigma^2}{n} + \eta_t^2\mathbb{E}\left[\left\|\frac{1}{n}\sum_{j=1}^{n}\nabla f_j(\boldsymbol{w}_j^{(t)})\right\|^2\right]$$

$$\leq \mathbb{E}\left[\|\boldsymbol{w}^{(t)} - \boldsymbol{w}^*\|^2\right] - 2\eta_t\mathbb{E}\left[\left\langle\nabla F(\boldsymbol{w}^{(t)}), \boldsymbol{w}^{(t)} - \boldsymbol{w}^*\right\rangle\right] + \eta_t^2\frac{\sigma^2}{n} + \eta_t^2\underbrace{\mathbb{E}\left[\left\|\frac{1}{n}\sum_{j=1}^{n}\nabla f_j(\boldsymbol{w}_j^{(t)})\right\|^2\right]}_{T_1}$$

$$\underbrace{- 2\eta_t\mathbb{E}\left[\left\langle\frac{1}{n}\sum_{j=1}^{n}\nabla f_j(\boldsymbol{w}_j^{(t)}) - \nabla F(\boldsymbol{w}^{(t)}), \boldsymbol{w}^{(t)} - \boldsymbol{w}^*\right\rangle\right]}_{T_2}$$

$$\leq (1 - \mu\eta_t)\mathbb{E}\left[\|\boldsymbol{w}^{(t)} - \boldsymbol{w}^*\|^2\right] - 2\eta_t(\mathbb{E}[F(\boldsymbol{w}^{(t)})] - F(\boldsymbol{w}^*)) + \eta_t^2\frac{\sigma^2}{n} + T_1 + T_2, \quad (12)$$

where at the last step we used the strongly convex property.

Now we are going to bound $T_1$. By the Jensen's inequality and smoothness, we have:

$$T_1 \leq 2\eta_t^2 \mathbb{E}\left[\left\|\frac{1}{n}\sum_{j=1}^n \nabla f_j(\boldsymbol{w}_j^{(t)}) - \nabla F(\boldsymbol{w}^{(t)})\right\|^2\right] + 2\eta_t^2 \mathbb{E}\left[\left\|\nabla F(\boldsymbol{w}^{(t)})\right\|^2\right]$$

$$\leq 2\eta_t^2 L^2 \frac{1}{n}\sum_{j=1}^n \mathbb{E}\left[\|\boldsymbol{w}_j^{(t)} - \boldsymbol{w}^{(t)}\|^2\right] + 4\eta_t^2 L\left(\mathbb{E}\left[F(\boldsymbol{w}^{(t)})\right] - F(\boldsymbol{w}^*)\right) \tag{13}$$

Then, we bound $T_2$ as:

$$T_2 \leq \eta_t\left(\frac{2}{\mu}\mathbb{E}\left[\left\|\frac{1}{n}\sum_{j=1}^n \nabla f_j(\boldsymbol{w}_j^{(t)}) - \nabla F(\boldsymbol{w}^{(t)})\right\|^2\right] + \frac{\mu}{2}\mathbb{E}\left[\|\boldsymbol{w}^{(t)} - \boldsymbol{w}^*\|^2\right]\right)$$

$$\leq \frac{2\eta_t L^2}{\mu}\frac{1}{n}\sum_{j=1}^n \mathbb{E}\left[\left\|\boldsymbol{w}_j^{(t)} - \boldsymbol{w}^{(t)}\right\|^2\right] + \frac{\mu\eta_t}{2}\mathbb{E}\left[\|\boldsymbol{w}^{(t)} - \boldsymbol{w}^*\|^2\right]. \tag{14}$$

Now, by plugging back $T_1$ and $T_2$ from (13) and (14) in (12), we have:

$$\mathbb{E}\left[\|\boldsymbol{w}^{(t+1)} - \boldsymbol{w}^*\|^2\right]$$

$$\leq \left(1 - \frac{\mu\eta_t}{2}\right)\mathbb{E}\left[\|\boldsymbol{w}^{(t)} - \boldsymbol{w}^*\|^2\right]\underbrace{-(2\eta_t - 4\eta_t^2 L)}_{\leq -\eta_t}\left(\mathbb{E}\left[F(\boldsymbol{w}^{(t)})\right] - F(\boldsymbol{w}^*)\right) + \eta_t^2\frac{\sigma^2}{n}$$

$$+ \left(\frac{2\eta_t L^2}{\mu} + 2\eta_t^2 L^2\right)\frac{1}{n}\sum_{j=1}^n \mathbb{E}\left[\left\|\boldsymbol{w}_j^{(t)} - \boldsymbol{w}^{(t)}\right\|^2\right] \tag{15}$$

$$\leq \left(1 - \frac{\mu\eta_t}{2}\right)\mathbb{E}\left[\|\boldsymbol{w}^{(t)} - \boldsymbol{w}^*\|^2\right] + \eta_t^2\frac{\sigma^2}{n} + \left(\frac{2\eta_t L^2}{\mu} + 2\eta_t^2 L^2\right)\frac{1}{n}\sum_{j=1}^n \mathbb{E}\left[\left\|\boldsymbol{w}_j^{(t)} - \boldsymbol{w}^{(t)}\right\|^2\right].$$

Now, by using Lemma 3 we have:

$$\mathbb{E}\left[\|\boldsymbol{w}^{(t+1)} - \boldsymbol{w}^*\|^2\right]$$

$$\leq \left(1 - \frac{\mu\eta_t}{2}\right)\mathbb{E}\left[\|\boldsymbol{w}^{(t)} - \boldsymbol{w}^*\|^2\right] + \left(\frac{2\eta_t L^2}{\mu} + 2\eta_t^2 L^2\right)3\tau\left(\sigma^2 + 2\tau\frac{\zeta}{n}\right)\eta_{t-1}^2 + \eta_t^2\frac{\sigma^2}{n}.$$

Note that $(1 - \frac{\mu\eta_t}{2})\frac{p_t}{\eta_t} = \frac{\mu(t+a)^2(t-8+a)}{16} \leq \frac{\mu(t-1+a)^3}{16} = \frac{p_{t-1}}{\eta_{t-1}}$, so we multiply $\frac{p_t}{\eta_t}$ on both sides and do the telescoping sum:

$$\frac{p_T}{\eta_T}\mathbb{E}\left[\|\boldsymbol{w}^{(T+1)} - \boldsymbol{w}^*\|^2\right]$$

$$\leq \frac{p_0}{\eta_0}\mathbb{E}\left[\|\boldsymbol{w}^{(1)} - \boldsymbol{w}^*\|^2\right] + \sum_{t=1}^T\left(\frac{2L^2}{\mu} + 2\eta_t L^2\right)3\tau\left(\sigma^2 + 2\tau\frac{\zeta}{n}\right)p_t\eta_{t-1}^2 + \sum_{t=1}^T p_t\eta_t\frac{\sigma^2}{n}$$

$$\leq \frac{p_0}{\eta_0}\mathbb{E}\left[\|\boldsymbol{w}^{(1)} - \boldsymbol{w}^*\|^2\right] + \sum_{t=1}^T\left(\frac{2L^2}{\mu} + 2\eta_t L^2\right)3\tau\left(\sigma^2 + 2\tau\frac{\zeta}{n}\right)\frac{256a^2}{\mu^2(a-1)^2} + \sum_{t=1}^T p_t\eta_t\frac{\sigma^2}{n}. \tag{16}$$

Then, by re-arranging the terms will conclude the proof:

$$\mathbb{E}\left[\|\boldsymbol{w}^{(T+1)} - \boldsymbol{w}^*\|^2\right]$$

$$\leq \frac{a^3}{(T+a)^3}\mathbb{E}\left[\|\boldsymbol{w}^{(1)} - \boldsymbol{w}^*\|^2\right]$$

$$+ \left(T + 16\left(\frac{1}{a+1} + \ln(T+a)\right)\right)\frac{1536a^2\tau\left(\sigma^2 + 2\tau\frac{\zeta}{n}\right)L^2}{(a-1)^2\mu^4(T+a)^3} + \frac{128\sigma^2 T(T+2a)}{n\mu^2(T+a)^3},$$

where we use the inequality $\sum_{t=1}^T \frac{1}{t+a} \leq \frac{1}{a+1} + \int_1^T \frac{1}{t+a} < \frac{1}{a+1} + \ln(T+a)$. $\qquad\square$

### E.1.2 PROOF OF THEOREM 4

*Proof.* According to (15) and (16) in the proof of Lemma 4 we have:

$$\frac{p_T}{\eta_T}\mathbb{E}\left[\|\boldsymbol{w}^{(T+1)}-\boldsymbol{w}^*\|^2\right] \le \frac{p_0}{\eta_0}\mathbb{E}\left[\|\boldsymbol{w}^{(1)}-\boldsymbol{w}^*\|^2\right] - \sum_{t=1}^T p_t\left(\mathbb{E}\left[F(\boldsymbol{w}^{(t)})\right]-F(\boldsymbol{w}^*)\right)$$
$$+\sum_{t=1}^T\left(\frac{2L^2}{\mu}+2\eta_t L^2\right)3\tau\left(\sigma^2+2\tau\frac{\zeta}{n}\right)\frac{256a^2}{\mu^2(a-1)^2}+\sum_{t=1}^T p_t\eta_t\frac{\sigma^2}{n},$$

re-arranging term and dividing both sides by $S_T = \sum_{t=1}^T p_t > T^3$ yields:

$$\frac{1}{S_T}\sum_{t=1}^T p_t\left(\mathbb{E}\left[F(\boldsymbol{w}^{(t)})\right]-F(\boldsymbol{w}^*)\right) \le \frac{p_0}{S_T\eta_0}\mathbb{E}\left[\|\boldsymbol{w}^{(1)}-\boldsymbol{w}^*\|^2\right]$$
$$+\frac{1}{S_T}\sum_{t=1}^T\left(\frac{2L^2}{\mu}+2\eta_t L^2\right)3\tau\left(\sigma^2+2\tau\frac{\zeta}{n}\right)\frac{256a^2}{\mu^2(a-1)^2}+\frac{1}{S_T}\sum_{t=1}^T p_t\eta_t\frac{\sigma^2}{n}$$
$$\le O\left(\frac{\mu}{T^3}\right)+O\left(\frac{\kappa^2\tau\left(\sigma^2+2\tau\frac{\zeta}{n}\right)}{\mu T^2}\right)+O\left(\frac{\kappa^2\tau\left(\sigma^2+2\tau\frac{\zeta}{n}\right)\ln T}{\mu T^3}\right)+O\left(\frac{\sigma^2}{nT}\right).$$

Recall that $\hat{\boldsymbol{w}} = \frac{1}{nS_T}\sum_{t=1}^T\sum_{j=1}^n \boldsymbol{w}_j^{(t)}$ and convexity of $F$, we can conclude that:

$$\mathbb{E}\left[F(\hat{\boldsymbol{w}})\right]-F(\boldsymbol{w}^*) \le O\left(\frac{\mu}{T^3}\right)+O\left(\frac{\kappa^2\tau\left(\sigma^2+2\tau\frac{\zeta}{n}\right)}{\mu T^2}\right)+O\left(\frac{\kappa^2\tau\left(\sigma^2+2\tau\frac{\zeta}{n}\right)\ln T}{\mu T^3}\right)+O\left(\frac{\sigma^2}{nT}\right).$$

$\square$

### E.1.3 PROOF OF THEOREM 5

*Proof.* Recall that we defined virtual sequences $\{\boldsymbol{w}^{(t)}\}_{t=1}^T$ where $\boldsymbol{w}^{(t)} = \frac{1}{n}\sum_{i=1}^n \boldsymbol{w}_i^{(t)}$ and $\hat{\boldsymbol{v}}_i^{(t)} = \alpha_i\boldsymbol{v}_i^{(t)}+(1-\alpha_i)\boldsymbol{w}^{(t)}$, then by the updating rule and non-expensiveness of projection we have:

$$\mathbb{E}\left[\|\hat{\boldsymbol{v}}_i^{(t+1)}-\boldsymbol{v}_i^*\|^2\right]$$
$$\le \mathbb{E}\left[\left\|\hat{\boldsymbol{v}}_i^{(t)}-\alpha_i^2\eta_t\nabla f_i(\bar{\boldsymbol{v}}_i^{(t)})-(1-\alpha_i)\eta_t\frac{1}{n}\sum_{j=1}^n\nabla f_j(\boldsymbol{w}_j^{(t)})-\boldsymbol{v}_i^*\right\|^2\right]$$
$$+\mathbb{E}\left[\left\|\alpha_i^2\eta_t(\nabla f_i(\bar{\boldsymbol{v}}_i^{(t)})-\nabla f_i(\bar{\boldsymbol{v}}_i^{(t)};\xi_i^t))+(1-\alpha_i)\eta_t\frac{1}{n}\sum_{j\in U_t}\left(\nabla f_j(\boldsymbol{w}_j^{(t)})-\nabla f_j(\boldsymbol{w}_j^{(t)};\xi_j^t)\right)\right\|^2\right]$$
$$\le \mathbb{E}\left[\|\hat{\boldsymbol{v}}_i^{(t)}-\boldsymbol{v}_i^*\|^2\right]-2\mathbb{E}\left[\left\langle\alpha_i^2\eta_t\nabla f_i(\bar{\boldsymbol{v}}_i^{(t)})+(1-\alpha_i)\eta_t\frac{1}{n}\sum_{j=1}^n\nabla f_j(\boldsymbol{w}_j^{(t)}),\hat{\boldsymbol{v}}_i^{(t)}-\boldsymbol{v}_i^*\right\rangle\right]$$
$$+\eta_t^2\mathbb{E}\left[\left\|\alpha_i^2\nabla f_i(\bar{\boldsymbol{v}}_i^{(t)})+(1-\alpha_i)\frac{1}{n}\sum_{j=1}^n\nabla f_j(\boldsymbol{w}_j^{(t)})\right\|^2\right]+\alpha_i^2\eta_t^2\sigma^2+(1-\alpha_i)^2\eta_t^2\frac{\sigma^2}{n}$$
$$= \mathbb{E}\left[\|\hat{\boldsymbol{v}}_i^{(t)}-\boldsymbol{v}_i^*\|^2\right]\underbrace{-2(\alpha_i^2+1-\alpha_i)\eta_t\mathbb{E}\left[\left\langle\nabla f_i(\bar{\boldsymbol{v}}_i^{(t)}),\hat{\boldsymbol{v}}_i^{(t)}-\boldsymbol{v}_i^*\right\rangle\right]}_{T_1}$$
$$\underbrace{-2\eta_t(1-\alpha_i)\mathbb{E}\left[\left\langle\frac{1}{n}\sum_{j=1}^n\nabla f_j(\boldsymbol{w}_j^{(t)})-\nabla f_i(\bar{\boldsymbol{v}}_i^{(t)}),\hat{\boldsymbol{v}}_i^{(t)}-\boldsymbol{v}_i^*\right\rangle\right]}_{T_2}$$
$$\underbrace{+\eta_t^2\mathbb{E}\left[\|\alpha_i^2\nabla f_i(\bar{\boldsymbol{v}}_i^{(t)})+(1-\alpha_i)\frac{1}{n}\sum_{j=1}^n\nabla f_j(\boldsymbol{w}_j^{(t)})\|^2\right]}_{T_3}+\alpha_i^2\eta_t^2\sigma^2+(1-\alpha_i)^2\eta_t^2\frac{\sigma^2}{n}. \tag{17}$$

Now, we bound the term $T_1$ as follows:

$$
\begin{aligned}
T_1 &= -2\eta_t(\alpha_i^2 + 1 - \alpha_i)\mathbb{E}\left[\left\langle \nabla f_i(\hat{\boldsymbol{v}}_i^{(t)}), \hat{\boldsymbol{v}}_i^{(t)} - \boldsymbol{v}_i^* \right\rangle\right] \\
&\quad - 2\eta_t(\alpha_i^2 + 1 - \alpha_i)\mathbb{E}\left[\left\langle \nabla f_i(\bar{\boldsymbol{v}}_i^{(t)}) - \nabla f_i(\hat{\boldsymbol{v}}_i^{(t)}), \hat{\boldsymbol{v}}_i^{(t)} - \boldsymbol{v}_i^* \right\rangle\right] \\
&\leq -2\eta_t(\alpha_i^2 + 1 - \alpha_i)\left(\mathbb{E}\left[f_i(\hat{\boldsymbol{v}}_i^{(t)})\right] - f_i(\boldsymbol{v}_i^*) + \frac{\mu}{2}\mathbb{E}\left[\|\hat{\boldsymbol{v}}_i^{(t)} - \boldsymbol{v}_i^*\|^2\right]\right) \\
&\quad + (\alpha_i^2 + 1 - \alpha_i)\eta_t\left(\frac{8L^2}{\mu(1 - 8(\alpha_i - \alpha_i^2))}\mathbb{E}\left[\|\hat{\boldsymbol{v}}_i^{(t)} - \bar{\boldsymbol{v}}_i^{(t)}\|^2\right] + \frac{\mu(1 - 8(\alpha_i - \alpha_i^2))}{8}\mathbb{E}\left[\|\hat{\boldsymbol{v}}_i^{(t)} - \boldsymbol{v}_i^*\|^2\right]\right) \\
&\leq -2\eta_t(\alpha_i^2 + 1 - \alpha_i)\left(\mathbb{E}\left[f_i(\hat{\boldsymbol{v}}_i^{(t)})\right] - f_i(\boldsymbol{v}_i^*) + \frac{\mu}{2}\mathbb{E}\left[\|\hat{\boldsymbol{v}}_i^{(t)} - \boldsymbol{v}_i^*\|^2\right]\right) \\
&\quad + \eta_t\left(\frac{8L^2(1 - \alpha_i)^2}{\mu(1 - 8(\alpha_i - \alpha_i^2))}\mathbb{E}\left[\|\boldsymbol{w}^{(t)} - \boldsymbol{w}_i^{(t)}\|^2\right] + \frac{\mu(1 - 8(\alpha_i - \alpha_i^2))}{8}\mathbb{E}\left[\|\hat{\boldsymbol{v}}_i^{(t)} - \boldsymbol{v}_i^*\|^2\right]\right) \\
&\leq -2\eta_t(\alpha_i^2 + 1 - \alpha_i)\left(\mathbb{E}\left[f_i(\hat{\boldsymbol{v}}_i^{(t)})\right] - f_i(\boldsymbol{v}_i^*)\right) - \frac{7\mu\eta_t}{8}\mathbb{E}\left[\|\hat{\boldsymbol{v}}_i^{(t)} - \boldsymbol{v}_i^*\|^2\right] \\
&\quad + \frac{8\eta_t L^2(1 - \alpha_i)^2}{\mu(1 - 8(\alpha_i - \alpha_i^2))}\mathbb{E}\left[\|\boldsymbol{w}^{(t)} - \boldsymbol{w}_i^{(t)}\|^2\right],
\end{aligned} \tag{18}
$$

where we use the fact $(\alpha_i^2 + 1 - \alpha_i) \leq 1$. Note that, because we set $\alpha_i \geq \max\{1 - \frac{1}{4\sqrt{6}\kappa}, 1 - \frac{1}{4\sqrt{6}\kappa\sqrt{\mu}}\}$, and hence $1 - 8(\alpha_i - \alpha_i^2) \geq 0$, so in the second inequality we can use the arithmetic-geometry inequality.

Next, we turn to bounding the term $T_2$ in (17):

$$
\begin{aligned}
T_2 &= -2\eta_t(1 - \alpha_i)\mathbb{E}\left[\left\langle \frac{1}{n}\sum_{j=1}^{n}\nabla f_j(\boldsymbol{w}_j^{(t)}) - \nabla f_i(\bar{\boldsymbol{v}}_i^{(t)}), \hat{\boldsymbol{v}}_i^{(t)} - \boldsymbol{v}_i^* \right\rangle\right] \\
&\leq \eta_t(1 - \alpha_i)\left(\frac{2(1 - \alpha_i)}{\mu}\mathbb{E}\left[\left\|\nabla f_i(\bar{\boldsymbol{v}}_i^{(t)}) - \frac{1}{n}\sum_{j=1}^{n}\nabla f_j(\boldsymbol{w}_j^{(t)})\right\|^2\right] + \frac{\mu}{2(1 - \alpha_i)}\mathbb{E}\left[\|\hat{\boldsymbol{v}}_i^{(t)} - \boldsymbol{v}_i^*\|^2\right]\right) \\
&\leq \frac{6(1 - \alpha_i)^2\eta_t}{\mu}\left(\mathbb{E}\left[\left\|\nabla f_i(\bar{\boldsymbol{v}}_i^{(t)}) - \nabla f_i(\hat{\boldsymbol{v}}_i^{(t)})\right\|^2\right] + \mathbb{E}\left[\left\|\nabla f_i(\hat{\boldsymbol{v}}_i^{(t)}) - \nabla F(\boldsymbol{w}^{(t)})\right\|^2\right]\right. \\
&\qquad \left. + \mathbb{E}\left[\left\|\nabla F(\boldsymbol{w}^{(t)}) - \frac{1}{n}\sum_{j=1}^{n}\nabla f_j(\boldsymbol{w}_j^{(t)})\right\|^2\right]\right) + \frac{\eta_t\mu}{2}\mathbb{E}\left[\|\hat{\boldsymbol{v}}_i^{(t)} - \boldsymbol{v}_i^*\|^2\right] \\
&\leq \frac{6(1 - \alpha_i)^2\eta_t}{\mu}\left(L^2\mathbb{E}\left[\left\|\boldsymbol{w}^{(t)} - \boldsymbol{w}_i^{(t)}\right\|^2\right] + \mathbb{E}\left[\left\|\nabla f_i(\hat{\boldsymbol{v}}_i^{(t)}) - \nabla F(\boldsymbol{w}^{(t)})\right\|^2\right]\right. \\
&\qquad \left. + \mathbb{E}\left[\left\|\nabla F(\boldsymbol{w}^{(t)}) - \frac{1}{n}\sum_{j=1}^{n}\nabla f_j(\boldsymbol{w}_j^{(t)})\right\|^2\right]\right) + \frac{\eta_t\mu}{2}\mathbb{E}\left[\|\hat{\boldsymbol{v}}_i^{(t)} - \boldsymbol{v}_i^*\|^2\right].
\end{aligned} \tag{19}
$$

And finally, we bound the term $T_3$ in (17) as follows:

$$
\begin{aligned}
T_3 &= \mathbb{E}\left[\left\|\alpha_i^2\nabla f_i(\bar{\boldsymbol{v}}_i^{(t)}) + (1 - \alpha_i)\frac{1}{n}\sum_{j=1}^{n}\nabla f_j(\boldsymbol{w}_j^{(t)})\right\|^2\right] \\
&\leq 2(\alpha_i^2 + 1 - \alpha_i)^2\mathbb{E}\left[\|\nabla f_i(\bar{\boldsymbol{v}}_i^{(t)})\|^2\right] + 2\mathbb{E}\left[\left\|(1 - \alpha_i)\left(\frac{1}{n}\sum_{j=1}^{n}\nabla f_j(\boldsymbol{w}_j^{(t)}) - \nabla f_i(\bar{\boldsymbol{v}}_i^{(t)})\right)\right\|^2\right] \\
&\leq 2\left(2(\alpha_i^2 + 1 - \alpha_i)^2\mathbb{E}\left[\|\nabla f_i(\hat{\boldsymbol{v}}_i^{(t)}) - \nabla f_i^*\|^2\right] + 2(\alpha_i^2 + 1 - \alpha_i)^2\mathbb{E}\left[\|\nabla f_i(\bar{\boldsymbol{v}}_i^{(t)}) - \nabla f_i(\hat{\boldsymbol{v}}_i^{(t)})\|^2\right]\right) \\
&\quad + 2(1 - \alpha_i)^2\mathbb{E}\left[\left\|\frac{1}{n}\sum_{j=1}^{n}\nabla f_j(\boldsymbol{w}_j^{(t)}) - \nabla f_i(\bar{\boldsymbol{v}}_i^{(t)})\right\|^2\right] \\
&\leq 8L(\alpha_i^2 + 1 - \alpha_i)\left(\mathbb{E}\left[f_i(\hat{\boldsymbol{v}}_i^{(t)})\right] - f_i^*\right) + 4(1 - \alpha_i)^2 L^2\mathbb{E}\left[\|\boldsymbol{w}^{(t)} - \boldsymbol{w}_i^{(t)}\|^2\right] \\
&\quad + 6(1 - \alpha_i)^2\left(L^2\mathbb{E}\left[\left\|\boldsymbol{w}^{(t)} - \boldsymbol{w}_i^{(t)}\right\|^2\right] + \mathbb{E}\left[\left\|\nabla f_i(\hat{\boldsymbol{v}}_i^{(t)}) - \nabla F(\boldsymbol{w}^{(t)})\right\|^2\right]\right. \\
&\qquad \left. + \frac{1}{n}\sum_{j=1}^{n}L^2\mathbb{E}\left[\left\|\boldsymbol{w}^{(t)} - \boldsymbol{w}_j^{(t)}\right\|^2\right]\right).
\end{aligned} \tag{20}
$$

Now, using Lemma 3, $(1 - \alpha_i)^2 \le 1$ and plugging back $T_1$, $T_2$, and $T_3$ from (18), (19), and (20) into (17), yields:

$$
\mathbb{E}\left[\|\hat{\boldsymbol{v}}_i^{(t+1)} - \boldsymbol{v}_i^*\|^2\right]
$$

$$
\le \left(1 - \frac{3\mu\eta_t}{8}\right) \mathbb{E}\left[\|\hat{\boldsymbol{v}}_i^{(t)} - \boldsymbol{v}_i^*\|^2\right] - 2(\eta_t - 4\eta_t^2 L)(\alpha_i^2 + 1 - \alpha_i)\left(\mathbb{E}\left[f_i(\hat{\boldsymbol{v}}_i^{(t)})\right] - f_i(\boldsymbol{v}_i^*)\right)
$$

$$
+ \left(\frac{8\eta_t L^2(1-\alpha_i)^2}{\mu(1 - 8(\alpha_i - \alpha_i^2))} + \frac{6(1-\alpha_i)^2\eta_t L^2}{\mu} + 10(1-\alpha_i)^2\eta_t^2 L^2\right) \mathbb{E}\left[\left\|\boldsymbol{w}^{(t)} - \boldsymbol{w}_i^{(t)}\right\|^2\right]
$$

$$
+ \left(\frac{6(1-\alpha_i)^2\eta_t L^2}{\mu} + 6(1-\alpha_i)^2\eta_t^2 L^2\right) \frac{1}{n}\sum_{j=1}^n \mathbb{E}\left[\left\|\boldsymbol{w}^{(t)} - \boldsymbol{w}_j^{(t)}\right\|^2\right]
$$

$$
+ \left(\frac{6\eta_t}{\mu} + 6\eta_t^2\right)(1-\alpha_i)^2 \mathbb{E}\left[\left\|\nabla F(\boldsymbol{w}^{(t)}) - \nabla f_i(\hat{\boldsymbol{v}}_i^{(t)})\right\|^2\right] + \alpha_i^2\eta_t^2\sigma^2 + (1-\alpha_i)^2\eta_t^2\frac{\sigma^2}{n},
$$

$$
\le \left(1 - \frac{3\mu\eta_t}{8}\right) \mathbb{E}\left[\|\hat{\boldsymbol{v}}_i^{(t)} - \boldsymbol{v}_i^*\|^2\right] - 2(\eta_t - 4\eta_t^2 L)(\alpha_i^2 + 1 - \alpha_i)\left(\mathbb{E}\left[f_i(\hat{\boldsymbol{v}}_i^{(t)})\right] - f_i(\boldsymbol{v}_i^*)\right)
$$

$$
+ \left(\frac{8\eta_t L^2(1-\alpha_i)^2}{\mu(1 - 8(\alpha_i - \alpha_i^2))} + \frac{6(1-\alpha_i)^2\eta_t L^2}{\mu} + 10(1-\alpha_i)^2\eta_t^2 L^2\right) 3\tau\left(\sigma^2 + (\zeta_i + \frac{\varsigma}{n})\tau\right)\eta_{t-1}^2
$$

$$
+ \left(\frac{6(1-\alpha_i)^2\eta_t L^2}{\mu} + 6(1-\alpha_i)^2\eta_t^2 L^2\right) 3\tau\left(\sigma^2 + 2\frac{\varsigma}{n}\tau\right)\eta_{t-1}^2
$$

$$
+ \underbrace{\left(\frac{6\eta_t}{\mu} + 6\eta_t^2\right)(1-\alpha_i)^2 \mathbb{E}\left[\left\|\nabla F(\boldsymbol{w}^{(t)}) - \nabla f_i(\hat{\boldsymbol{v}}_i^{(t)})\right\|^2\right]}_{T_4} + \alpha_i^2\eta_t^2\sigma^2 + (1-\alpha_i)^2\eta_t^2\frac{\sigma^2}{n}, \quad (21)
$$

where using Lemma 2 we can bound $T_4$ as:

$$
T_4 \le \frac{6\eta_t}{\mu}(1-\alpha_i)^2\left(2L^2\mathbb{E}\left[\|\hat{\boldsymbol{v}}_i^{(t)} - \boldsymbol{v}^*\|^2\right] + 6\zeta_i + 6L^2\mathbb{E}\left[\|\boldsymbol{w}^{(t)} - \boldsymbol{w}^*\|^2\right] + 6L^2\Delta_i\right)
$$

$$
+ 6\eta_t^2(1-\alpha_i)^2\left(2L^2\mathbb{E}\left[\|\hat{\boldsymbol{v}}_i^{(t)} - \boldsymbol{v}^*\|^2\right] + 6\zeta_i + 6L^2\mathbb{E}\left[\|\boldsymbol{w}^{(t)} - \boldsymbol{w}^*\|^2\right] + 6L^2\Delta_i\right). \quad (22)
$$

Note that we choose $\alpha_i \ge \max\{1 - \frac{1}{4\sqrt{6}\kappa}, 1 - \frac{1}{4\sqrt{6}\kappa\sqrt{\mu}}\}$, hence $\frac{12L^2(1-\alpha_i)^2}{\mu} \le \frac{\mu}{8}$ and $12L^2(1 - \alpha_i)^2 \le \frac{\mu}{8}$, thereby we have:

$$
T_4 \le \frac{\mu\eta_t}{4}\|\hat{\boldsymbol{v}}_i^{(t)} - \boldsymbol{v}^*\|^2 + 36\eta_t\left(\frac{1}{\mu} + \eta_t\right)(1-\alpha_i)^2\left(\zeta_i + L^2\mathbb{E}\left[\|\boldsymbol{w}^{(t)} - \boldsymbol{w}^*\|^2\right] + L^2\Delta_i\right).
$$

Now, using Lemma 4 we have:

$$
T_4 \le \frac{\mu\eta_t}{4}\mathbb{E}\left[\|\hat{\boldsymbol{v}}_i^{(t)} - \boldsymbol{v}^*\|^2\right] + 36\eta_t\left(\frac{1}{\mu} + \eta_t\right)(1-\alpha_i)^2
$$

$$
\left(\zeta_i + L^2\left(\frac{a^3}{(t+a-1)^3}\mathbb{E}\left[\|\boldsymbol{w}^{(1)} - \boldsymbol{w}^*\|^2\right]\right.\right.
$$

$$
\left.\left. + \left(t + 16\left(\frac{1}{a+1} + \ln(t+a)\right)\right)\frac{1536\tau\left(\sigma^2 + 2\tau\frac{\varsigma}{n}\right)L^2}{\mu^4(t+a-1)^3} + \frac{128\sigma^2 t(t+2a)}{n\mu^2(t+a-1)^3}\right) + L^2\Delta_i\right). \quad (23)
$$

By plugging back $T_4$ from (23) in (21) and using the fact $-(\eta_t - 4\eta_t^2 L) \leq -\frac{1}{2}\eta_t$, and $(\alpha_i^2 + 1 - \alpha_i) \geq \frac{3}{4}$, we have:

$$
\mathbb{E}\left[\|\hat{\boldsymbol{v}}_i^{(t+1)} - \boldsymbol{v}_i^*\|^2\right]
$$

$$
\leq (1 - \frac{\mu\eta_t}{8})\mathbb{E}\left[\|\hat{\boldsymbol{v}}_i^{(t)} - \boldsymbol{v}_i^*\|^2\right] - \frac{3\eta_t}{4}\left(\mathbb{E}\left[f_i(\hat{\boldsymbol{v}}_i^{(t)})\right] - f_i(\boldsymbol{v}_i^*)\right) + \alpha_i^2\eta_t^2\sigma^2 + (1-\alpha_i)^2\eta_t^2\frac{\sigma^2}{n}
$$

$$
+ \left(\frac{8\eta_t L^2(1-\alpha_i)^2}{\mu(1 - 8(\alpha_i - \alpha_i^2))} + \frac{6(1-\alpha_i)^2\eta_t L^2}{\mu} + 10(1-\alpha_i)^2\eta_t^2 L^2\right)3\tau\left(\sigma^2 + (\zeta_i + \frac{\zeta}{n})\tau\right)\eta_{t-1}^2
$$

$$
+ \left(\frac{6(1-\alpha_i)^2\eta_t L^2}{\mu} + 6(1-\alpha_i)^2\eta_t^2 L^2\right)3\tau\left(\sigma^2 + 2\frac{\zeta}{n}\tau\right)\eta_{t-1}^2
$$

$$
+ 36\eta_t\left(\frac{1}{\mu} + \eta_t\right)(1-\alpha_i)^2\left(\zeta_i + L^2\left(\frac{a^3\mathbb{E}\left[\|\boldsymbol{w}^{(1)} - \boldsymbol{w}^*\|^2\right]}{(t-1+a)^3}\right.\right.
$$

$$
+ \left(t + 16\left(\frac{1}{a+1} + \ln(t+a)\right)\right)\frac{1536\tau\left(\sigma^2 + 2\tau\frac{\zeta}{n}\right)L^2}{\mu^4(t+a-1)^3} + \frac{128\sigma^2 t(t+2a)}{n\mu^2(t-1+a)^3} + \Delta_i\bigg)\bigg).
$$

Note that $(1 - \frac{\mu\eta_t}{8})\frac{p_t}{\eta_t} \leq \frac{p_{t-1}}{\eta_{t-1}}$ where $p_t = (t+a)^2$, so, we multiply $\frac{p_t}{\eta_t}$ on both sides, and re-arrange the terms:

$$
\frac{3p_t}{4}\left(\mathbb{E}\left[f_i(\hat{\boldsymbol{v}}_i^{(t)})\right] - f_i(\boldsymbol{v}_i^*)\right)
$$

$$
\leq \frac{p_{t-1}}{\eta_{t-1}}\mathbb{E}\left[\|\hat{\boldsymbol{v}}_i^{(t)} - \boldsymbol{v}_i^*\|^2\right] - \frac{p_t}{\eta_t}\mathbb{E}\left[\|\hat{\boldsymbol{v}}_i^{(t+1)} - \boldsymbol{v}_i^*\|^2\right] + p_t\eta_t\left(\alpha_i^2\sigma^2 + (1-\alpha_i)^2\frac{\sigma^2}{n}\right)
$$

$$
+ \left(\frac{8L^2(1-\alpha_i)^2}{\mu(1 - 8(\alpha_i - \alpha_i^2))} + \frac{6(1-\alpha_i)^2 L^2}{\mu} + 10(1-\alpha_i)^2\eta_t L^2\right)3\tau\left(\sigma^2 + (\zeta_i + \frac{\zeta}{n})\tau\right)p_t\eta_{t-1}^2
$$

$$
+ \left(\frac{6(1-\alpha_i)^2 L^2}{\mu} + 6(1-\alpha_i)^2\eta_t L^2\right)3\tau\left(\sigma^2 + 2\frac{\zeta}{n}\tau\right)p_t\eta_{t-1}^2 + 36p_t\left(\frac{1}{\mu} + \eta_t\right)(1-\alpha_i)^2\left(\zeta_i + L^2\Delta_i\right)
$$

$$
+ 36p_t\left(\frac{1}{\mu} + \eta_t\right)(1-\alpha_i)^2
$$

$$
L^2\left(\frac{a^3}{(t-1+a)^3} + (t + 16\Theta(\ln t))\frac{1536\tau\left(\sigma^2 + 2\tau\frac{\zeta}{n}\right)L^2}{\mu^4(t+a-1)^3} + \frac{128\sigma^2 t(t+2a)}{n\mu^2(t-1+a)^3}\right).
$$

By applying the telescoping sum and dividing both sides by $S_T = \sum_{t=1}^{T} p_t \geq T^3$ we have:

$$f_i(\hat{\boldsymbol{v}}_i) - f_i(\boldsymbol{v}_i^*)$$

$$\leq \frac{1}{S_T} \sum_{t=1}^{T} p_t (f_i(\hat{\boldsymbol{v}}_i^{(t)}) - f_i(\boldsymbol{v}_i^*))$$

$$\leq \frac{4p_0 \mathbb{E}\left[\|\hat{\boldsymbol{v}}_i^{(1)} - \boldsymbol{v}_i^*\|^2\right]}{3\eta_0 S_T} + \frac{1}{S_T} \frac{4}{3} \sum_{t=1}^{T} p_t \eta_t \left(\alpha_i^2 \sigma^2 + (1-\alpha_i)^2 \frac{\sigma^2}{n}\right)$$

$$+ \frac{1}{S_T} \frac{4}{3} \sum_{t=1}^{T} \left(\frac{8L^2(1-\alpha_i)^2}{\mu(1-8(\alpha_i - \alpha_i^2))} + \frac{6(1-\alpha_i)^2 L^2}{\mu} + 10(1-\alpha_i)^2 \eta_t L^2\right) 3\tau \left(\sigma^2 + (\zeta_i + \frac{\zeta}{n})\tau\right) p_t \eta_{t-1}^2$$

$$+ \frac{1}{S_T} \frac{4}{3} \sum_{t=1}^{T} \left(\frac{6(1-\alpha_i)^2 L^2}{\mu} + 6(1-\alpha_i)^2 \eta_t L^2\right) 3\tau \left(\sigma^2 + 2\frac{\zeta}{n}\tau\right) p_t \eta_{t-1}^2$$

$$+ 48(1-\alpha_i)^2$$

$$\frac{L^2}{S_T} \sum_{t=1}^{T} p_t \left(\frac{1}{\mu} + \eta_t\right) \left(\frac{a^3}{(t-1+a)^3} + (t + 16\Theta(\ln t)) \frac{1536\tau \left(\sigma^2 + 2\tau \frac{\zeta}{n}\right) L^2}{\mu^4 (t+a-1)^3} + \frac{128\sigma^2 t(t+2a)}{n\mu^2 (t-1+a)^3}\right)$$

$$+ 48(1-\alpha_i)^2 \left(\zeta_i + L^2 \Delta_i\right) \frac{1}{S_T} \sum_{t=1}^{T} p_t \left(\frac{1}{\mu} + \eta_t\right)$$

$$\leq \frac{4p_0 \mathbb{E}\left[\|\hat{\boldsymbol{v}}_i^{(1)} - \boldsymbol{v}_i^*\|^2\right]}{3\eta_0 S_T} + \frac{32T(T+a)}{3\mu S_T} \left(\alpha_i^2 \sigma^2 + (1-\alpha_i)^2 \frac{\sigma^2}{n}\right)$$

$$+ \frac{4(1-\alpha_i)^2}{3} \left(\frac{8L^2 T}{\mu(1-8(\alpha_i - \alpha_i^2))S_T} + \frac{6L^2 T}{\mu S_T} + \frac{10L^2 \Theta(\ln T)}{\mu S_T}\right) 3\tau \left(\sigma^2 + (\zeta_i + \frac{\zeta}{n})\tau\right) \frac{256a^2}{\mu^2 (a-1)^2}$$

$$+ \frac{4}{3} \left(\frac{6(1-\alpha_i)^2 L^2 T}{\mu S_T} + \frac{6(1-\alpha_i)^2 L^2 \Theta(\ln T)}{\mu S_T}\right) 3\tau \left(\sigma^2 + 2\frac{\zeta}{n}\tau\right) \frac{256a^2}{\mu^2 (a-1)^2}$$

$$+ 48(1-\alpha_i)^2 L^2 \frac{a^2}{(a-1)^2 S_T}$$

$$\left(\frac{a^3 \Theta(\ln T)}{\mu} + \left(\frac{T}{a} + \Theta(\ln T)\right) \frac{1536L^2 \tau \left(\sigma^2 + 2\tau \frac{\zeta}{n}\right)}{\mu^5} + \frac{64(2a+1)\sigma^2 T(T+a)}{na\mu^3}\right)$$

$$+ 48(1-\alpha_i)^2 L^2 \frac{a^2}{(a-1)^2 S_T} \left(\frac{16a^3 \pi^2}{6\mu} + (\Theta(\ln T)) \frac{1536L^2 \tau \left(\sigma^2 + 2\tau \frac{\zeta}{n}\right)}{\mu^5} + \frac{2048(2a+1)\sigma^2}{na\mu^3} T\right)$$

$$+ 48(1-\alpha_i)^2 \left(\zeta_i + L^2 \Delta_i\right) \frac{1}{S_T} \left(\frac{S_T}{\mu} + \frac{8T(T+2a)}{\mu}\right)$$

$$= O\left(\frac{\mu}{T^3}\right) + \alpha_i^2 O\left(\frac{\sigma^2}{\mu T}\right) + (1-\alpha_i)^2 O\left(\frac{\zeta_i}{\mu} + \kappa L \Delta_i\right)$$

$$+ (1-\alpha_i)^2 \left(O\left(\frac{\kappa L \ln T}{T^3}\right) + O\left(\frac{\kappa^2 \sigma^2}{\mu n T}\right) + O\left(\frac{\kappa^2 \tau^2 (\zeta_i + \frac{\zeta}{n}) + \kappa^2 \tau \sigma^2}{\mu T^2}\right) + O\left(\frac{\kappa^4 \tau \left(\sigma^2 + 2\tau \frac{\zeta}{n}\right)}{\mu T^2}\right)\right).$$

where we use the convergence of $\sum_{t=1}^{\infty} \frac{\ln t}{t^2} \to O(1)$, and $\sum_{t=1}^{\infty} \frac{1}{t^2} \to \frac{\pi^2}{6}$.

$\square$

### E.2 PROOF OF CONVERGENCE OF APFL WITH SAMPLING

In this section we will provide the formal proof of the Theorem 2. Before proceed to the proof, we would like to give the convergence of global model here first. The following theorem establishes the convergence of global model in APFL.

**Theorem 6** (Global model convergence of Local Descent APFL). *If each client's objective function is $\mu$-strongly convex and $L$-smooth, and satisfies Assumption 1, using Algorithm 1, by choosing the learning rate $\eta_t = \frac{16}{\mu(t+a)}$, where $a = \max\{128\kappa, \tau\}$, $\kappa = \frac{L}{\mu}$, and using average scheme $\hat{\boldsymbol{w}} = \frac{1}{KS_T} \sum_{t=1}^{T} p_t \sum_{j \in U_t} \boldsymbol{w}_j^{(t)}$, where $p_t = (t+a)^2$, $S_T = \sum_{t=1}^{T} p_t$, and letting $F^*$ to denote the*

*minimum of the F, then the following convergence holds:*

$$\mathbb{E}\left[F(\hat{\boldsymbol{w}})\right] - F^* \leq O\left(\frac{\mu}{T^3}\right) + O\left(\frac{\kappa^2 \tau \left(\sigma^2 + 2\tau \frac{\zeta}{K}\right)}{\mu T^2}\right) + O\left(\frac{\kappa^2 \tau \left(\sigma^2 + 2\tau \frac{\zeta}{K}\right) \ln T}{\mu T^3}\right) + O\left(\frac{\sigma^2}{KT}\right),$$

(24)

*where $\tau$ is the number of local updates (i.e., synchronization gap) .*

*Proof.* The proof is provided in Appendix E.2.2. □

**Remark 5.** *It is noticeable that the obtained rate matches the convergence rate of the FedAvg, and if we choose $\tau = \sqrt{T/K}$, we recover the rate $O(\sqrt{1/KT})$, which is the convergence rate of well-known local SGD with periodic averaging (Woodworth et al., 2020a).*

Now we switch to the proof of the Theorem 2. The proof pipeline is similar to what we did in Appendix E.1.3, non-sampling setting. The only difference is that we use sampling method here, hence, we will introduce the variance depending on sampling size $K$. Now we first begin with the proof of some technique lemmas.

### E.2.1 PROOF OF USEFUL LEMMAS

**Lemma 5.** *For Algorithm 1, at each iteration, the gap between local gradient and global gradient is bounded by*

$$\mathbb{E}\left[\left\|\nabla f_i(\hat{\boldsymbol{v}}_i^{(t)}) - \frac{1}{K}\sum_{j \in U_t} \nabla f_j(\boldsymbol{w}^{(t)})\right\|^2\right]$$

$$\leq 2L^2 \mathbb{E}\left[\|\hat{\boldsymbol{v}}_i^{(t)} - \boldsymbol{v}^*\|^2\right] + 6\left(2\zeta_i + 2\frac{\zeta}{K}\right) + 6L^2 \mathbb{E}\left[\|\boldsymbol{w}^{(t)} - \boldsymbol{w}^*\|^2\right] + 6L^2 \Delta_i.$$

*Proof.* From the smoothness assumption and by applying the Jensen's inequality we have:

$$\mathbb{E}\left[\|\nabla f_i(\hat{\boldsymbol{v}}_i^{(t)}) - \frac{1}{K}\sum_{j \in U_t} \nabla f_j(\boldsymbol{w}^{(t)})\|^2\right]$$

$$\leq 2\mathbb{E}\left[\|\nabla f_i(\hat{\boldsymbol{v}}_i^{(t)}) - \nabla f_i(\boldsymbol{v}_i^*)\|^2\right] + 2\mathbb{E}\left[\|\nabla f_i(\boldsymbol{v}_i^*) - \frac{1}{K}\sum_{j \in U_t} \nabla f_j(\boldsymbol{w}^{(t)})\|^2\right]$$

$$\leq 2L^2 \mathbb{E}\left[\|\hat{\boldsymbol{v}}_i^{(t)} - \boldsymbol{v}^*\|^2\right] + 6\mathbb{E}\left[\|\nabla f_i(\boldsymbol{v}_i^*) - \nabla f_i(\boldsymbol{w}^*)\|^2\right]$$

$$+ 6\mathbb{E}\left[\|\nabla f_i(\boldsymbol{w}^*) - \frac{1}{K}\sum_{j \in U_t} \nabla f_j(\boldsymbol{w}^*)\|^2\right] + 6\mathbb{E}\left[\|\nabla \frac{1}{K}\sum_{j \in U_t} \nabla f_j(\boldsymbol{w}^*) - \frac{1}{K}\sum_{j \in U_t} \nabla f_j(\boldsymbol{w}^{(t)})\|^2\right]$$

$$\leq 2L^2 \mathbb{E}\left[\|\hat{\boldsymbol{v}}_i^{(t)} - \boldsymbol{v}^*\|^2\right] + 6L^2 \mathbb{E}\left[\|\boldsymbol{v}_i^* - \boldsymbol{w}^*\|^2\right] + 6\left(2\zeta_i + 2\frac{1}{K}\sum_{j \in U_t} \zeta_j\right) + 6L^2 \mathbb{E}\left[\|\boldsymbol{w}^{(t)} - \boldsymbol{w}^*\|^2\right]$$

$$\leq 2L^2 \mathbb{E}\left[\|\hat{\boldsymbol{v}}_i^{(t)} - \boldsymbol{v}^*\|^2\right] + 6L^2 \Delta_i + 6\left(2\zeta_i + 2\frac{\zeta}{K}\right) + 6L^2 \mathbb{E}\left[\|\boldsymbol{w}^{(t)} - \boldsymbol{w}^*\|^2\right].$$

□

**Lemma 6** (Local model deviation with sampling). *For Algorithm 1, at each iteration, the deviation between each local version of the global model $\boldsymbol{w}_i^{(t)}$ and the global model $\boldsymbol{w}^{(t)}$ is bounded by:*

$$\mathbb{E}\left[\|\boldsymbol{w}^{(t)} - \boldsymbol{w}_i^{(t)}\|^2\right] \leq 3\tau\sigma^2\eta_{t-1}^2 + 3(\zeta_i + \frac{\zeta}{K})\tau^2\eta_{t-1}^2,$$

$$\frac{1}{K}\sum_{i \in U_t} \mathbb{E}\left[\|\boldsymbol{w}^{(t)} - \boldsymbol{w}_i^{(t)}\|^2\right] \leq 3\tau\sigma^2\eta_{t-1}^2 + 6\tau^2\frac{\zeta}{K}\eta_{t-1}^2.$$

*where $\frac{\zeta}{K} = \frac{1}{K}\sum_{i=1}^n \zeta_i$.*

*Proof.* According to Lemma 8 in Woodworth et al. (2020a):

$$\mathbb{E}\left[\|\boldsymbol{w}^{(t)} - \boldsymbol{w}_i^{(t)}\|^2\right] \leq \frac{1}{K}\sum_{j\in U_t}\mathbb{E}\left[\|\boldsymbol{w}_j^{(t)} - \boldsymbol{w}_i^{(t)}\|^2\right]$$

$$\leq 3\left(\sigma^2 + \zeta_i\tau + \frac{\zeta}{K}\tau\right)\sum_{p=t_c}^{t-1}\eta_p^2\prod_{q=p+1}^{t-1}(1-\mu\eta_q)$$

$$\frac{1}{n}\sum_{i=1}^{n}\mathbb{E}\left[\|\boldsymbol{w}^{(t)} - \boldsymbol{w}_i^{(t)}\|^2\right] \leq \frac{1}{n^2}\sum_{i=1}^{n}\sum_{j=1}^{n}\mathbb{E}\left[\|\boldsymbol{w}_j^{(t)} - \boldsymbol{w}_i^{(t)}\|^2\right]$$

$$\leq 3\left(\sigma^2 + 2\tau\frac{\zeta}{K}\right)\sum_{p=t_c}^{t-1}\eta_p^2\prod_{q=p+1}^{t-1}(1-\mu\eta_q).$$

Then the rest of the proof follows Lemma 3. $\square$

**Lemma 7.** *(Convergence of Global Model) Let* $\boldsymbol{w}^{(t)} = \frac{1}{K}\sum_{j\in U_t}\boldsymbol{w}_j^{(t)}$. *In Theorem 2's setting, using Algorithm 1 by choosing learning rate as* $\eta_t = \frac{16}{\mu(t+a)}$, *we have:*

$$\mathbb{E}\left[\|\boldsymbol{w}^{(T+1)} - \boldsymbol{w}^*\|^2\right] \leq \frac{a^3}{(T+a)^3}\mathbb{E}\left[\|\boldsymbol{w}^{(1)} - \boldsymbol{w}^*\|^2\right]$$

$$+ \left(T + 16\left(\frac{1}{a+1} + \ln(T+a)\right)\right)\frac{1536a^2\tau\left(\sigma^2 + 2\tau\frac{\zeta}{K}\right)L^2}{(a-1)^2\mu^4(T+a)^3} + \frac{128\sigma^2T(T+2a)}{K\mu^2(T+a)^3}.$$

*Proof.* According to the updating rule and non-expensiveness of projection, and the strong convexity we have:

$$\mathbb{E}\left[\|\boldsymbol{w}^{(t+1)} - \boldsymbol{w}^*\|^2\right]$$

$$\leq \mathbb{E}\left[\left\|\boldsymbol{w}^{(t)} - \eta_t\frac{1}{K}\sum_{j\in U_t}\nabla f_j(\boldsymbol{w}_j^{(t)};\xi_j^t) - \boldsymbol{w}^*\right\|^2\right]$$

$$\leq \mathbb{E}\left[\|\boldsymbol{w}^{(t)} - \boldsymbol{w}^*\|^2\right] - 2\eta_t\mathbb{E}\left[\left\langle\frac{1}{K}\sum_{j\in U_t}\nabla f_j(\boldsymbol{w}_j^{(t)}), \boldsymbol{w}^{(t)} - \boldsymbol{w}^*\right\rangle\right]$$

$$+ \eta_t^2\mathbb{E}\left[\left\|\frac{1}{K}\sum_{j\in U_t}\nabla f_j(\boldsymbol{w}_j^{(t)})\right\|^2\right] + \eta_t^2\frac{\sigma^2}{K}$$

$$\leq (1-\mu\eta_t)\mathbb{E}\left[\|\boldsymbol{w}^{(t)} - \boldsymbol{w}^*\|^2\right] - (2\eta_t - 2L\eta_t^2)\mathbb{E}\left[F(\boldsymbol{w}^{(t)}) - F(\boldsymbol{w}^*)\right] + \eta_t^2\frac{\sigma^2}{K}$$

$$+ \eta_t^2\frac{1}{K}\sum_{j\in U_t}L^2\mathbb{E}\left[\left\|\boldsymbol{w}_j^{(t)} - \boldsymbol{w}^{(t)}\right\|^2\right] - 2\eta_t\mathbb{E}\left[\left\langle\frac{1}{K}\sum_{j\in U_t}\nabla f_j(\boldsymbol{w}_j^{(t)}) - \nabla f_j(\boldsymbol{w}^{(t)}), \boldsymbol{w}^{(t)} - \boldsymbol{w}^*\right\rangle\right]$$

$$\leq (1-\mu\eta_t)\mathbb{E}\left[\|\boldsymbol{w}^{(t)} - \boldsymbol{w}^*\|^2\right]\underbrace{-(2\eta_t - 4L\eta_t^2)}_{\leq -\eta_t}\mathbb{E}\left[F(\boldsymbol{w}^{(t)}) - F(\boldsymbol{w}^*)\right] + \eta_t^2\frac{\sigma^2}{K}$$

$$+ 2\eta_t^2L^2\frac{1}{K}\sum_{j\in U_t}\mathbb{E}\left[\left\|\boldsymbol{w}_j^{(t)} - \boldsymbol{w}^{(t)}\right\|^2\right] + \frac{2\eta_tL^2}{\mu}\frac{1}{K}\sum_{j\in U_t}\mathbb{E}\left[\left\|\boldsymbol{w}_j^{(t)} - \boldsymbol{w}^{(t)}\right\|^2\right] + \frac{\mu\eta_t}{2}\mathbb{E}\left[\|\boldsymbol{w}^{(t)} - \boldsymbol{w}^*\|^2\right].$$

$$\tag{25}$$

Then, merging the term, multiplying both sides with $\frac{p_t}{\eta_t}$, and do the telescoping sum yields:

$$
\frac{p_T}{\eta_T} \mathbb{E}\left[\|\boldsymbol{w}^{(T+1)} - \boldsymbol{w}^*\|^2\right] \le \frac{p_0}{\eta_0} \mathbb{E}\left[\|\boldsymbol{w}^{(1)} - \boldsymbol{w}^*\|^2\right] - \mathbb{E}[F(\boldsymbol{w}^{(t)}) - F(\boldsymbol{w}^*)]
$$
$$
+ \sum_{t=1}^{T} \left(\frac{2L^2}{\mu} + 2\eta_t L^2\right) p_t \frac{1}{K} \sum_{j \in U_t} \mathbb{E}\left[\left\|\boldsymbol{w}_j^{(t)} - \boldsymbol{w}^{(t)}\right\|^2\right] + \sum_{t=1}^{T} p_t \eta_t \frac{\sigma^2}{K}.
$$
(26)

Plugging Lemma 6 into (26) yields:

$$
\frac{p_T}{\eta_T} \mathbb{E}\left[\|\boldsymbol{w}^{(T+1)} - \boldsymbol{w}^*\|^2\right] \le \frac{p_0}{\eta_0} \mathbb{E}\left[\|\boldsymbol{w}^{(1)} - \boldsymbol{w}^*\|^2\right] - \mathbb{E}[F(\boldsymbol{w}^{(t)}) - F(\boldsymbol{w}^*)]
$$
$$
+ \sum_{t=1}^{T} \left(\frac{2L^2}{\mu} + 2\eta_t L^2\right) 3 p_t \eta_{t-1}^2 \tau \left(\sigma^2 + 2\tau\frac{\zeta}{K}\right) + \sum_{t=1}^{T} p_t \eta_t \frac{\sigma^2}{K}.
$$
(27)

Then, by re-arranging the terms will conclude the proof as

$$
\mathbb{E}\left[\|\boldsymbol{w}^{(T+1)} - \boldsymbol{w}^*\|^2\right] \le \frac{a^3}{(T+a)^3} \mathbb{E}\left[\|\boldsymbol{w}^{(1)} - \boldsymbol{w}^*\|^2\right]
$$
$$
+ \left(T + 16\left(\frac{1}{a+1} + \ln(T+a)\right)\right) \frac{1536 a^2 L^2 \tau \left(\sigma^2 + 2\tau\frac{\zeta}{K}\right)}{(a-1)^2 \mu^4 (T+a)^3}
$$
$$
+ \frac{128 \sigma^2 T(T+2a)}{K\mu^2(T+a)^3}.
$$

$\square$

### E.2.2 PROOF OF THEOREM 6

*Proof.* According to (28) we have:

$$
\frac{p_T}{\eta_T} \mathbb{E}\left[\|\boldsymbol{w}^{(T+1)} - \boldsymbol{w}^*\|^2\right] \le \frac{p_0}{\eta_0} \mathbb{E}\left[\|\boldsymbol{w}^{(1)} - \boldsymbol{w}^*\|^2\right] - \mathbb{E}[F(\boldsymbol{w}^{(t)}) - F(\boldsymbol{w}^*)]
$$
$$
+ \sum_{t=1}^{T} \left(\frac{2L^2}{\mu} + 2\eta_t L^2\right) 3 p_t \eta_{t-1}^2 \tau \left(\sigma^2 + 2\tau\frac{\zeta}{K}\right) + \sum_{t=1}^{T} p_t \eta_t \frac{\sigma^2}{K}.
$$
(28)

By re-arranging the terms and dividing both sides by $S_T = \sum_{t=1}^{T} p_t > T^3$ yields:

$$
\frac{1}{S_T} \sum_{t=1}^{T} p_t \left(\mathbb{E}\left[F(\boldsymbol{w}^{(t)})\right] - F(\boldsymbol{w}^*)\right)
$$
$$
\le \frac{p_0}{S_T \eta_0} \mathbb{E}\left[\|\boldsymbol{w}^{(1)} - \boldsymbol{w}^*\|^2\right] + \frac{1}{S_T} \sum_{t=1}^{T} \left(\frac{2L^2}{\mu} + 2\eta_t L^2\right) 3 p_t \eta_{t-1}^2 \tau \left(\sigma^2 + 2\tau\frac{\zeta}{K}\right) + \frac{1}{S_T} \sum_{t=1}^{T} p_t \eta_t \frac{\sigma^2}{K}
$$
$$
\le O\left(\frac{\mu \mathbb{E}\left[\|\boldsymbol{w}^{(1)} - \boldsymbol{w}^*\|^2\right]}{T^3}\right) + O\left(\frac{\kappa^2 \tau \left(\sigma^2 + 2\tau\frac{\zeta}{K}\right)}{\mu T^2}\right) + O\left(\frac{\kappa^2 \tau \left(\sigma^2 + 2\tau\frac{\zeta}{K}\right) \ln T}{\mu T^3}\right) + O\left(\frac{\sigma^2}{KT}\right).
$$

Recalling that $\hat{\boldsymbol{w}} = \frac{1}{nS_T} \sum_{t=1}^{T} p_t \sum_{j=1}^{n} \boldsymbol{w}_j^{(t)}$, from the convexity of $F(\cdot)$, we can conclude that

$$
\mathbb{E}\left[F(\hat{\boldsymbol{w}})\right] - F(\boldsymbol{w}^*) \le O\left(\frac{\mu}{T^3}\right) + O\left(\frac{\kappa^2 \tau \left(\sigma^2 + 2\tau\frac{\zeta}{K}\right)}{\mu T^2}\right) + O\left(\frac{\kappa^2 \tau \left(\sigma^2 + 2\tau\frac{\zeta}{K}\right) \ln T}{\mu T^3}\right) + O\left(\frac{\sigma^2}{KT}\right).
$$

$\square$

### E.2.3 PROOF OF THEOREM 2

Now we provide the formal proof of Theorem 2. The main difference from without-sampling setting is that only a subset of local models get updated each period due to partial participation of devices, i.e., $K$ out of all $n$ devices that are sampled uniformly at random. To generalize the proof, we will use an indicator function to model this stochastic update, and show that while the stochastic gradient is unbiased, the variance is changed.

*Proof.* Recall that we defined virtual sequences of $\{\boldsymbol{w}^{(t)}\}_{t=1}^{T}$ where $\boldsymbol{w}^{(t)} = \frac{1}{K}\sum_{j\in U_t}\boldsymbol{w}_i^{(t)}$ and $\hat{\boldsymbol{v}}_i^{(t)} = \alpha_i\boldsymbol{v}_i^{(t)} + (1-\alpha_i)\boldsymbol{w}^{(t)}$. We also define an indicator variable to denote whether $i$th client was selected at iteration $t$:

$$\mathbb{I}_i^t = \left\{ \begin{array}{ll} 1 & if \ i \in U_t \\ 0 & else \end{array} \right.$$

obviously, $\mathbb{E}\left[\mathbb{I}_i^t\right] = \frac{K}{n}$. Then, according to updating rule and non-expensiveness of projection we have:

$$\mathbb{E}\left[\|\hat{\boldsymbol{v}}_i^{(t+1)} - \boldsymbol{v}_i^*\|^2\right]$$

$$\leq \mathbb{E}\left[\left\|\hat{\boldsymbol{v}}_i^{(t)} - \alpha_i^2\mathbb{I}_i^t\eta_t\nabla f_i(\bar{\boldsymbol{v}}_i^{(t)}) - (1-\alpha_i)\eta_t\frac{1}{K}\sum_{j\in U_t}\nabla f_j(\boldsymbol{w}_j^{(t)}) - \boldsymbol{v}_i^*\right\|^2\right]$$

$$+ \mathbb{E}\left[\left\|\alpha_i^2\mathbb{I}_i^t\eta_t\left(\nabla f_i(\bar{\boldsymbol{v}}_i^{(t)}) - \nabla f_i(\bar{\boldsymbol{v}}_i^{(t)};\xi_i^t)\right) + (1-\alpha_i)\eta_t\left(\frac{1}{K}\sum_{j\in U_t}\nabla f_j(\boldsymbol{w}_j^{(t)}) - \frac{1}{K}\sum_{j\in U_t}\nabla f_j(\boldsymbol{w}_j^{(t)};\xi^t)\right)\right\|^2\right]$$

$$= \mathbb{E}\left[\|\hat{\boldsymbol{v}}_i^{(t)} - \boldsymbol{v}_i^*\|^2\right] - 2\left\langle \frac{K}{n}\alpha_i^2\eta_t\nabla f_i(\bar{\boldsymbol{v}}_i^{(t)}) + (1-\alpha_i)\eta_t\frac{1}{K}\sum_{j\in U_t}\nabla f_j(\boldsymbol{w}_j^{(t)}), \hat{\boldsymbol{v}}_i^{(t)} - \boldsymbol{v}_i^*\right\rangle$$

$$+ \eta_t^2\mathbb{E}\left[\left\|\alpha_i^2\mathbb{I}_i^t\nabla f_i(\bar{\boldsymbol{v}}_i^{(t)}) + (1-\alpha_i)\frac{1}{K}\sum_{j\in U_t}\nabla f_j(\boldsymbol{w}_j^{(t)})\right\|^2\right] + \alpha_i^2\eta_t^2\frac{2K^2\sigma^2}{n^2} + (1-\alpha_i)^2\eta_t^2\frac{2\sigma^2}{K}.$$

$$= \mathbb{E}\left[\|\hat{\boldsymbol{v}}_i^{(t)} - \boldsymbol{v}_i^*\|^2\right] \underbrace{-2\eta_t\left\langle\left(\frac{K}{n}\alpha_i^2 + 1 - \alpha_i\right)\nabla f_i(\bar{\boldsymbol{v}}_i^{(t)}), \hat{\boldsymbol{v}}_i^{(t)} - \boldsymbol{v}_i^*\right\rangle}_{T_1}$$

$$\underbrace{-2\eta_t(1-\alpha_i)\mathbb{E}\left[\left\langle\frac{1}{K}\sum_{j\in U_t}\nabla f_j(\boldsymbol{w}_j^{(t)}) - \nabla f_i(\bar{\boldsymbol{v}}_i^{(t)}), \hat{\boldsymbol{v}}_i^{(t)} - \boldsymbol{v}_i^*\right\rangle\right]}_{T_2}$$

$$+ \underbrace{\eta_t^2\mathbb{E}\left[\left\|\alpha_i^2\mathbb{I}_i^t\nabla f_i(\bar{\boldsymbol{v}}_i^{(t)}) + (1-\alpha_i)\frac{1}{K}\sum_{j\in U_t}\nabla f_j(\boldsymbol{w}_j^{(t)})\right\|^2\right]}_{T_3} + \alpha_i^2\eta_t^2\frac{2K^2\sigma^2}{n^2} + (1-\alpha_i)^2\eta_t^2\frac{2\sigma^2}{K}.$$

Now we switch to bound $T_1$:

$$
\begin{aligned}
T_1 &= -2\eta_t(\frac{K}{n}\alpha_i^2 + 1 - \alpha_i)\mathbb{E}\left[\left\langle \nabla f_i(\hat{\boldsymbol{v}}_i^{(t)}), \hat{\boldsymbol{v}}_i^{(t)} - \boldsymbol{v}_i^* \right\rangle\right] \\
&\quad - 2\eta_t(\frac{K}{n}\alpha_i^2 + 1 - \alpha_i)\mathbb{E}\left[\left\langle \nabla f_i(\bar{\boldsymbol{v}}_i^{(t)}) - \nabla f_i(\hat{\boldsymbol{v}}_i^{(t)}), \hat{\boldsymbol{v}}_i^{(t)} - \boldsymbol{v}_i^* \right\rangle\right] \\
&\leq -2\eta_t(\frac{K}{n}\alpha_i^2 + 1 - \alpha_i)\left(\mathbb{E}\left[f_i(\hat{\boldsymbol{v}}_i^{(t)})\right] - f_i(\boldsymbol{v}_i^*) + \frac{\mu}{2}\mathbb{E}\left[\|\hat{\boldsymbol{v}}_i^{(t)} - \boldsymbol{v}_i^*\|^2\right]\right) \\
&\quad + (\frac{K}{n}\alpha_i^2 + 1 - \alpha_i)\eta_t\left(\frac{8L^2}{\mu(1 - 8(\alpha_i - \alpha_i^2\frac{K}{n}))}\mathbb{E}\left[\|\hat{\boldsymbol{v}}_i^{(t)} - \bar{\boldsymbol{v}}_i^{(t)}\|^2\right]\right. \\
&\qquad\qquad \left. + \frac{\mu(1 - 8(\alpha_i - \alpha_i^2\frac{K}{n}))}{8}\mathbb{E}\left[\|\hat{\boldsymbol{v}}_i^{(t)} - \boldsymbol{v}_i^*\|^2\right]\right) \\
&\leq -2\eta_t(\frac{K}{n}\alpha_i^2 + 1 - \alpha_i)\left(\mathbb{E}\left[f_i(\hat{\boldsymbol{v}}_i^{(t)})\right] - f_i(\boldsymbol{v}_i^*) + \frac{\mu}{2}\mathbb{E}\left[\|\hat{\boldsymbol{v}}_i^{(t)} - \boldsymbol{v}_i^*\|^2\right]\right) \\
&\quad + \eta_t\left(\frac{8L^2(1 - \alpha_i)^2}{\mu(1 - 8(\alpha_i - \frac{K}{n}\alpha_i^2))}\mathbb{E}\left[\|\boldsymbol{w}^{(t)} - \boldsymbol{w}_i^{(t)}\|^2\right] + \frac{\mu(1 - 8(\alpha_i - \frac{K}{n}\alpha_i^2))}{8}\mathbb{E}\left[\|\hat{\boldsymbol{v}}_i^{(t)} - \boldsymbol{v}_i^*\|^2\right]\right) \\
&\leq -2\eta_t(\frac{K}{n}\alpha_i^2 + 1 - \alpha_i)\left(\mathbb{E}\left[f_i(\hat{\boldsymbol{v}}_i^{(t)})\right] - f_i(\boldsymbol{v}_i^*)\right) - \frac{7\mu\eta_t}{8}\mathbb{E}\left[\|\hat{\boldsymbol{v}}_i^{(t)} - \boldsymbol{v}_i^*\|^2\right] \\
&\quad + \frac{8\eta_t L^2(1 - \alpha_i)^2}{\mu(1 - 8(\alpha_i - \alpha_i^2))}\mathbb{E}\left[\|\boldsymbol{w}^{(t)} - \boldsymbol{w}_i^{(t)}\|^2\right],
\end{aligned}
\tag{29}
$$

For $T_2$, we use the same approach as we did in (19); To deal with $T_3$, we also employ the similar technique in (20):

$$
\begin{aligned}
T_3 &= \mathbb{E}\left[\left\|\alpha_i^2\mathbb{I}_i^t\nabla f_i(\bar{\boldsymbol{v}}_i^{(t)}) + (1 - \alpha_i)\frac{1}{K}\sum_{j \in U_t}\nabla f_j(\boldsymbol{w}_j^{(t)})\right\|^2\right] \\
&\leq 2(\frac{K}{n}\alpha_i^2 + 1 - \alpha_i)^2\mathbb{E}\left[\|\nabla f_i(\bar{\boldsymbol{v}}_i^{(t)})\|^2\right] + 2\mathbb{E}\left[\left\|(1 - \alpha_i)\left(\frac{1}{K}\sum_{j \in U_t}\nabla f_j(\boldsymbol{w}_j^{(t)}) - \nabla f_i(\bar{\boldsymbol{v}}_i^{(t)})\right)\right\|^2\right] \\
&\leq 2\left(2(\frac{K}{n}\alpha_i^2 + 1 - \alpha_i)^2\mathbb{E}\left[\|\nabla f_i(\hat{\boldsymbol{v}}_i^{(t)}) - \nabla f_i^*\|^2\right]\right. \\
&\qquad\qquad \left. + 2(\frac{K}{n}\alpha_i^2 + 1 - \alpha_i)^2\mathbb{E}\left[\|\nabla f_i(\bar{\boldsymbol{v}}_i^{(t)}) - \nabla f_i(\hat{\boldsymbol{v}}_i^{(t)})\|^2\right]\right) \\
&\quad + 2(1 - \alpha_i)^2\mathbb{E}\left[\left\|\frac{1}{K}\sum_{j \in U_t}\nabla f_j(\boldsymbol{w}_j^{(t)}) - \nabla f_i(\bar{\boldsymbol{v}}_i^{(t)})\right\|^2\right] \\
&\leq 8L(\frac{K}{n}\alpha_i^2 + 1 - \alpha_i)\left(\mathbb{E}\left[f_i(\hat{\boldsymbol{v}}_i^{(t)})\right] - f_i^*\right) + 4(1 - \alpha_i)^2 L^2\mathbb{E}\left[\|\boldsymbol{w}^{(t)} - \boldsymbol{w}_i^{(t)}\|^2\right] \\
&\quad + 6(1 - \alpha_i)^2\left(L^2\mathbb{E}\left[\left\|\boldsymbol{w}^{(t)} - \boldsymbol{w}_i^{(t)}\right\|^2\right] + \mathbb{E}\left[\left\|\nabla f_i(\hat{\boldsymbol{v}}_i^{(t)}) - \nabla F(\boldsymbol{w}^{(t)})\right\|^2\right]\right. \\
&\qquad\qquad \left. + \frac{1}{K}\sum_{j \in U_t}L^2\mathbb{E}\left[\left\|\boldsymbol{w}^{(t)} - \boldsymbol{w}_j^{(t)}\right\|^2\right]\right).
\end{aligned}
\tag{30}
$$

Then plugging $T_1, T_2, T_3$ back, we obtain the similar formulation as the without sampling case in (17). Thus:

$$
\mathbb{E}\left[\|\hat{\boldsymbol{v}}_i^{(t+1)} - \boldsymbol{v}_i^*\|^2\right]
$$
$$
\leq \left(1 - \frac{3\mu\eta_t}{8}\right) \mathbb{E}\left[\|\hat{\boldsymbol{v}}_i^{(t)} - \boldsymbol{v}_i^*\|^2\right] - 2(\eta_t - 4\eta_t^2 L)\left(\alpha_i^2 \frac{K}{n} + 1 - \alpha_i\right)\left(\mathbb{E}\left[f_i(\hat{\boldsymbol{v}}_i^{(t)})\right] - f_i(\boldsymbol{v}_i^*)\right)
$$
$$
+ \alpha_i^2 \eta_t^2 \frac{2K\sigma^2}{n} + (1-\alpha_i)^2 \eta_t^2 \frac{2\sigma^2}{K}
$$
$$
+ \left(\frac{8\eta_t L^2(1-\alpha_i)^2}{\mu(1 - 8(\alpha_i - \alpha_i^2 \frac{K}{n}))} + \frac{6(1-\alpha_i)^2 \eta_t L^2}{\mu} + 10(1-\alpha_i)^2 \eta_t^2 L^2\right) \mathbb{E}\left[\left\|\boldsymbol{w}^{(t)} - \boldsymbol{w}_i^{(t)}\right\|^2\right]
$$
$$
+ \left(\frac{6(1-\alpha_i)^2 \eta_t L^2}{\mu} + 6(1-\alpha_i)^2 \eta_t^2 L^2\right) \frac{1}{K} \sum_{j \in U_t} \mathbb{E}\left[\left\|\boldsymbol{w}^{(t)} - \boldsymbol{w}_j^{(t)}\right\|^2\right]
$$
$$
+ \left(\frac{6\eta_t}{\mu} + 6\eta_t^2\right)(1-\alpha_i)^2 \mathbb{E}\left[\left\|\frac{1}{K}\sum_{j \in U_t} \nabla f_j(\boldsymbol{w}^{(t)}) - \nabla f_i(\hat{\boldsymbol{v}}_i^{(t)})\right\|^2\right]. \tag{31}
$$

we then examine the lower bound of $\alpha_i^2 \frac{K}{n} + 1 - \alpha_i$. Notice that: $\alpha_i^2 \frac{K}{n} + 1 - \alpha_i = \frac{K}{n}((\alpha_i - \frac{n}{2K})^2 + \frac{n}{K} - \frac{n^2}{4K^2})$.

**Case 1:** $\frac{n}{2K} \geq 1$   The lower bound is attained when $\alpha_i = 1$: $\alpha_i^2 \frac{K}{n} + 1 - \alpha_i \geq \frac{K}{n}$.

**Case 2:** $\frac{n}{2K} < 1$   The lower bound is attained when $\alpha_i = \frac{n}{2K}$: $\alpha_i^2 \frac{K}{n} + 1 - \alpha_i \geq 1 - \frac{n}{4K} > \frac{1}{2}$.

So $\alpha_i^2 \frac{K}{n} + 1 - \alpha_i \geq b := \min\{\frac{K}{n}, \frac{1}{2}\}$ always holds.

Now we plug it and Lemma 6 back to (31):

$$
\mathbb{E}\left[\|\hat{\boldsymbol{v}}_i^{(t+1)} - \boldsymbol{v}_i^*\|^2\right]
$$
$$
\leq \left(1 - \frac{3\mu\eta_t}{8}\right) \mathbb{E}\left[\|\hat{\boldsymbol{v}}_i^{(t)} - \boldsymbol{v}_i^*\|^2\right] - b\eta_t \left(\mathbb{E}\left[f_i(\hat{\boldsymbol{v}}_i^{(t)})\right] - f_i(\boldsymbol{v}_i^*)\right) + \alpha_i^2 \eta_t^2 \frac{2K\sigma^2}{n} + (1-\alpha_i)^2 \eta_t^2 \frac{2\sigma^2}{K}
$$
$$
+ \left(\frac{8\eta_t L^2(1-\alpha_i)^2}{\mu(1 - 8(\alpha_i - \alpha_i^2 \frac{K}{n}))} + \frac{6(1-\alpha_i)^2 \eta_t L^2}{\mu} + 10(1-\alpha_i)^2 \eta_t^2 L^2\right) 3\tau\eta_{t-1}^2\left(\sigma^2 + (\zeta_i + \frac{\zeta}{K})\tau\right)
$$
$$
+ \left(\frac{6(1-\alpha_i)^2 \eta_t L^2}{\mu} + 6(1-\alpha_i)^2 \eta_t^2 L^2\right) 3\tau\eta_{t-1}^2\left(\sigma^2 + 2\frac{\zeta}{K}\tau\right)
$$
$$
+ \left(\frac{6\eta_t}{\mu} + 6\eta_t^2\right)(1-\alpha_i)^2 \mathbb{E}\left[\left\|\frac{1}{K}\sum_{j \in U_t} \nabla f_j(\boldsymbol{w}^{(t)}) - \nabla f_i(\hat{\boldsymbol{v}}_i^{(t)})\right\|^2\right]. \tag{32}
$$

Plugging Lemma 5 yields:

$$
\mathbb{E}\left[\|\hat{\boldsymbol{v}}_i^{(t+1)} - \boldsymbol{v}_i^*\|^2\right]
$$
$$
\leq \left(1 - \frac{3\mu\eta_t}{8}\right) \mathbb{E}\left[\|\hat{\boldsymbol{v}}_i^{(t)} - \boldsymbol{v}_i^*\|^2\right] - b\eta_t \left(\mathbb{E}\left[f_i(\hat{\boldsymbol{v}}_i^{(t)})\right] - f_i(\boldsymbol{v}_i^*)\right) + \alpha_i^2 \eta_t^2 \frac{2K\sigma^2}{n} + (1-\alpha_i)^2 \eta_t^2 \frac{2\sigma^2}{K}
$$
$$
+ \left(\frac{8\eta_t L^2(1-\alpha_i)^2}{\mu(1 - 8(\alpha_i - \alpha_i^2 \frac{K}{n}))} + \frac{6(1-\alpha_i)^2 \eta_t L^2}{\mu} + 10(1-\alpha_i)^2 \eta_t^2 L^2\right) 3\tau\eta_{t-1}^2\left(\sigma^2 + (\zeta_i + \frac{\zeta}{K})\tau\right)
$$
$$
+ \left(\frac{6(1-\alpha_i)^2 \eta_t L^2}{\mu} + 6(1-\alpha_i)^2 \eta_t^2 L^2\right) 3\tau\eta_{t-1}^2\left(\sigma^2 + 2\frac{\zeta}{K}\tau\right)
$$
$$
+ \left(\frac{6\eta_t}{\mu} + 6\eta_t^2\right)(1-\alpha_i)^2 \left[2L^2 \mathbb{E}\left[\|\hat{\boldsymbol{v}}_i^{(t)} - \boldsymbol{v}^*\|^2\right] + 6\left(2\zeta_i + 2\frac{\zeta}{K}\right) + 6L^2 \mathbb{E}\left[\|\boldsymbol{w}^{(t)} - \boldsymbol{w}^*\|^2\right] + 6L^2 \Delta_i\right].
$$

Then following the same procedure in Appendix E.1.3, together with the application of Lemma 7 we can conclude that:

$$
f_i(\hat{\boldsymbol{v}}_i) - f_i(\boldsymbol{v}_i^*)
$$

$$
\leq \frac{1}{S_T} \sum_{t=1}^{T} p_t(f_i(\hat{\boldsymbol{v}}_i^{(t)}) - f_i(\boldsymbol{v}_i^*))
$$

$$
\leq \frac{p_0 \mathbb{E}\left[\|\hat{\boldsymbol{v}}_i^{(1)} - \boldsymbol{v}_i^*\|^2\right]}{b\eta_0 S_T} + \frac{1}{bS_T} \sum_{t=1}^{T} p_t \eta_t \left(\alpha_i^2 \eta_t^2 \frac{2K\sigma^2}{n} + (1-\alpha_i)^2 \eta_t^2 \frac{2\sigma^2}{K}\right)
$$

$$
+ \frac{1}{bS_T} \sum_{t=1}^{T} (1-\alpha_i)^2 L^2 \left(\frac{8}{\mu(1 - 8(\alpha_i - \alpha_i^2 \frac{K}{n}))} + \frac{6}{\mu} + 10\eta_t\right) 3\tau p_t \eta_{t-1}^2 \left(\sigma^2 + (\zeta_i + \frac{\zeta}{K})\tau\right)
$$

$$
+ \frac{1}{bS_T} \sum_{t=1}^{T} (1-\alpha_i)^2 L^2 \left(\frac{6}{\mu} + 10\eta_t\right) 3\tau p_t \eta_{t-1}^2 \left(\sigma^2 + 2\frac{\zeta}{K}\tau\right)
$$

$$
+ 36(1-\alpha_i)^2 \frac{L^2}{bS_T} \sum_{t=1}^{T} p_t \left(\frac{1}{\mu} + \eta_t\right) \left(\frac{a^3}{(t-1+a)^3} \mathbb{E}\left[\|\boldsymbol{w}^{(1)} - \boldsymbol{w}^*\|^2\right]\right.
$$

$$
\left. + \left(t + 16\left(\frac{1}{a+1} + \ln(t+a)\right)\right) \frac{1536a^2\tau\left(\sigma^2 + 2\tau\frac{\zeta}{K}\right)L^2}{(a-1)^2\mu^4(t-1+a)^3} + \frac{128\sigma^2 t(t+2a)}{K\mu^2(t-1+a)^3}\right)
$$

$$
+ 36(1-\alpha_i)^2 \left(2\zeta_i + 2\frac{\zeta}{K} + L^2\Delta_i\right) \frac{1}{bS_T} \sum_{t=1}^{T} p_t \left(\frac{1}{\mu} + \eta_t\right).
$$

$$
= O\left(\frac{\mu}{bT^3}\right) + \alpha_i^2 O\left(\frac{\sigma^2}{\mu bT}\right) + (1-\alpha_i)^2 O\left(\frac{2\zeta_i + 2\frac{\zeta}{K}}{\mu b} + \frac{\kappa L\Delta_i}{b}\right)
$$

$$
+ (1-\alpha_i)^2 \left(O\left(\frac{\kappa L \ln T}{bT^3}\right) + O\left(\frac{\kappa^2\sigma^2}{\mu bKT}\right) + O\left(\frac{\kappa^2\tau^2(\zeta_i + \frac{\zeta}{K}) + \kappa^2\tau\sigma^2}{\mu bT^2}\right)\right.
$$

$$
\left. + O\left(\frac{\kappa^4\tau\left(\sigma^2 + 2\tau\frac{\zeta}{K}\right)}{\mu bT^2}\right)\right).
$$

$\square$

# F  CONVERGENCE RATE WITHOUT ASSUMPTION ON $\alpha_i$

In this section, we provide the convergence results of Algorithm 1 without assumption on $\alpha_i$. The following Theorem establish the convergence rate:

**Theorem 7** (Personalized model convergence of Local Descent APFL without assumption on $\alpha_i$). *If each client's objective function is $\mu$-strongly-convex and $L$-smooth, and its gradient is bounded by $G$, using Algorithm 1, learning rate: $\eta_t = \frac{8}{\mu(t+a)}$, where $a = \max\{64\kappa, \tau\}$, and using average scheme $\hat{\boldsymbol{v}}_i = \frac{1}{S_T} \sum_{t=1}^{T} p_t(\alpha_i \boldsymbol{v}_i^{(t)} + (1-\alpha_i)\frac{1}{K}\sum_{j\in U_t} \boldsymbol{w}_j^{(t)})$, where $p_t = (t+a)^2$, $S_T = \sum_{t=1}^{T} p_t$, and $f_i^*$ is the local minimum of the ith client, then the following convergence holds for all $i \in [n]$:*

$$
\mathbb{E}[f_i(\hat{\boldsymbol{v}}_i)] - f_i^* \leq O\left(\frac{\mu}{bT^3}\right) + \alpha_i^2 O\left(\frac{\sigma^2}{\mu bT}\right) + (1-\alpha_i)^2 O\left(\frac{G^2}{\mu b}\right)
$$

$$
+ (1-\alpha_i)^2 \left(O\left(\frac{\kappa L \ln T}{bT^3}\right) + O\left(\frac{\kappa^2\sigma^2}{\mu bKT}\right) + O\left(\frac{\kappa^2\tau^2(\zeta_i + \frac{\zeta}{K}) + \kappa^2\tau\sigma^2}{\mu bT^2}\right)\right.
$$

$$
\left. + O\left(\frac{\kappa^4\tau\left(\sigma^2 + 2\tau\frac{\zeta}{K}\right)}{\mu bT^2}\right)\right), \tag{33}
$$

*where $b = \min\{\frac{K}{n}, \frac{1}{2}\}$*

**Remark 6.** *Here we remove the assumption $\alpha_i \geq \max\{1 - \frac{1}{4\sqrt{6}\kappa}, 1 - \frac{1}{4\sqrt{6}\kappa\sqrt{\mu}}\}$. The key difference is that we can only show the residual error with dependency on $G$, instead of more accurate quantities $\zeta_i$ and $\Delta_i$. Apparently, when the diversity among data shards is small, $\zeta_i$ and $\Delta_i$ terms become small which leads to a tighter convergence rate. Also notice that, to realize the bounded gradient assumption, we need to require the parameters come from a bounded domain $\mathcal{W}$. Thus, we need to do projection during parameter update, which is inexpensive.*

*Proof.* According to (32):

$$\mathbb{E}\left[\|\hat{\boldsymbol{v}}_i^{(t+1)} - \boldsymbol{v}_i^*\|^2\right]$$

$$\leq \left(1 - \frac{3\mu\eta_t}{8}\right)\mathbb{E}\left[\|\hat{\boldsymbol{v}}_i^{(t)} - \boldsymbol{v}_i^*\|^2\right] - b\eta_t\left(\mathbb{E}\left[f_i(\hat{\boldsymbol{v}}_i^{(t)})\right] - f_i(\boldsymbol{v}_i^*)\right)$$

$$+ \alpha_i^2\eta_t^2\frac{2K\sigma^2}{n} + (1-\alpha_i)^2\eta_t^2\frac{2\sigma^2}{K}$$

$$+ \left(\frac{8\eta_t L^2(1-\alpha_i)^2}{\mu(1 - 8(\alpha_i - \alpha_i^2\frac{K}{n}))} + \frac{6(1-\alpha_i)^2\eta_t L^2}{\mu} + 10(1-\alpha_i)^2\eta_t^2 L^2\right)3\tau\eta_{t-1}^2\left(\sigma^2 + (\zeta_i + \frac{\zeta}{K})\tau\right)$$

$$+ \left(\frac{6(1-\alpha_i)^2\eta_t L^2}{\mu} + 6(1-\alpha_i)^2\eta_t^2 L^2\right)3\tau\eta_{t-1}^2\left(\sigma^2 + 2\frac{\zeta}{K}\tau\right)$$

$$+ \left(\frac{6\eta_t}{\mu} + 6\eta_t^2\right)(1-\alpha_i)^2\mathbb{E}\left[\left\|\frac{1}{K}\sum_{j\in U_t}\nabla f_j(\boldsymbol{w}^{(t)}) - \nabla f_i(\hat{\boldsymbol{v}}_i^{(t)})\right\|^2\right].$$

Here, we directly use the bound $\mathbb{E}\left[\left\|\frac{1}{K}\sum_{j\in U_t}\nabla f_j(\boldsymbol{w}^{(t)}) - \nabla f_i(\hat{\boldsymbol{v}}_i^{(t)})\right\|^2\right] \leq 2G^2$. Then we have:

$$\mathbb{E}\left[\|\hat{\boldsymbol{v}}_i^{(t+1)} - \boldsymbol{v}_i^*\|^2\right]$$

$$\leq \left(1 - \frac{\mu\eta_t}{4}\right)\mathbb{E}\left[\|\hat{\boldsymbol{v}}_i^{(t)} - \boldsymbol{v}_i^*\|^2\right] - b\eta_t\left(\mathbb{E}\left[f_i(\hat{\boldsymbol{v}}_i^{(t)})\right] - f_i(\boldsymbol{v}_i^*)\right)$$

$$+ \alpha_i^2\eta_t^2\frac{2K\sigma^2}{n} + (1-\alpha_i)^2\eta_t^2\frac{2\sigma^2}{K}$$

$$+ \left(\frac{8\eta_t L^2(1-\alpha_i)^2}{\mu(1 - 8(\alpha_i - \alpha_i^2\frac{K}{n}))} + \frac{6(1-\alpha_i)^2\eta_t L^2}{\mu} + 10(1-\alpha_i)^2\eta_t^2 L^2\right)3\tau\eta_{t-1}^2\left(\sigma^2 + (\zeta_i + \frac{\zeta}{K})\tau\right)$$

$$+ \left(\frac{6(1-\alpha_i)^2\eta_t L^2}{\mu} + 6(1-\alpha_i)^2\eta_t^2 L^2\right)3\tau\eta_{t-1}^2\left(\sigma^2 + 2\frac{\zeta}{K}\tau\right)$$

$$+ \left(\frac{12\eta_t}{\mu} + 12\eta_t^2\right)(1-\alpha_i)^2 G^2.$$

Then following the same procedure in Appendix E.1.3, we can conclude that:

$$f_i(\hat{\boldsymbol{v}}_i) - f_i(\boldsymbol{v}_i^*)$$

$$\leq \frac{1}{S_T} \sum_{t=1}^{T} p_t (f_i(\hat{\boldsymbol{v}}_i^{(t)}) - f_i(\boldsymbol{v}_i^*))$$

$$\leq \frac{p_0 \mathbb{E}\left[\|\hat{\boldsymbol{v}}_i^{(1)} - \boldsymbol{v}_i^*\|^2\right]}{b\eta_0 S_T} + \frac{1}{bS_T} \sum_{t=1}^{T} p_t \eta_t \left( \alpha_i^2 \eta_t^2 \frac{2K\sigma^2}{n} + (1-\alpha_i)^2 \eta_t^2 \frac{2\sigma^2}{K} \right)$$

$$+ \frac{1}{bS_T} \sum_{t=1}^{T} (1-\alpha_i)^2 L^2 \left( \frac{8}{\mu(1 - 8(\alpha_i - \alpha_i^2 \frac{K}{n}))} + \frac{6}{\mu} + 10\eta_t \right) 3\tau p_t \eta_{t-1}^2 \left( \sigma^2 + (\zeta_i + \frac{\zeta}{K})\tau \right)$$

$$+ \frac{1}{bS_T} \sum_{t=1}^{T} (1-\alpha_i)^2 L^2 \left( \frac{6}{\mu} + 10\eta_t \right) 3\tau p_t \eta_{t-1}^2 \left( \sigma^2 + 2\frac{\zeta}{K}\tau \right)$$

$$+ 12(1-\alpha_i)^2 G^2 \frac{1}{bS_T} \sum_{t=1}^{T} p_t \left( \frac{1}{\mu} + \eta_t \right).$$

$$= O\left(\frac{\mu}{bT^3}\right) + \alpha_i^2 O\left(\frac{\sigma^2}{\mu bT}\right) + (1-\alpha_i)^2 O\left(\frac{G^2}{\mu b}\right)$$

$$+ (1-\alpha_i)^2 \left( O\left(\frac{\kappa L \ln T}{bT^3}\right) + O\left(\frac{\kappa^2 \sigma^2}{\mu bKT}\right) + O\left(\frac{\kappa^2 \tau^2 (\zeta_i + \frac{\zeta}{K}) + \kappa^2 \tau \sigma^2}{\mu bT^2}\right) + O\left(\frac{\kappa^4 \tau (\sigma^2 + 2\tau \frac{\zeta}{K})}{\mu bT^2}\right) \right).$$

$\square$

# G PROOF OF CONVERGENCE RATE IN NONCONVEX SETTING

In this section we will provide the proof of convergence results on nonconvex functions. Let us first present the convergence rate of the global model of APFL, on nonconvex function:

**Theorem 8** (Global model convergence of Local Descent APFL). *If each client's objective function is $L$-smooth, using Algorithm 1 with full gradient, by choosing $K = n$ and learning rate $\eta = \frac{1}{2\sqrt{5}L\sqrt{T}}$, then the following convergence holds:*

$$\frac{1}{T} \sum_{t=1}^{T} \left\| \nabla F(\boldsymbol{w}^{(t)}) \right\|^2 \leq O\left(\frac{L}{\sqrt{T}}\right) + O\left(\frac{\tau^2 \zeta}{nT}\right).$$

*Proof.* The proof is provided in Appendix G.2. $\square$

As usual, let us introduce several useful lemmas before the formal proof of Theorem 3 and 8.

## G.1 PROOF OF TECHNICAL LEMMAS

**Lemma 8.** *Under Theorem 3's assumptions, the following statement holds true:*

$$f_i(\hat{\boldsymbol{v}}_i^{(t+1)}) \leq f_i(\hat{\boldsymbol{v}}_i^{(t)}) - \frac{1}{8}\eta\|\nabla f_i(\hat{\boldsymbol{v}}_i)\|^2$$

$$+ \frac{3}{2}(1-\alpha_i)^2 \eta \frac{1}{n} \sum_{j=1}^{n} \left\| \boldsymbol{w}^{(t)} - \boldsymbol{w}_j^{(t)} \right\|^2 + 3\alpha_i^4 (1-\alpha)^2 \eta L^2 \left\| \boldsymbol{w}^{(t)} - \boldsymbol{w}_i^{(t)} \right\|^2$$

$$+ 6\eta(1-\alpha_i^2)^2 \zeta_i + 12\eta(1-\alpha_i^2)^2 L^2 D_{\mathcal{W}}^2 + 12\eta(\alpha_i - \alpha_i^2)^2 \left\| \nabla F(\boldsymbol{w}^{(t)}) \right\|^2.$$

*Proof.* Define the following quantities:

$$\boldsymbol{g}^{(t)} = \alpha_i^2 \nabla f_i(\bar{\boldsymbol{v}}_i^{(t)}) + (1 - \alpha_i) \frac{1}{n} \sum_{j=1}^{n} \nabla f_j(\boldsymbol{w}_j^{(t)})$$

$$P_{\mathcal{W}}(\hat{\boldsymbol{v}}_i^{(t)}, \boldsymbol{g}^{(t)}, \eta) = \frac{1}{\eta} \left[ \hat{\boldsymbol{v}}_i^{(t)} - \prod_{\mathcal{W}} \left( \hat{\boldsymbol{v}}_i^{(t)} - \eta \left( \alpha_i^2 \nabla f_i(\bar{\boldsymbol{v}}_i^{(t)}) + (1 - \alpha_i) \frac{1}{n} \sum_{j=1}^{n} \nabla f_j(\boldsymbol{w}_j^{(t)}) \right) \right) \right]$$

According to Ghadimi et al. (2016) Lemma 1 :For all $\boldsymbol{w} \in \mathcal{W} \subset \mathbb{R}^d$, $g \in \mathbb{R}^d$ and $\eta > 0$, we have:

$$\langle g, P_{\mathcal{W}}(\boldsymbol{w}, g, \eta) \rangle \geq \| P_{\mathcal{W}}(\boldsymbol{w}, g, \eta) \|^2.$$

According to the updating rule and smoothness of $f_i$, we have:

$$
\begin{aligned}
f_i(\hat{\boldsymbol{v}}_i^{(t+1)}) &\leq f_i(\hat{\boldsymbol{v}}_i^{(t)}) + \left\langle \nabla f_i(\hat{\boldsymbol{v}}_i^{(t)}), \hat{\boldsymbol{v}}_i^{(t+1)} - \hat{\boldsymbol{v}}_i^{(t)} \right\rangle + \frac{L}{2} \left\| \hat{\boldsymbol{v}}_i^{(t+1)} - \hat{\boldsymbol{v}}_i^{(t)} \right\|^2 . \\
&\leq f_i(\hat{\boldsymbol{v}}_i^{(t)}) + \left\langle \nabla f_i(\hat{\boldsymbol{v}}_i^{(t)}), \hat{\boldsymbol{v}}_i^{(t+1)} - \hat{\boldsymbol{v}}_i^{(t)} \right\rangle + \frac{L}{2} \left\| \hat{\boldsymbol{v}}_i^{(t+1)} - \hat{\boldsymbol{v}}_i^{(t)} \right\|^2 \\
&\leq f_i(\hat{\boldsymbol{v}}_i^{(t)}) - \eta \left\langle \nabla f_i(\hat{\boldsymbol{v}}_i^{(t)}), P_{\mathcal{W}}(\hat{\boldsymbol{v}}_i^{(t)}, \boldsymbol{g}^{(t)}, \eta) \right\rangle + \frac{\eta^2 L}{2} \left\| P_{\mathcal{W}}(\hat{\boldsymbol{v}}_i^{(t)}, \boldsymbol{g}^{(t)}, \eta) \right\|^2 \\
&\leq f_i(\hat{\boldsymbol{v}}_i^{(t)}) - \eta \left\langle \boldsymbol{g}^{(t)}, P_{\mathcal{W}}(\hat{\boldsymbol{v}}_i^{(t)}, \boldsymbol{g}^{(t)}, \eta) \right\rangle - \eta \left\langle \nabla f_i(\hat{\boldsymbol{v}}_i^{(t)}) - \boldsymbol{g}^{(t)}, P_{\mathcal{W}}(\hat{\boldsymbol{v}}_i^{(t)}, \boldsymbol{g}^{(t)}, \eta) \right\rangle \\
&\quad + \frac{\eta^2 L}{2} \left\| P_{\mathcal{W}}(\hat{\boldsymbol{v}}_i^{(t)}, \boldsymbol{g}^{(t)}, \eta) \right\|^2
\end{aligned}
$$

$$(34)$$

Using the identity:

$$\left\langle \boldsymbol{g}^{(t)}, P_{\mathcal{W}}(\hat{\boldsymbol{v}}_i^{(t)}, \boldsymbol{g}^{(t)}, \eta) \right\rangle \geq \| \boldsymbol{g}^{(t)} \|^2,$$

and Cauchy-Schwartz inequality that

$$\left\langle \nabla f_i(\hat{\boldsymbol{v}}_i^{(t)}) - \boldsymbol{g}^{(t)}, P_{\mathcal{W}}(\hat{\boldsymbol{v}}_i^{(t)}, \boldsymbol{g}^{(t)}, \eta) \right\rangle \leq \frac{1}{2} \| \nabla f_i(\hat{\boldsymbol{v}}_i^{(t)}) - \boldsymbol{g}^{(t)} \|^2 + \frac{1}{2} \| P_{\mathcal{W}}(\hat{\boldsymbol{v}}_i^{(t)}, \boldsymbol{g}^{(t)}, \eta) \|^2$$

we have:

$$f_i(\hat{\boldsymbol{v}}_i^{(t+1)}) \leq f_i(\hat{\boldsymbol{v}}_i^{(t)}) - \left\|\boldsymbol{g}^{(t)}\right\|^2 + \frac{\eta}{2}\left\|P_{\mathcal{W}}(\hat{\boldsymbol{v}}_i^{(t)}, \boldsymbol{g}^{(t)}, \eta)\right\|^2 + \frac{\eta^2 L}{2}\left\|P_{\mathcal{W}}(\hat{\boldsymbol{v}}_i^{(t)}, \boldsymbol{g}^{(t)}, \eta)\right\|^2$$

$$+ \frac{\eta}{2}\left\|\nabla f_i(\hat{\boldsymbol{v}}_i^{(t)}) - \alpha_i^2 \nabla f_i(\bar{\boldsymbol{v}}_i^{(t)}) - (1-\alpha_i)\frac{1}{n}\sum_{j=1}^{n}\nabla f_j(\boldsymbol{w}_j^{(t)})\right\|^2$$

$$\leq f_i(\hat{\boldsymbol{v}}_i^{(t)}) - \underbrace{\left(\frac{\eta}{2} - \frac{\eta^2 L}{2}\right)}_{\leq \frac{1}{4}\eta}\left\|\boldsymbol{g}^{(t)}\right\|^2$$

$$+ \frac{\eta}{2}\left\|\nabla f_i(\hat{\boldsymbol{v}}_i^{(t)}) - \alpha_i^2 \nabla f_i(\bar{\boldsymbol{v}}_i^{(t)}) - (1-\alpha_i)\frac{1}{n}\sum_{j=1}^{n}\nabla f_j(\boldsymbol{w}_j^{(t)})\right\|^2$$

$$\leq f_i(\hat{\boldsymbol{v}}_i^{(t)}) - \frac{1}{4}\eta\left\|\boldsymbol{g}^{(t)}\right\|^2 + (1-\alpha_i)^2\eta\left\|\nabla F(\boldsymbol{w}^{(t)}) - \frac{1}{n}\sum_{j=1}^{n}\nabla f_j(\boldsymbol{w}_j^{(t)})\right\|^2$$

$$+ \eta\left\|\alpha_i^2\left(\nabla f_i(\hat{\boldsymbol{v}}_i^{(t)}) - \nabla f_i(\bar{\boldsymbol{v}}_i^{(t)})\right) - (1-\alpha_i)\nabla F(\boldsymbol{w}^{(t)}) + (1-\alpha_i^2)\nabla f_i(\hat{\boldsymbol{v}}_i^{(t)})\right\|^2$$

$$\leq f_i(\hat{\boldsymbol{v}}_i^{(t)}) - \frac{1}{4}\eta\left\|\boldsymbol{g}^{(t)}\right\|^2 + (1-\alpha_i)^2\eta\frac{1}{n}\sum_{j=1}^{n}\left\|\boldsymbol{w}^{(t)} - \boldsymbol{w}_j^{(t)}\right\|^2 + 2\eta\left\|\alpha_i^2\left(\nabla f_i(\hat{\boldsymbol{v}}_i^{(t)}) - \nabla f_i(\bar{\boldsymbol{v}}_i^{(t)})\right)\right\|^2$$

$$+ 2\eta\left\|(1-\alpha_i^2)\nabla f_i(\hat{\boldsymbol{v}}_i^{(t)}) - (1-\alpha_i^2)\nabla F(\hat{\boldsymbol{v}}_i^{(t)}) + (1-\alpha_i^2)\nabla F(\hat{\boldsymbol{v}}_i^{(t)}) - (1-\alpha_i)\nabla F(\boldsymbol{w}^{(t)})\right\|^2$$

$$\leq f_i(\hat{\boldsymbol{v}}_i^{(t)}) - \frac{1}{4}\eta\left\|\boldsymbol{g}^{(t)}\right\|^2 + (1-\alpha_i)^2\eta\frac{1}{n}\sum_{j=1}^{n}\left\|\boldsymbol{w}^{(t)} - \boldsymbol{w}_j^{(t)}\right\|^2 + 2\alpha_i^4\eta L^2\left\|\hat{\boldsymbol{v}}_i^{(t)} - \bar{\boldsymbol{v}}_i^{(t)}\right\|^2$$

$$+ 4\eta\left\|(1-\alpha_i^2)\nabla f_i(\hat{\boldsymbol{v}}_i^{(t)}) - (1-\alpha_i^2)\nabla F(\hat{\boldsymbol{v}}_i^{(t)})\right\|^2 + 4\eta\left\|(1-\alpha_i^2)\nabla F(\hat{\boldsymbol{v}}_i^{(t)}) - (1-\alpha_i)\nabla F(\boldsymbol{w}^{(t)})\right\|^2$$

$$\leq f_i(\hat{\boldsymbol{v}}_i^{(t)}) - \frac{1}{4}\eta\left\|\boldsymbol{g}^{(t)}\right\|^2 + (1-\alpha_i)^2\eta\frac{1}{n}\sum_{j=1}^{n}\left\|\boldsymbol{w}^{(t)} - \boldsymbol{w}_j^{(t)}\right\|^2 + 2\alpha_i^4(1-\alpha)^2\eta L^2\left\|\boldsymbol{w}^{(t)} - \boldsymbol{w}_i^{(t)}\right\|^2$$

$$+ 4\eta(1-\alpha_i^2)^2\zeta_i + 8\eta\left\|(1-\alpha_i^2)\nabla F(\hat{\boldsymbol{v}}_i^{(t)}) - (1-\alpha_i^2)\nabla F(\boldsymbol{w}^{(t)})\right\|^2 + 8\eta\left\|(\alpha_i - \alpha_i^2)\nabla F(\boldsymbol{w}^{(t)})\right\|^2$$

$$\leq f_i(\hat{\boldsymbol{v}}_i^{(t)}) - \frac{1}{4}\eta\left\|\boldsymbol{g}^{(t)}\right\|^2 + (1-\alpha_i)^2\eta\frac{1}{n}\sum_{j=1}^{n}\left\|\boldsymbol{w}^{(t)} - \boldsymbol{w}_j^{(t)}\right\|^2 + 2\alpha_i^4(1-\alpha)^2\eta L^2\left\|\boldsymbol{w}^{(t)} - \boldsymbol{w}_i^{(t)}\right\|^2$$

$$+ 4\eta(1-\alpha_i^2)^2\zeta_i + 8\eta(1-\alpha_i^2)^2 L^2 D_{\mathcal{W}}^2 + 8\eta(\alpha_i - \alpha_i^2)^2\left\|\nabla F(\boldsymbol{w}^{(t)})\right\|^2$$

Using the following inequality to replace $\|\boldsymbol{g}^{(t)}\|^2$:

$$\|\nabla f_i(\hat{\boldsymbol{v}}_i)\|^2 \leq 2\|\nabla f_i(\hat{\boldsymbol{v}}_i) - \boldsymbol{g}^{(t)}\|^2 + 2\|\boldsymbol{g}^{(t)}\|^2$$

hence we can conclude the proof:

$$f_i(\hat{\boldsymbol{v}}_i^{(t+1)}) \le f_i(\hat{\boldsymbol{v}}_i^{(t)}) - \frac{1}{4}\eta \left[\frac{1}{2}\|\nabla f_i(\hat{\boldsymbol{v}}_i)\|^2 - \|\nabla f_i(\hat{\boldsymbol{v}}_i) - \boldsymbol{g}^{(t)}\|^2\right]$$

$$+ (1-\alpha_i)^2\eta\frac{1}{n}\sum_{j=1}^{n}\left\|\boldsymbol{w}^{(t)} - \boldsymbol{w}_j^{(t)}\right\|^2 + 2\alpha_i^4(1-\alpha)^2\eta L^2\left\|\boldsymbol{w}^{(t)} - \boldsymbol{w}_i^{(t)}\right\|^2$$

$$+ 4\eta(1-\alpha_i^2)^2\zeta_i + 8\eta(1-\alpha_i^2)^2L^2D_{\mathcal{W}}^2 + 8\eta(\alpha_i - \alpha_i^2)^2\left\|\nabla F(\boldsymbol{w}^{(t)})\right\|^2$$

$$\le f_i(\hat{\boldsymbol{v}}_i^{(t)}) - \frac{1}{8}\eta\|\nabla f_i(\hat{\boldsymbol{v}}_i)\|^2 + \frac{1}{4}\eta\left\|\nabla f_i(\hat{\boldsymbol{v}}_i) - \left(\alpha_i^2\nabla f_i(\bar{\boldsymbol{v}}_i^{(t)}) + (1-\alpha_i)\frac{1}{n}\sum_{j=1}^{n}\nabla f_j(\boldsymbol{w}_j^{(t)})\right)\right\|^2$$

$$+ (1-\alpha_i)^2\eta\frac{1}{n}\sum_{j=1}^{n}\left\|\boldsymbol{w}^{(t)} - \boldsymbol{w}_j^{(t)}\right\|^2 + 2\alpha_i^4(1-\alpha)^2\eta L^2\left\|\boldsymbol{w}^{(t)} - \boldsymbol{w}_i^{(t)}\right\|^2$$

$$+ 4\eta(1-\alpha_i^2)^2\zeta_i + 8\eta(1-\alpha_i^2)^2L^2D_{\mathcal{W}}^2 + 8\eta(\alpha_i - \alpha_i^2)^2\left\|\nabla F(\boldsymbol{w}^{(t)})\right\|^2$$

$$\le f_i(\hat{\boldsymbol{v}}_i^{(t)}) - \frac{1}{8}\eta\|\nabla f_i(\hat{\boldsymbol{v}}_i)\|^2$$

$$+ \frac{3}{2}(1-\alpha_i)^2\eta\frac{1}{n}\sum_{j=1}^{n}\left\|\boldsymbol{w}^{(t)} - \boldsymbol{w}_j^{(t)}\right\|^2 + 3\alpha_i^4(1-\alpha)^2\eta L^2\left\|\boldsymbol{w}^{(t)} - \boldsymbol{w}_i^{(t)}\right\|^2$$

$$+ 6\eta(1-\alpha_i^2)^2\zeta_i + 12\eta(1-\alpha_i^2)^2L^2D_{\mathcal{W}}^2 + 12\eta(\alpha_i - \alpha_i^2)^2\left\|\nabla F(\boldsymbol{w}^{(t)})\right\|^2.$$

$\square$

**Lemma 9.** *Under Theorem 3's assumptions, the following statement holds true:*

$$\frac{1}{T}\sum_{t=1}^{T}\left\|\nabla F(\boldsymbol{w}^{(t)})\right\|^2 \le \frac{8}{\eta T}F(\boldsymbol{w}^{(1)}) + 6L^2\frac{1}{T}\sum_{t=1}^{T}\frac{1}{n}\sum_{j=1}^{n}\left\|\boldsymbol{w}_j^{(t)} - \boldsymbol{w}^{(t)}\right\|^2.$$

*Proof.* Define the following quantities:

$$\bar{\boldsymbol{g}}^{(t)} = \frac{1}{n}\sum_{j=1}^{n}\nabla f_j(\boldsymbol{w}_j^{(t)})$$

$$R^{(t)} = P_{\mathcal{W}}(\boldsymbol{w}^{(t)}, \bar{\boldsymbol{g}}^{(t)}, \eta) = \frac{1}{\eta}\left[\boldsymbol{w}^{(t)} - \prod_{\mathcal{W}}\left(\boldsymbol{w}^{(t)} - \eta\frac{1}{n}\sum_{j=1}^{n}\nabla f_j(\boldsymbol{w}_j^{(t)})\right)\right]$$

According to the updating rule and smoothness of $f_i$, we have:

$$F(\boldsymbol{w}^{(t+1)}) \le F(\boldsymbol{w}^{(t)}) + \left\langle\nabla F(\boldsymbol{w}^{(t)}), \boldsymbol{w}^{(t+1)} - \boldsymbol{w}^{(t)}\right\rangle + \frac{L}{2}\left\|\boldsymbol{w}^{(t+1)} - \boldsymbol{w}^{(t)}\right\|^2$$

$$\le F(\boldsymbol{w}^{(t)}) - \eta\left\langle\nabla F(\boldsymbol{w}^{(t)}), P_{\mathcal{W}}(\boldsymbol{w}^{(t)}, \bar{\boldsymbol{g}}^{(t)}, \eta)\right\rangle + \frac{\eta^2 L}{2}\left\|P_{\mathcal{W}}(\boldsymbol{w}^{(t)}, \bar{\boldsymbol{g}}^{(t)}, \eta)\right\|^2$$

$$\le F(\boldsymbol{w}^{(t)}) - \eta\left\langle\bar{\boldsymbol{g}}^{(t)}, P_{\mathcal{W}}(\boldsymbol{w}^{(t)}, \bar{\boldsymbol{g}}^{(t)}, \eta)\right\rangle - \eta\left\langle\nabla F(\boldsymbol{w}^{(t)}) - \bar{\boldsymbol{g}}^{(t)}, P_{\mathcal{W}}(\boldsymbol{w}^{(t)}, \bar{\boldsymbol{g}}^{(t)}, \eta)\right\rangle$$

$$+ \frac{\eta^2 L}{2}\left\|P_{\mathcal{W}}(\boldsymbol{w}^{(t)}, \bar{\boldsymbol{g}}^{(t)}, \eta)\right\|^2$$

$$(35)$$

Using the identity: $\left\langle\bar{\boldsymbol{g}}^{(t)}, P_{\mathcal{W}}(\boldsymbol{w}^{(t)}, \bar{\boldsymbol{g}}^{(t)}, \eta)\right\rangle \ge \|\bar{\boldsymbol{g}}^{(t)}\|^2$, and Cauchy-Schwartz inequality that $\left\langle\nabla F(\boldsymbol{w}^{(t)}) - \bar{\boldsymbol{g}}^{(t)}, P_{\mathcal{W}}(\boldsymbol{w}^{(t)}, \bar{\boldsymbol{g}}^{(t)}, \eta)\right\rangle \le \frac{1}{2}\|\nabla F(\boldsymbol{w}^{(t)}) - \bar{\boldsymbol{g}}^{(t)}\|^2 + \frac{1}{2}\|P_{\mathcal{W}}(\boldsymbol{w}^{(t)}, \bar{\boldsymbol{g}}^{(t)}, \eta)\|^2$ we have:

$$F(\boldsymbol{w}^{(t+1)}) \le F(\boldsymbol{w}^{(t)}) - \eta \left\| \bar{\boldsymbol{g}}^{(t)} \right\|^2 + \left( \frac{\eta}{2} + \frac{\eta^2 L}{2} \right) \left\| P_{\mathcal{W}}(\boldsymbol{w}^{(t)}, \bar{\boldsymbol{g}}^{(t)}, \eta) \right\|^2 + \frac{\eta}{2} \left\| \nabla F(\boldsymbol{w}^{(t)}) - \bar{\boldsymbol{g}}^{(t)} \right\|^2$$

$$\le F(\boldsymbol{w}^{(t)}) - \eta \left\| \bar{\boldsymbol{g}}^{(t)} \right\|^2 + \left( \frac{\eta}{2} + \frac{\eta^2 L}{2} \right) \left\| \bar{\boldsymbol{g}}^{(t)} \right\|^2 + \frac{\eta L^2}{2} \frac{1}{n} \sum_{i=1}^{n} \left\| \boldsymbol{w}_j^{(t)} - \boldsymbol{w}^{(t)} \right\|^2.$$

$$\le F(\boldsymbol{w}^{(t)}) - \frac{1}{4} \eta \left\| \bar{\boldsymbol{g}}^{(t)} \right\|^2 + \frac{\eta L^2}{2} \frac{1}{n} \sum_{i=1}^{n} \left\| \boldsymbol{w}_j^{(t)} - \boldsymbol{w}^{(t)} \right\|^2.$$

Using the following inequality to replace $\|\boldsymbol{g}^{(t)}\|^2$:

$$\|\nabla F(\boldsymbol{w}^{(t)})\|^2 \le 2\|\nabla F(\boldsymbol{w}^{(t)}) - \bar{\boldsymbol{g}}^{(t)}\|^2 + 2\|\bar{\boldsymbol{g}}^{(t)}\|^2$$

hence we obtain:

$$F(\boldsymbol{w}^{(t+1)}) \le F(\boldsymbol{w}^{(t)}) - \frac{1}{4} \eta \left( \frac{1}{2} \|\nabla F(\boldsymbol{w}^{(t)})\|^2 - \|\nabla F(\boldsymbol{w}^{(t)}) - \bar{\boldsymbol{g}}^{(t)}\|^2 \right) + \frac{\eta L^2}{2} \frac{1}{n} \sum_{i=1}^{n} \left\| \boldsymbol{w}_j^{(t)} - \boldsymbol{w}^{(t)} \right\|^2$$

$$\le F(\boldsymbol{w}^{(t)}) - \frac{1}{8} \eta \|\nabla F(\boldsymbol{w}^{(t)})\|^2 + \frac{3\eta L^2}{4} \frac{1}{n} \sum_{i=1}^{n} \left\| \boldsymbol{w}_j^{(t)} - \boldsymbol{w}^{(t)} \right\|^2.$$

Re-arranging terms and doing the telescoping sum from $t = 1$ to $T$:

$$\frac{1}{T} \sum_{t=1}^{T} \left\| \nabla F(\boldsymbol{w}^{(t)}) \right\|^2 \le \frac{8}{\eta T} F(\boldsymbol{w}^{(1)}) + 6L^2 \frac{1}{T} \sum_{t=1}^{T} \frac{1}{n} \sum_{j=1}^{n} \left\| \boldsymbol{w}_j^{(t)} - \boldsymbol{w}^{(t)} \right\|^2.$$

$\square$

**Lemma 10.** *Under Theorem 3's assumptions, the following statement holds true:*

$$\frac{1}{T} \sum_{t=1}^{T} \frac{1}{n} \sum_{i=1}^{n} \left\| \boldsymbol{w}^{(t)} - \boldsymbol{w}_i^{(t)} \right\|^2 \le 10\tau^2 \eta^2 \frac{\zeta}{n},$$

$$\frac{1}{T} \sum_{t=1}^{T} \left\| \boldsymbol{w}^{(t)} - \boldsymbol{w}_i^{(t)} \right\|^2 \le 200 L^2 \tau^4 \eta^4 \frac{\zeta}{n} + 20\tau^2 \eta^2 \zeta_i.$$

*Proof.* For the first statement, we define $\gamma_t = \frac{1}{n} \sum_{i=1}^{n} \left\| \boldsymbol{w}^{(t)} - \boldsymbol{w}_i^{(t)} \right\|^2$, and let $t_c$ be the latest synchronization stage. Then we have:

$$\gamma_t = \frac{1}{n} \sum_{i=1}^{n} \left\| \boldsymbol{w}^{t_c} - \sum_{j=t_c}^{t} \frac{\eta}{n} \sum_{k=1}^{n} \nabla f_k(\boldsymbol{w}_k^{(j)}) - \left( \boldsymbol{w}^{t_c} - \sum_{j=t_c}^{t} \eta \nabla f_i(\boldsymbol{w}_i^{(j)}) \right) \right\|^2$$

$$= \tau \sum_{j=t_c}^{t} \frac{\eta^2}{n} \sum_{i=1}^{n} \left\| \frac{1}{n} \sum_{k=1}^{n} \nabla f_k(\boldsymbol{w}_k^{(j)}) - \nabla f_i(\boldsymbol{w}_i^{(j)}) \right\|^2$$

$$= \tau \sum_{j=t_c}^{t} \frac{\eta^2}{n} \sum_{i=1}^{n} \left\| \frac{1}{n} \sum_{k=1}^{n} \nabla f_k(\boldsymbol{w}_k^{(j)}) - \nabla f_k(\boldsymbol{w}^{(j)}) + \nabla f_k(\boldsymbol{w}^{(j)}) - \nabla f_i(\boldsymbol{w}^{(j)}) + \nabla f_i(\boldsymbol{w}^{(j)}) - \nabla f_i(\boldsymbol{w}_i^{(j)}) \right\|^2$$

$$\le \tau \sum_{j=t_c}^{t+\tau} 5\eta^2 \left( 2L^2 \gamma^j + \frac{\zeta}{n} \right).$$

Summing over $t$ from $t_c$ to $t_c + \tau$ yields:

$$\sum_{t=t_c}^{t_c+\tau} \gamma_t \leq \sum_{t=t_c}^{t_c+\tau} \sum_{j=t_c}^{t_c+\tau} 5\tau\eta^2 \left( 2L^2\gamma^j + \frac{\zeta}{n} \right)$$

$$\leq 10L^2\tau^2\eta^2 \sum_{j=r\tau}^{(r+1)\tau} \gamma^j + 5\tau^3\eta^2 \frac{\zeta}{n}.$$

Since $\eta \leq \frac{1}{2\sqrt{5}\tau L}$, we have $10L^2\tau^2\eta^2 \leq \frac{1}{2}$, hence by re-arranging the terms we have:

$$\sum_{t=t_c}^{t_c+\tau} \gamma_t \leq 10\tau^3\eta^2 \frac{\zeta}{n}.$$

Summing over all synchronization stages $t_c$, and dividing both sides by $T$ can conclude the proof of the first statement:

$$\frac{1}{T} \sum_{t=1}^{T} \gamma_t \leq 10\tau^2\eta^2 \frac{\zeta}{n}. \tag{36}$$

To prove the second statement, let $\delta_t^i = \left\| \boldsymbol{w}^{(t)} - \boldsymbol{w}_i^{(t)} \right\|^2$. Notice that:

$$\delta_t^i = \left\| \boldsymbol{w}^{t_c} - \sum_{j=t_c}^{t} \frac{\eta}{n} \sum_{k=1}^{n} \nabla f_k(\boldsymbol{w}_k^{(j)}) - \left( \boldsymbol{w}^{t_c} - \sum_{j=t_c}^{t} \eta \nabla f_i(\boldsymbol{w}_i^{(j)}) \right) \right\|^2$$

$$= \tau \sum_{j=t_c}^{t} \eta^2 \left\| \frac{1}{n} \sum_{k=1}^{n} \nabla f_k(\boldsymbol{w}_k^{(j)}) - \nabla f_i(\boldsymbol{w}_i^{(j)}) \right\|^2$$

$$= \tau \sum_{j=t_c}^{t} \eta^2 \left\| \frac{1}{n} \sum_{k=1}^{n} \nabla f_k(\boldsymbol{w}_k^{(j)}) - \nabla f_k(\boldsymbol{w}^{(j)}) + \nabla f_k(\boldsymbol{w}^{(j)}) - \nabla f_i(\boldsymbol{w}^{(j)}) + \nabla f_i(\boldsymbol{w}^{(j)}) - \nabla f_i(\boldsymbol{w}_i^{(j)}) \right\|^2$$

$$\leq \tau \sum_{j=t_c}^{t+\tau} 5\eta^2 \left( L^2\gamma_j + L^2\delta_j^i + \zeta_i \right).$$

Summing over $t$ from $t_c$ to $t_c + \tau$ yields:

$$\sum_{t=t_c}^{t_c+\tau} \gamma_t \leq \sum_{t=t_c}^{t_c+\tau} \sum_{j=t_c}^{t_c+\tau} 5\tau\eta^2 \left( L^2\gamma_j + L^2\delta_j^i + \zeta_i \right)$$

$$\leq 5L^2\tau^2\eta^2 \sum_{j=t_c}^{t_c+\tau} \gamma_j + 5L^2\tau^2\eta^2 \sum_{j=t_c}^{t_c+\tau} \delta_j^i + 5\tau^3\eta^2\zeta_i.$$

Since $\eta \leq \frac{1}{2\sqrt{5}\tau L}$, we have $5L^2\tau^2\eta^2 \leq \frac{1}{4}$, hence by re-arranging the terms we have:

$$\sum_{t=t_c}^{t_c+\tau} \delta_t^i \leq 20L^2\tau^2\eta^2 \sum_{j=t_c}^{t_c+\tau} \gamma_j + 20\tau^3\eta^2\zeta_i.$$

Summing over all synchronization stages $t_c$, and dividing both sides by $T$ can conclude the proof of the first statement:

$$\frac{1}{T} \sum_{t=1}^{T} \delta_t^i \leq 20L^2\tau^2\eta^2 \frac{1}{T} \sum_{t=1}^{T} \gamma_t + 20\tau^2\eta^2\zeta_i.$$

Using the result from (36) to bound $2\frac{1}{T} \sum_{t=1}^{T} \gamma_t$ we have:

$$\frac{1}{T} \sum_{t=1}^{T} \delta_t^i \leq 200L^2\tau^4\eta^4 \frac{\zeta}{n} + 20\tau^2\eta^2\zeta_i.$$

$\square$

## G.2  PROOF OF THEOREM 8

*Proof.* According to Lemma 9:

$$\frac{1}{T}\sum_{t=1}^{T}\left\|\nabla F(\boldsymbol{w}^{(t)})\right\|^2 \leq \frac{8}{\eta T}F(\boldsymbol{w}^{(1)}) + 6L^2\frac{1}{T}\sum_{t=1}^{T}\frac{1}{n}\sum_{j=1}^{n}\left\|\boldsymbol{w}_j^{(t)} - \boldsymbol{w}^{(t)}\right\|^2.$$

Then plugging in Lemma 10 will conclude the proof.

$\square$

## G.3  PROOF OF THEOREM 3

*Proof.* According to Lemma 8:

$$\begin{aligned}
f_i(\hat{\boldsymbol{v}}_i^{(t+1)}) \leq\ & f_i(\hat{\boldsymbol{v}}_i^{(t)}) - \frac{1}{8}\eta\|\nabla f_i(\hat{\boldsymbol{v}}_i)\|^2 \\
& + \frac{3}{2}(1-\alpha_i)^2\eta\frac{1}{n}\sum_{j=1}^{n}\left\|\boldsymbol{w}^{(t)} - \boldsymbol{w}_j^{(t)}\right\|^2 + 3\alpha_i^4(1-\alpha)^2\eta L^2\left\|\boldsymbol{w}^{(t)} - \boldsymbol{w}_i^{(t)}\right\|^2 \\
& + 6\eta(1-\alpha_i^2)^2\zeta_i + 12\eta(1-\alpha_i^2)^2L^2D_{\mathcal{W}}^2 + 12\eta(\alpha_i - \alpha_i^2)^2\left\|\nabla F(\boldsymbol{w}^{(t)})\right\|^2.
\end{aligned}$$

Re-arranging the terms, summing from $t=1$ to $T$, and dividing both sides with $T$ yields:

$$\begin{aligned}
&\frac{1}{T}\sum_{t=1}^{T}\left\|\nabla f_i(\hat{\boldsymbol{v}}_i^{(t)})\right\|^2 \\
\leq\ & \frac{8f_i(\hat{\boldsymbol{v}}_i^{(1)})}{\eta T} \\
& + 24\alpha_i^4(1-\alpha_i)^2L^2\frac{1}{T}\sum_{t=1}^{T}\left\|\boldsymbol{w}_i^{(t)} - \boldsymbol{w}^{(t)}\right\|^2 + 12(1-\alpha_i)^2L^2\frac{1}{n}\sum_{j=1}^{n}\frac{1}{T}\sum_{t=1}^{T}\left\|\boldsymbol{w}_j^{(t)} - \boldsymbol{w}^{(t)}\right\|^2 \\
& + 128(1-\alpha_i)^2\frac{1}{T}\sum_{t=1}^{T}\left\|\nabla F(\boldsymbol{w}^{(t)})\right\|^2 + 48(1-\alpha_i^2)^2\zeta_i + 128(1-\alpha_i^2)^2L^2D_{\mathcal{W}},
\end{aligned}$$

Then, plug in Lemma 9 and 10 :

$$\begin{aligned}
&\frac{1}{T}\sum_{t=1}^{T}\left\|\nabla f_i(\hat{\boldsymbol{v}}_i^{(t)})\right\|^2 \\
\leq\ & \frac{8f_i(\hat{\boldsymbol{v}}_i^{(1)})}{\eta T} + 48(1-\alpha_i^2)^2\zeta_i + 128(1-\alpha_i^2)^2L^2D_{\mathcal{W}} \\
& + 24\alpha_i^4(1-\alpha_i)^2L^2\frac{1}{T}\sum_{t=1}^{T}\left\|\boldsymbol{w}_i^{(t)} - \boldsymbol{w}^{(t)}\right\|^2 + 12(1-\alpha_i)^2L^2\frac{1}{n}\sum_{j=1}^{n}\frac{1}{T}\sum_{t=1}^{T}\left\|\boldsymbol{w}_j^{(t)} - \boldsymbol{w}^{(t)}\right\|^2 \\
& + 128(1-\alpha_i)^2\left(\frac{8}{\eta T}F(\boldsymbol{w}^{(1)}) + 6L^2\frac{1}{T}\sum_{t=1}^{T}\frac{1}{n}\sum_{j=1}^{n}\left\|\boldsymbol{w}_j^{(t)} - \boldsymbol{w}^{(t)}\right\|^2\right) \\
\leq\ & \frac{8f_i(\hat{\boldsymbol{v}}_i^{(1)})}{\eta T} + 48(1-\alpha_i^2)^2\zeta_i + 128(1-\alpha_i^2)^2L^2D_{\mathcal{W}} \\
& + 24\alpha_i^4(1-\alpha_i)^2L^2\left[200L^2\tau^4\eta^4\frac{\zeta}{n} + 20\tau^2\eta^2\zeta_i\right] \\
& + 7800\tau^2\eta^2(1-\alpha_i)^2L^2\frac{\zeta}{n} + \frac{1024(1-\alpha_i)^2}{\eta T}F(\boldsymbol{w}^{(1)}).
\end{aligned}$$

Plugging in $\eta = \frac{1}{2\sqrt{5}\sqrt{T}L}$ concludes the proof:

$$\frac{1}{T}\sum_{t=1}^{T}\left\|\nabla f_i(\hat{\boldsymbol{v}}_i^{(t)})\right\|^2$$

$$\leq O\left(\frac{L}{\sqrt{T}}\right) + (1-\alpha_i)^2\left(\frac{L}{\sqrt{T}}\right) + (1-\alpha_i^2)^2\left(\zeta_i + L^2 D_{\mathcal{W}}\right)$$

$$+ \alpha_i^4(1-\alpha_i)^2 O\left(\frac{\tau^4\zeta}{nT^2} + \frac{\tau^2\zeta_i}{T}\right) + (1-\alpha_i)^2 O\left(\frac{\tau^2\zeta}{nT}\right).$$

$\square$

