# OpenReview forum: "Adaptive Personalized Federated Learning"
_ICLR.cc/2021/Conference — Reject_

### Official Review · AnonReviewer1 · 2020-10-26
**when $\alpha$ gets smaller, the personalized models do not perform closer to the global model on validation data**

**Rating:** 6
**Confidence:** 3

**Review:**

In this paper, the authors propose a variant of FedAvg that not only produce the global training, but also a mixture of the local model and the global model, which is called personalized model. As a result, the personalized models show better validation accuracy. The theoretical and empirical analysis are provided.
In overall, I think the paper is technically sound, making reasonable contributions.

However, some parts of the paper is unclear to me, which may affect the final score:

1. In the exerpiments in Figure 2, it is still unclear to me how the "local validation" dataset is sampled, even after I read the appendix. The most important thing is, on each worker, does the validation dataset shares the same subsets of labels as the local training data? If not, then why call it "local"? Also, for the results in Table 1 and 2, are the validation dataset also "local"? My concern is that, if the local validation data is close to the local training data, and is not sampling from the global distribution, then this criterion is too beneficial to APFL, and not fair for the global models.

2. Theoretically, when $\alpha$ gets smaller, the personalized models also gets closer to the global model. Especially, when $\alpha=0$, the presonalized model is equivalent to the global model, which is equivalent to the result of FedAvg. However, the empirical results indicate the opposite. In all the experiments (Figure 2, Table 1 and 2), when $\alpha$ gets smaller, the validation accuracy gets farther away from FedAvg. Even with a pretty small $\alpha$ (=0.25), the gap between APFL and FedAvg is still huge. However, the $\alpha=0$ case seems not discussed in the theoretical analysis. Furthermore, the mismatch between the theoretical and empirical results when $\alpha$ gets smaller is not explained.

---

> ### Author Response · Authors · 2020-11-19
> **Response to Reviewer 1**
>
> Thanks for your valuable reviews. We are trying to address your concern as follows:
>
> - **Question 1**) ***Local validation dataset:*** For the local validation dataset, we use the local data and randomly split it into two parts of training and validation. As for personalization approaches, this seems to be a common approach in the literature and more realistic since in practice the user would have the same distribution as training for inference. Since the local validation data are drawn from the same distribution of training data, that is the key motivation that we propose to mix some local models to improve the generalization on local distributions. We also note that per  R4's suggestion, we also conducted experiments on an alternative data distribution schemas (Dirichlet distribution) and reported the results in Appendix B which is consistent with our earlier observations.
>
>
> - **Question 2**) ***Value of $\alpha$ gets smaller:*** First of all, thanks for this nice observation. If you look into the generalization bound, its dependency on $\alpha$ is not linear. So, we cannot simply assume that when $\alpha$ decreases, the personalized model performs more like FedAvg. In fact, since the generalization bound is quadratic in $\alpha$, it is totally possible that $\alpha = 0.25$ are close to the optimal $\alpha$, and thus the personalized model outperforms FedAvg too much. We added a discussion about this observation in the experiment section and highlighted for your consideration.

---

### Official Review · AnonReviewer2 · 2020-10-28
**Complete story on optimization and generalization; Questions regarding the formulation and the adaptive case**

**Rating:** 5
**Confidence:** 4

**Review:**

This paper studies the personalization aspect of the federated learning problem. The authors propose a new framework in which they replace the common global model in the original federated learning formulation with a convex combination of the global model and a local model. They later introduce an adaptive optimization algorithm for their formulation and provide generalization bounds as well as convergence guarantees for both strongly-convex and nonconvex settings.  Finally, numerical experiments and comparison with other methods in the literature are provided to support the theoretical results.

The paper has a complete story. It provides generalization bounds, optimization results for both strongly convex and nonconvex functions, as well as experiments and comparison with other works.

However, I am slightly confused regarding the proposed formulation. In Section 2.1, for any agent $i$, and for a mixing weight $\alpha_i$, $h_{loc,i}^*$ is defined as (I suppressed the hat notation over $h_{loc,i}^*$ and calligraphic font due to some compilation error here).

$h_{loc,i}^* = argmin_{h \in H} \hat{L}_{D_i}(\alpha_i h + (1-\alpha_i) \bar{h}^*).$

The authors then take $h_{\alpha_i} := \alpha_i h_{loc,i}^* + (1-\alpha_i) \bar{h}^*$ as the output, and claim that this is not the minimizer of $\hat{L}_{D_i}$, since "we optimize $h_{loc,i}^*$ with partially incorporating $\bar{h}^*$."

However, I am not sure if I follow the argument here, and it is not clear to me why $h_{\alpha_i}$ is not the minimizer of $\hat{L}_{D_i}$. Let's take $H = \mathbb{R}^d$, as stated in Section 4. Also, let's denote the minimizer of $\hat{L}_{D_i}$ as $h_{opt,i}$. Then, the solution to the minimization problem above would be

$h_{loc,i}^* = (h_{opt,i} - (1-\alpha_i) \bar{h}^*) / \alpha_i$

which leads to having $h_{\alpha_i} = h_{opt,i}$, the minimizer of $\hat{L}_{D_i}$.

I have the same confusion regarding Section 3 as well. There, the goal is to solve two minimization problems

$ w^* = argmin_{w \in \mathbb{R}^d} \frac{1}{n} \sum_{i=1}^n f_i(w), \quad v_i^* =  argmin_{v \in \mathbb{R}^d} f_i(\alpha_i v+ (1-\alpha_i)w^*),$

and the output of the algorithm, for instance in Theorem 2, is $\alpha_i v_i+ (1-\alpha_i)w^*$ which is the minimizer of $f_i$. If this is the case, why we need to find $w^*$, and why not solving $argmin_{v \in \mathbb{R}^d} f_i(v)$ directly?

It is completely possible that I am missing something here. But I would appreciate it if authors provide a clarification on this.

As a second question, I am wondering if there is any connection between the sequence $\alpha_i^{(t)}$ and the optimal $\alpha_i^*$ derived in Section 2. In other words, for instance for the strongly convex case, does the sequence $\alpha_i^{(t)}$ converge to any particular value close to $\alpha_i^*$?

---

> ### Author Response · Authors · 2020-11-19
> **Response to reviewer 2**
>
> Thanks for your very constructive comments. Your argument is correct. If the hypothesis class is unbounded (like $\mathbb{R}^d$), then yes the mixed model will coincide with the local ERM model. Hence we need to consider the bounded hypothesis class, e.g., $\mathcal{H} = \\{\boldsymbol{h}: \|\boldsymbol{h}\|_2 \leq 1 \\}$. We note that our generalization relies on the assumption that the loss function is bounded and in Remark 3 we highlight that this assumption requires learning from a bounded hypothesis space. We revised the manuscript and clarified the bounded hypothesis class assumption, and also updated the corresponding part in the algorithm and convergence theory to make them consistent. The changes and new convergence theorem are highlighted in blue. We really appreciate raising this concern that enhanced the clarity of statements.
>
>
>
> Let us illustrate  a simple situation where mixed model does not necessarily coincide with local ERM model. To this end, consider a setting where the hypothesis class $\mathcal{H}$ is the set of all vectors in $\mathbb{R}^2$, lying in $\ell_2$ unit ball:
>
> $\mathcal{H}  = \\{\boldsymbol{h} \in \mathbb{R}^2: \|\boldsymbol{h}\|_2 \leq 1 \\}$
>
> Assume the local empirical minimizer is known to be $[1,0]^{\top}$, and $\bar{\boldsymbol{h}}^* = [-1,0]^{\top}$, and $\alpha$ is set to be 0.5. Now, if we wish to  find a $\\hat{\\boldsymbol{h}}_{loc,i}^*$, such that
>
> $\boldsymbol{h}_{\alpha_i} = \alpha \hat{\boldsymbol{h}}_{loc,i}^* + (1-\alpha) \bar{\boldsymbol{h}}^*$
>
>  coincides with local empirical minimizer, we have to solve:
>
> $0.5\boldsymbol{h} + 0.5[-1,0]^{\top} = [1,0]^{\top}$, subject to $\|\boldsymbol{h}\|_2 \leq 1$.
>
> This equation has no feasible solution. That is why we claim that it is not necessarily true that $\boldsymbol{h}_{\alpha_i}$ coincides with local empirical minimizer.
>
>
>
> For your second question about the convergence of $\alpha$, we admit that we have not developed the convergence theory for the proposed adaptive schema, since when there are three parameters to optimize, the analysis will be more involved and we leave it as a future direction.

---

> > ### Comment · AnonReviewer2 · 2020-11-25
> > **Response to authors**
> >
> > Thank you very much for for your response. Yes, I agree that now, and with this boundedness assumption imposed on the hypothesis class, the solution to the local ERM model and the mixed model might be different.
> >
> > I went over the paper one more time to review the updated results. Regarding the non-convex setting, I am wondering why the given guarantees are with respect to the gradient norm. The gradient norm might not even get zero over the constrained set (let's say set $\mathcal{C}$). So the result cannot really show how far you are from the optimal over  $\mathcal{C}$.
> >
> > In fact, in the constrained setting, we know [1] that if $x^*$ is local-minima then $\nabla f(x^*)^\top (x-x^*) \geq 0$ for any $x \in \mathcal{C}$.  As a result, a more common approach for non-convex constrained problem is to find $x^*$ such that
> > $\nabla f(x^*)^\top (x-x^*) \geq -\epsilon$ (see [2] for instance).
> >
> > References
> >
> > [1] Dimitri P Bertsekas. Nonlinear programming. Athena scientific Belmont, 1999.
> >
> > [2] Simon Lacoste-Julien. Convergence rate of Frank-Wolfe for non-convex objectives. arXiv preprint arXiv:1607.00345, 2016.

---

### Official Review · AnonReviewer3 · 2020-10-30
**Good story about federated learning.**

**Rating:** 7
**Confidence:** 3

**Review:**

## 1. Summary

I do apologize for delaying the review process. I do spend lots of time and carefully read the paper. All comments listed below intend to help authors improve the quality of the manuscript. They are based on my understanding which might contain misunderstanding points if any. I hope comments are helpful and even the critiques are not discouraging your endeavor in the following.

First of all, the manuscript proposed an adaptive personalized federated learning algorithm, where each client will train their local models while contributing to the global model. The convergence analysis and generalization bound for the algorithm are conducted.

## 2. Major Comments:

- Could please emphasize the motivation in figure 1, I cannot follow your statement.

- The authors defined the average distribution $\bar{\mathcal{D}}=1/n\sum_{i=1}^n\mathcal{D}_i$. Why the global model can be obtained by minimizing the joint empirical distribution $\bar{\mathcal{D}}$ in the federated learning?

- Why you only consider the smooth loss?

- The optimal mixing parameter is the theoretical one. It seems cannot be used in practices. How do you think?

## 3. Minor Comments:

- Page 5, the first line, it should be ``model $v_i^{(t)}$''.

- Definition 2 you should cite ``Li, Xiang, Kaixuan Huang, Wenhao Yang, Shusen Wang, and Zhihua Zhang. "On the convergence of fedavg on non-iid data." arXiv preprint arXiv:1907.02189 (2019).''

---

> ### Author Response · Authors · 2020-11-19
> **Response to Reviewer 3**
>
> Thanks for your valuable reviewing. We will try to address your concerns as follows:
>
> - **Question 1**) ***Elaboration on plot in Figure 1:*** Sorry for the confusion. In Fig. 1, the x-axis is the number of classes of training data we use to sample the  data for each client, from 10 to 1 in MNIST dataset. If each client has less classes of training data, the heterogeneity among them will be high. If each of them has all classes of training data, their training data are almost identical, and thus heterogeneity is low. As we can see in Fig. 1, when the heterogeneity goes higher (the number of classes of training data per client decreases), FedAvg or SCAFFOLD generalizes worse, while personalized model significantly outperforms them.
>
>
> - **Question 2**) ***Why the global model can be obtained by minimizing average ERM in federated learning:*** Since we are minimizing the average empirical loss $\frac{1}{n}\sum_{i}^n f_i(\boldsymbol{w})$ (usually each local empirical loss is weighted by $m_i/m$), which is equivalent to minimize the risk on empirical distribution $\hat{\mathcal{D}} = \frac{1}{n}\sum_{i=1}^n \hat{\mathcal{D}}_i$, where $\hat{\mathcal{D}}_i$ is the empirical distribution of $i$th client (please see [1]). Hence, we obtained global model from empirical average distribution $\hat{\mathcal{D}}$. However, the key motivation of our work is that this model, while minimizing the ERM, does not necessarily exhibit good generalization performance on individual local distributions when the heterogeneity among local distributions is high.
>
> - **Question 3**) ***Why only smooth losses:*** So far we can only do convergence analysis of local descent method on smooth loss, which is a common assumption in the FL community. As far as we know, there is no convergence analysis for local methods for non-smooth losses which is an interesting open question. We note that due to periodic averaging, this might be involved or requires some tweaks to make it work.
>
> - **Question 4**) ***Optimal mixing in practice:*** Yes the optimal mixing parameter is hard to obtain in practice, and that is why we propose our adaptive optimization method on $\alpha$. We note that since raw data can not be shared, it is challenging to measure the quality of individual data sources to assess the heterogeneity and it is impediment to devise adaptive mechanisms.
>
> We would also appreciate the additional reference, but Li et al seemingly used different heterogeneity characterization instead of gradient dissimilarity as employed in our analysis which is customary in doing ERM in federated setting.
>
> [1] Mohri, Mehryar, Gary Sivek, and Ananda Theertha Suresh. "Agnostic federated learning." arXiv preprint arXiv:1902.00146 (2019).

---

### Official Review · AnonReviewer4 · 2020-11-02
**Significant overlap with prior work**

**Rating:** 3
**Confidence:** 4

**Review:**

==================== Post rebuttal ====================

I thank the authors for their response and additional experiments. Please see my comments below.

**Relation to [1]** The first version of [1] appeared on arxiv in February. I am not sure if it has been published since then or not, but regardless it is sufficiently ahead of the ICLR submission deadline to consider it a prior work. [1] provides a Mapper optimization algorithm in Figure 4. There are some minor differences with your algorithm, but I don't see how they make your algorithm more communication efficient. Generalization analysis in [1] is for the mixing parameter learned from data (i.e. adaptive), that is why there is no dependency on it. Your analysis is for a fixed mixing parameter, but since it is not possible to know it in advance, learning it from data seems to be more reasonable. So I am not sure what is the advantage of a theorem with explicit dependence on it. I agree that the convergence analysis is new, but [1] also has two more algorithms.

**Experiments** Despite that your paper has more experiments, EMNIST experiment in [1] is better suited for studying personalization in my opinion. EMNIST results in Appendix B in the submission are for **digits only** (and also use fewer clients, but that is less important). One of the reasons that a centrally trained model on EMNIST performs worse than personalized models is the shift in the distribution of characters and digits across clients. This is not the case for MNIST/CIFAR10: if I train a neural net using all of MNIST train data it will easily achieve 99+ average test accuracy (without any personalization). How the test data is split across clients does not matter because accuracy on each digit is roughly the same. That is what I meant by a "single global model with good performance (i.e. trained on the full dataset)".

Having worked on FL with personalization myself, I've noticed that it is quite hard to achieve a meaningful improvement over the vanilla FedAvg + fine-tuning. This is also evident from Table 2 of [1], where none of their algorithms offer a convincing improvement. This paper claims that an algorithm very similar to Mapper [1] indeed outperforms FedAvg + fine-tuning. I'd be happy if it is so, but I do not find the provided experimental evidence sufficient. I recommend reproducing the EMNIST experiment of [1] (please don't discard characters, use the same number of clients per communication, etc.). If your algorithm can achieve 91+ accuracy, I'd consider it an improvement. In that case, a more detailed discussion of the differences with Mapper [1] that enabled the improvement would also be great.

Regarding the number of parameters, based on eq. (1), personalized model is a convex combination of **two** models. So if I understood correctly, the number of parameters is increased both during training and testing.

====================================================


This paper considers the problem of personalization in federated learning. The proposed approach consists of allowing each client to have a local model for personalization and taking a convex combination of global and local models as personalized preditors. Authors also provide a theoretical analysis of the generalization and convergence guarantees and empirical evaluation.

The proposed method is equivalent to the Mapper algorithm from [1]. Generalization analysis is also very similar.

The empirical study is not conducted carefully:
- Allowing for a separate local model for personalization increases the number of parameters which is not accounted for
- Reported performance of the global model trained with FedAvg on MNIST and CIFAR10 is overly pessimistic. Prior works (e.g. [2]) report much better performance for the non-personalized model in similar heterogeneous data splitting scenarios.
- Splitting MNIST and CIFAR10 datasets by classes is not a realistic way of simulating heterogeneous distributions in my opinion (see [3,4] for more recent strategies). It also does not seem appropriate for studying personalization as we know that there is a single global model with good performance (i.e. trained on the full dataset) and the key challenge is the optimization over heterogeneous datasets rather than the need for personalization. I recommend datasets such as FEMNIST [5] for studying personalization.

[1] Three Approaches for Personalization with Applications to Federated Learning
[2] Communication-Efficient Learning of Deep Networks from Decentralized Data
[3] Bayesian Nonparametric Federated Learning of Neural Networks
[4] Measuring the Effects of Non-Identical Data Distribution for Federated Visual Classification
[5] https://leaf.cmu.edu/

---

> ### Author Response · Authors · 2020-11-19
> **Response to reviewer 4**
>
> Thanks for your careful reviewing. Please see below for clarification on the issues raised:
>
> - **Question 1**) ***Similarity to MAPPER***: We agree with R4 that in  **parallel** to the present work, a similar (but not exactly the same) interpolation  schema for personalization is proposed in [1],   however, there are still several significant differences and contributions (algorithmic, generalization, and numerical studies) that distinguishes this work from [1] as detailed below:
>     - **[a]** (***optimization***). Unlike [1] that does not provide an optimization algorithm to learn the model, we also propose a communication efficient algorithm to learn the personalized model and establish its convergence rates, and show that it enjoys reduced communication cost both in convex and nonconvex functions.
>     - **[b]** (***generalization***) The generalization guarantee in [1], does NOT show how the mixing parameter affects the generalization, while our bound gives explicit dependency on the mixing parameter, which is a significant difference to characterize the effectiveness of the mixed model in achieving better generalization performance (Remark 1 in the paper).
>     - **[c]** (***numerical studies***) We also have conducted thorough empirical studies to demonstrate the effectiveness of the proposed schema to overcome the curse of heterogeneity.
> - **Question 2**) ***Carefulness of empirical studies***: For the comments on the experiment part we should mention these points:
>     - **[2A]**. We agree that keeping two models will increase memory usage. However, we should mention that this is only for the training part, while for the inference we only need one model, which is the personalized model.
>     - **[2B]**. For your comment on the reported performance of FedAvg on MNIST and CIFAR10, we should emphasize the following points:
>         - **[a]**. The experiment for the CIFAR10 dataset on [2] is for the IID setting, while ours in a non-IID setting. Hence, by no means this comparison seems to be fair.
>         - **[b]**. The setup is different in the two papers. In ours, we have run the experiments for 100 rounds of communication as it is mentioned in the paper, with only having 1 epoch per each round of communication. However, in [2] the experiments on CIFAR10 and MNIST dataset are run for 2000 rounds of communication with 5-50 epochs per communication. Since with $\alpha=0$ we will reduce to the simple FedAvg, we will have the same results given the same setup for the experiment.
>         - **[c]**. More importantly, the ultimate goal here is to show that with even far fewer number of communication rounds and while the global model's performance is not perfect yet, our personalized model can have the best generalization performance on the local validation data. With having more communications, and hence, a better global model, the personalized model will be better as well.
>         - **[d]**. Nonetheless, we will run the experiments for more communication rounds to show that our global model has the same performance as other baselines in FedAvg. We already have run experiments for the MNIST dataset with the same setup as in [1]. Due to the time limit, we have it run for 300 rounds of communication and will add the results to the paper once it is complete.
>     - **[2C]**. On the question on other datasets and how to split datasets to induce heterogeneity:
>         - **[a]**. We note that the results for applying our algorithm on EMNIST, or as it is called  FEMNIST dataset in [5], have been presented in Figure 5 in Appendix B of the original manuscript. The figure shows the clear advantage of APFL in that dataset as well.
>         - **[b]** Also, with regards to how to partition datasets among clients, we would like to thank you for bringing these papers to our attention. Per your suggestion, we are running experiments with the suggested distribution scheme in [3,4]. Initial results, for instance on the MNIST dataset with an MLP model, show consistency with the results of our distribution scheme. We have added new results in Appendix B, Table 3 (highlighted in blue). We will add complete results with this scheme to the paper once the experiments finish.
>         - **[c]** We disagree with your statement that a single global model can have good performance, since a global model is trained on multiple clients' dataset, then it cannot guarantee good performance on the local distribution (potential negative knowledge transfer among datasets due to heterogeneity), as we have shown in experiments. If you are referring to the global model trained only on one local client's data, it will not have good generalization performance either, since one client only has a small number of training data.

---

### Decision · Program_Chairs · 2021-01-07
**Final Decision**

**Decision:**

Reject

**Comment:**

The paper studies personalized federated learning, mixing a global model with locally trained models.  Reviewers agreed on the relevance of the problem and that the work contains valuable contributions, such as the generalization bounds.
After discussion, unfortunately consensus remained that the paper remains narrowly below the bar in the current form.
Concerns remained on novelty over the Mapper optimization algorithm which also has adaptivity to the local/global combination of models, the dependence of the generalization bound on the mixing parameter as it converges to the global model,
as well as on the strength of the experimental findings compared to well-known FedAvg and related method in a realistic benchmark environment (such as e.g. Leaf), since the dataset choice (and even more its partition among clients) is a crucial aspect for measuring personalization in a fair way. We hope the feedback helps to strengthen the paper for a future occasion.